# Post-Contextual-Bandit Inference

**Aurélien Bibaut**
Netflix
abibaut@netflix.com

**Maria Dimakopoulou**
Netflix
mdimakopoulou@netflix.com

**Nathan Kallus**
Cornell University and Netflix
kallus@cornell.edu

**Antoine Chambaz**
Université de Paris
antoine.chambaz@u-paris.fr

**Mark van der Laan**
University of California, Berkeley
laan@stat.berkeley.edu

## Abstract

Contextual bandit algorithms are increasingly replacing non-adaptive A/B tests in e-commerce, healthcare, and policymaking because they can both improve outcomes for study participants and increase the chance of identifying good or even best policies. To support credible inference on novel interventions at the end of the study, nonetheless, we still want to construct valid confidence intervals on average treatment effects, subgroup effects, or value of new policies. The adaptive nature of the data collected by contextual bandit algorithms, however, makes this difficult: standard estimators are no longer asymptotically normally distributed and classic confidence intervals fail to provide correct coverage. While this has been addressed in non-contextual settings by using stabilized estimators, variance-stabilized estimators in the contextual setting pose unique challenges that we tackle for the first time in this paper. We propose the Contextual Adaptive Doubly Robust (CADR) estimator, a novel estimator for policy value that is asymptotically normal under contextual adaptive data collection. The main technical challenge in constructing CADR is designing adaptive and consistent conditional standard deviation estimators for stabilization. Extensive numerical experiments using 57 OpenML datasets demonstrate that confidence intervals based on CADR uniquely provide correct coverage.

## 1 Introduction

Contextual bandits, where personalized decisions are made sequentially and simultaneously with data collection, are increasingly used to address important decision-making problems where data is limited and/or expensive to collect, with applications in product recommendation [Li et al., 2010], revenue management [Kallus and Udell, 2020, Qiang and Bayati, 2016], and personalized medicine [Tewari and Murphy, 2017]. Adaptive experiments, whether based on bandit algorithms or Bayesian optimization, are increasingly being considered in place of classic randomized trials in order to improve both the outcomes for study participants and the chance of identifying the best treatment allocations [Athey et al., 2018, Quinn et al., 2019, Kasy and Sautmann, 2021, Bakshy et al., 2018].

But, at the end of the study, we still want to construct valid confidence intervals on average treatment effects, subgroup effects, or the value of new personalized interventions. Such confidence intervals are, for example, crucial for enabling credible inference on the presence or absence of improvement of novel policies. However, due to the adaptive nature of the data collection, unlike classic randomized trials, standard estimates and their confidence intervals actually fail to provide correct coverage, that is, contain the true parameter with the desired confidence probability (*e.g.*, 95%). A variety of recent work has recognized this and offered remedies [Hadad et al., 2019, Luedtke and van der Laan, 2016], but only for the case of non-contextual adaptive data collection. Like classic

confidence intervals, when data comes from a contextual bandit – or any other context-dependent adaptive data collection – these intervals also fail to provide correct coverage. In this paper, we propose a novel asymptotically normal estimator for the value of a (possibly contextual) policy from *context-dependent* adaptively collected data. This asymptotic normality leads directly to the construction of valid confidence intervals.

Our estimator takes the form of a *stabilized* doubly robust estimator, that is, a weighted time average of an estimate of the so-called canonical gradient using plug in estimators for the outcome model, where each time point is inversely weighted by its estimated conditional standard deviation given the past. We term this the Contextual Adaptive Doubly Robust (CADR) estimator. We show that, given consistent conditional variance estimates which at each time point only depend on previous data, the CADR estimator is asymptotically normal, and as a result we can easily construct asymptotically valid confidence intervals. This normality is in fact robust to misspecifying the outcome model. A significant technical challenge is actually constructing such variance estimators. We resolve this using an adaptive variance estimator based on the importance-sampling ratio of current to past (adaptive) policies at each time point. We also show that we can reliably estimate outcome models from the adaptively-collected data so that we can plug them in. Extensive experiments using 57 OpenML datasets demonstrate the failure of previous approaches and the success of ours at constructing confidence intervals with correct coverage.

## 1.1 Problem Statement and Notation

**The data.** Our data consists of a sequence of observations indexed $t = 1, \ldots, T$ comprising of context $X(t) \in \mathcal{X}$, action $A(t) \in \mathcal{A}$, and outcome $Y(t) \in \mathcal{Y} \subset \mathbb{R}$ generated by an adaptive experiment, such as a contextual bandit algorithm. Roughly, at each round $t = 1, 2, \ldots, T$, an agent formed a contextual policy $g_t(a \mid x)$ based on all past observations, then observed an independently drawn context vector $X(t) \sim Q_{0,X}$, carried out an action $A(t)$ drawn from its current policy $g_t(\cdot \mid X(t))$, and observed an outcome $Y(t) \sim Q_{0,Y}(\cdot \mid A(t), X(t))$ depending only on the present context and action. The action and context measurable spaces $\mathcal{X}, \mathcal{A}$ are arbitrary, *e.g.*, finite or continuous. We assume outcomes are bounded and, without loss of generality, take $\mathcal{Y} = [-1, 1]$.

More formally, we let $O(t) := (X(t), A(t), Y(t))$ and make the following assumptions about the sequence $O(1), \ldots, O(T)$ comprising our dataset. First, we assume $X(t)$ is independent of all else given $A(t)$ and has a time-independent marginal distribution that we denote by $Q_{0,X}$. Second, we assume $A(t)$ is independent of all else given $O(1), \ldots, O(t-1), X(t)$ and we set $g_t(\cdot \mid X(t))$ to its (random) conditional distribution given $O(1), \ldots, O(t-1), X(t)$. Third, we assume $Y(t)$ is independent of all else given $X(t), A(t)$ and has a time-independent conditional distribution given $X(t) = x, A(t) = a$ that is denoted by $Q_{0,Y}(\cdot \mid A, X)$. The distributions $Q_{0,X}$ and $Q_{0,Y}$ are unknown, while the policies $g_t(a \mid x)$ are known, as would be the case when running an adaptive experiment. To simplify presentation we endow $\mathcal{A}$ with a base measure $\mu_{\mathcal{A}}$ (*e.g.*, counting for finite actions or Lebesgue for continuous actions) and identify policies $g_t$ with conditional densities with respect to (w.r.t.) $\mu_{\mathcal{A}}$. In the case of $K < \infty$ actions, policies are maps from $\mathcal{X}$ to the $K$-simplex.

Note that, as the agent updates its policy based on already collected observations, $g_t$ is a random $O(1), \ldots, O(t-1)$-measurable object. This is the major departure from the setting considered in other literature on off-policy evaluation, which only consider a fixed logging policy, $g_t = g$, that is independent of the data. See Section 1.2.

**The target parameter.** We are interested in inference on a *generalized average causal effect* expressed as a functional of the unknown distributions above, $\Psi_0 = \Psi(Q_{0,X}, Q_{0,Y})$, where for any distributions $Q_X, Q_Y$, we define

$$\Psi(Q_X, Q_Y) := \int y Q_X(dx) g^*(a \mid x) d\mu_{\mathcal{A}}(a) Q_Y(dy \mid a, x),$$

where $g^*(a \mid x) : \mathcal{A} \times \mathcal{X} \to [-G, G]$ is a given fixed, bounded function. Two examples are: (a) when $g^*$ is a policy (conditional density), then $\Psi_0$ is its value; (b) when $g^*$ is the difference between two policies then $\Psi_0$ is the difference between their values. A prominent example of the latter is when $\mathcal{A} = \{+1, -1\}$ and $g^*(a \mid x) = a$, which is known as the average treatment effect. If we include an indicator for $x$ being in some set, then we get the subgroup effect. Defining the conditional mean outcome,

$$\bar{Q}_0(a,x) := E_{Q_{0,Y}(\cdot \mid x,a)}[Y] = \int y Q_{0,Y}(dy \mid a,x),$$

we note that the target parameter only depends on $Q_{0,Y}$ via $\bar{Q}_0$, so we also overload notation and write $\Psi(Q_X, \bar{Q}) = \int \bar{Q}(a,x)Q_X(dx)g(a \mid x)d\mu_{\mathcal{A}}(a)$ for any function $\bar{Q} : \mathcal{A} \times \mathcal{X} \to \mathcal{Y}$. Note that when $|\mathcal{A}| < \infty$ and $\mu_{\mathcal{A}}$ is the counting measure, the integral over $a$ is a simple sum.

**Canonical gradient.**  We will make repeated use of the following function: for any conditional density $(a,x) \mapsto g(a \mid x)$, any probability distribution $Q_X$ over the context space $\mathcal{X}$, and any function $\bar{Q} : \mathcal{A} \times \mathcal{X}$, we define the function $D'(g, \bar{Q}) : \mathcal{O} \to \mathbb{R}$ by

$$D'(g, \bar{Q})(x,a,y) := \frac{g^*(a \mid x)}{g(a \mid x)}(y - \bar{Q}(a,x)) + \int \bar{Q}(a',x)g^*(a' \mid x)d\mu_{\mathcal{A}}(a').$$

Further, define $D(g, Q_X, \bar{Q}) = D'(g, Q_X, \bar{Q}) - \Psi(Q_X, \bar{Q})$, which coincides with the so-called canonical gradient of the target parameter $\Psi$ w.r.t. the usual nonparametric statistical model comprising all joint distributions over $\mathcal{O}$ [van der Vaart, 2000, van der Laan and Robins, 2003].

**Integration operator notation.**  For any policy $g$ and distributions $Q_X, Q_Y$, denote by $P_{Q,g}$ the induced distribution on $\mathcal{O} := \mathcal{X} \times \mathcal{A} \times \mathcal{Y}$. For any function $f : \mathcal{O} \to \mathbb{R}$, we use the integration operator notation

$$P_{Q,g}f = \int f(x,a,y)Q_X(dx)g(a \mid x)d\mu_{\mathcal{A}}(a)Q_Y(dy \mid a,x),$$

that is, the expectation w.r.t. $P_{Q,g}$ *alone*. Then, for example, for any $O(1), \ldots, O(s-1)$-measurable random function $f : \mathcal{O} \to \mathbb{R}$, we have that $P_{Q_0, g_s}f = E_{Q_0, g_s}[f(O(s)) \mid O(1), \ldots, O(s-1)]$.

## 1.2 Related Literature and Challenges for Post-Contextual-Bandit Inference

**Off-policy evaluation.**  In non-adaptive settings, where $g_t = g$ is fixed and does not depend on previous observations, common off-the shelf estimators for the mean outcome under $g^*$ include the Inverse Propensity Scoring (IPS) estimator [Beygelzimer and Langford, 2009, Li et al., 2011] and and the Doubly Robust (DR) estimator [Dudík et al., 2011, Robins et al., 1994]:

$$\widehat{\Psi}^{\text{IPS}} := \frac{1}{T}\sum_{t=1}^{T}D'(g,0), \qquad \widehat{\Psi}^{\text{DR}} := \frac{1}{T}\sum_{t=1}^{T}D'(g,\widehat{\bar{Q}})$$

where $\widehat{\bar{Q}}$ is an estimator of the outcome model $\bar{Q}_0(a,x)$. If we use cross-fitting to estimate $\widehat{\bar{Q}}$ [Chernozhukov et al., 2018], then both the IPS and DR estimators are unbiased and asymptotically normal, permitting straightforward inference using Wald confidence intervals (*i.e.*, $\pm 1.96$ of the estimated standard error). There also exist many variants of the IPS and DR estimators that, rather than plugging in the importance sampling (IS) ratios $(g^*/g_t)(A(t) \mid X(t))$ and/or outcome-model estimators, instead choose them directly with the aim to minimize error [*e.g.* Kallus, 2018, Farajtabar et al., 2018, Thomas and Brunskill, 2016, Wang et al., 2017, Kallus and Uehara, 2019].

**Inference challenges in adaptive settings.**  In the adaptive setting, it is easy to see that, if in the $t$th term for DR we use an outcome model $\widehat{\bar{Q}}_{t-1}$ fit using only the observations $O(1), \ldots, O(t-1)$, then both the IPS and DR estimators both remain unbiased. However, neither generally converges to a normal distribution. One key difference between the non-adaptive and adaptive settings is that the IS ratios $(g^*/g_t)(A(t) \mid X(t))$ can both diverge to infinity or converge to zero. As a result of this, the above two estimators may either be dominated by their first terms or their last terms. At a more theoretical level, this violates the classical condition of martingale central limit theorems that the conditional variance of the terms given previous observations stabilizes asymptotically.

**Stabilized estimators.**  The issue for inference due to instability of the DR estimator terms was recognized by Luedtke and van der Laan [2016] in another setting. They work in the non-adaptive setting but consider the problem of inferring the maximum mean outcome over all policies when the optimal policy is non-unique. Their proposal is a so-called *stabilized estimator*, in which each term is inversely weighted by an estimate of its conditional standard deviation given the previous terms.

This stabilization trick has been also been reused for off-policy inference from *non-contextual* bandit data by Hadad et al. [2019], as the stabilized estimator remains asymptotically normal, permitting inference. In their non-contextual setting, an estimate of the conditional standard deviation of the terms can easily be obtained by the inverse square root propensities. In contrast, in our *contextual* setting, obtaining valid stabilization weights is more challenging and requires a construction involving adaptive training on past data. Zhang et al. [2021] use similar weights to Hadad et al. [2019] in order to stabilize an $M$-estimator fit to adaptively collected data for the purpose of inference on a true model for $\bar{Q}_0$. While they do consider a contextual setting, their method requires correct specification, else the inferential target depends on the logging policies. We focus only on off-policy evaluation but do not require well-specified models.

**Concurrent work of Zhan et al. [2021].**  The paper of Zhan et al. [2021], which was published after the submission of our paper, is also motivated by establishing an unbiased and asymptotically normal estimator from data collected by a contextual bandit. Both our method and theirs rely on adaptively weighted DR scores and yield, under certain conditions, asymptotically normal estimators. However, the way they construct their adaptive weights is quite different from ours. In particular, their adaptive weights at time $t$ (a) do not incorporate an outcome model and do not rely on it and (b) depend only on the known propensity score, which is plugged-in. As a result of these different constructions, our analysis and theirs require different sets of conditions. While we require slightly more stringent conditions on the exploration rate and the complexity of the logging policy class than they do (our assumptions 5 and 6), they require some form of asymptotic stabilization of the logging policies (their conditions (8) and (12)). An instance where the logging policy does not stabilize is, for example, when various actions have the same expected reward on certain subsets of the context space. Our approach in this paper is to use the variance-stabilization weights for the contextual bandit setting, just like Hadad et al. [2019] do the in the non-contextual bandit setting and Luedtke and van der Laan [2016] do in an iid setting with multiple optimal policies. Although Zhan et al. [2021] also tackle the contextual bandit setting, they do not use the variance-stabilization weights; correspondingly their conditions and results apply in different, complementary settings. The key challenge we tackle in applying variance-stabilization weights in the contextual bandit setting is dealing with the need to use sequentially estimated weights.

## 1.3  Contributions

In this paper, we construct and analyze a stabilized estimator for policy evaluation from context-dependent adaptively collected data, such as the result of running a contextual bandit algorithm. This then immediately enables inference. After constructing a generic extension of the stabilization trick, the main technical challenge is to construct a sequence of estimators $\widehat{\sigma}_1, \ldots, \widehat{\sigma}_T$ of the conditional standard deviations that are both consistent and such that for each $t$, $\widehat{\sigma}_t$ only uses the previous data points $O(1), \ldots, O(t-1)$. We show in extensive experiments across a large set of contextual bandit environments that our confidence intervals uniquely achieve close to nominal coverage.

## 2  Construction and Analysis of the Generic Contextual Stabilized Estimator

In this section, we give a generic construction of a stabilized estimator in our contextual and adaptive setting. That is, given generic plug-ins for outcome model and conditional standard deviation. We then provide conditions under which the estimator is asymptotically normal, as desired. To develop CADR, we will then proceed to construct appropriate plug in estimators in the proceeding sections.

### 2.1  Construction of the Estimator

**Outcome and variance estimators.**  Our estimator uses a sequence $(\widehat{\bar{Q}}_t)_{t \geq 1}$ of estimators of the outcome model $\bar{Q}_0$, such that, for every $t$, $\widehat{\bar{Q}}_t$ is $O(1), \ldots, O(t)$-measurable, that is, is trained using *only* the data up to time $t$. A key part of our estimator is the conditional variance estimators. Also, we require estimates of the conditional standard deviation of the canonical gradient. Define:

$$\sigma_{0,t} := \sigma_{0,t}(g_t),$$
$$\text{where } \sigma_{0,t}^2(g) := \text{Var}_{Q_0,g}\left( D'(g, \widehat{\bar{Q}}_{t-1})(O(t)) \mid O(1), \ldots, O(t-1) \right).$$

Let $(\widehat{\sigma}_t)_{t\geq 1}$ be a given sequence of estimates of $\sigma_{0,t}$ such that $\widehat{\sigma}_t$ is $O(1), \ldots, O(t-1)$-measurable, that is, is estimated using *only* the data up to time $t$.

**The generic form of the estimator.** The generic contextual stabilized estimator is then defined as:

$$\widehat{\Psi}_T := \left(\frac{1}{T}\sum_{t=1}^T \widehat{\sigma}_t^{-1}\right)^{-1} \frac{1}{T}\sum_{t=1}^T \widehat{\sigma}_t^{-1} D'(g, \widehat{\bar{Q}}_{t-1}). \tag{1}$$

### 2.2 Asymptotic normality guarantees

We next characterize the asymptotic distribution of $\widehat{\Psi}_T$ under some assumptions.

**Assumption 1** (Non degenerate efficiency bound). $\inf_g P_{Q_0,g} D^2(g, \bar{Q}_0, Q_{0,X}) > 0$.

Assumption 1 states that there is no fixed logging policy $g$ such that the efficiency bound for estimation of $\Psi(\bar{Q}_0, Q_{0,X})$ in the nonparametric model, from i.i.d. draws of $P_{Q_0,g}$, is zero. If assumption 1 does not hold, there exists a logging policy $g$ such that, if $O = (X, A, Y) \sim P_{Q_0,g}$, then $(g^*(A \mid X)/g(A \mid X))Y$ equals $\Psi(\bar{Q}_0, Q_{0,X})$ with probability 1. In other words, if assumption 1 does not hold, there exists a logging policy $g$ such that $\Psi(\bar{Q}_0, Q_{0,X})$ can be estimated with no error with probability 1 from a single draw of $P_{Q_0,g}$. Thus, it is very lax. An easy sufficient condition for Assumption 1 is that the outcome model has nontrivial variance in that $\text{Var}_{Q_{0,X}}(\int \bar{Q}(a, X)g^*(a \mid x)d\mu_{\mathcal{A}}(a)) > 0$.

**Assumption 2** (Consistent standard deviation estimators.). $\widehat{\sigma}_t - \sigma_{0,t} \xrightarrow{t\to\infty} 0$ *almost surely.*

In the next section we will proceed to construct specific estimators $\widehat{\sigma}_t$ that satisfy Assumption 2, leading to our proposed CADR estimator and confidence intervals.

**Assumption 3** (Exploration rate). $\|\sup_{a\in\mathcal{A},x\in\mathcal{X}} g_t^{-1}(a \mid x)\|_\infty = \omega(t^{-1/2})$.

Assumption 3 requires that the exploration rate of the adaptive experiment does not decay too quickly. Crucially, Assumption 3 requires that we have nontrivial propensities of pulling any one arm in any one context, conditioned on the history. While this may be satisfied by inherently randomized algorithms such as $\epsilon$-greedy and Thompson sampling, this is not satisfied by algorithms such as UCB, which are deterministic once given the history. This limitation is the same as for Hadad et al. [2019], Zhan et al. [2021].

**Remark 1.** *One can weaken Assumption 3 assumption to just requiring $\omega(t^{-1})$ decay by instead slightly strengthening Assumption 1 to require that $\sigma_{0,t}^2 \gtrsim \|\sup_{a\in\mathcal{A},x\in\mathcal{X}} g_t^{-1}(a \mid x)\|_\infty$, which would be implied by $g^*$ placing some mass on actions with decaying propensity and $Y$ having everywhere-nontrivial conditional variance. Here $a_t \gtrsim b_t$ means that for some constant $c > 0$, we have $a_t \geq cb_t$ for all $t \geq 1$. However, as we will in fact strengthen Assumption 3 anyway in Assumption 6 below, which we will use in constructing estimators $\widehat{\sigma}_t$ that satisfy Assumption 2, we do not discuss this potential weakening of Assumption 3.*

Based on these assumptions, we have the following asymptotic normality result:

**Theorem 1.** *Denote $\Gamma_T := \left(T^{-1}\sum_{t=1}^T \widehat{\sigma}_t^{-1}\right)^{-1}$. Under Assumptions 1 to 3, it holds that*

$$\Gamma_T^{-1}\sqrt{T}\left(\widehat{\Psi}_T - \Psi_0\right) \xrightarrow{d} \mathcal{N}(0, 1).$$

**Remark 2.** *Theorem 1 does not require the outcome model estimator to converge at all. As we will see in Section 3, our conditional variance estimator does require that the outcome model converges to a fixed limit $\bar{Q}_1$, but this limit does not have to be the true outcome model $\bar{Q}_0$. In other words, consistency of the outcome model is not required at any point of our analysis.*

## 3 Construction of the Conditional Variance Estimator and CADR

We now tackle the construction of $\widehat{\sigma}_t$ satisfying our assumptions; namely, they must be adaptively trained only on past data at each $t$ and they must be consistent. Observe that $\sigma_{0,t}^2 = \sigma_0^2(g_t, \widehat{\bar{Q}}_{t-1})$,

---

**Algorithm 1** The CADR Estimator and Confidence Interval

---

**Input:** Data $O(1), \ldots, O(T)$, policies $g_1, \ldots, g_T$, target $g^*$, outcome regression estimator
**for** $t = 1, 2, \ldots, T$ **do**

    Train $\widehat{\bar{Q}}_{t-1}$ on $O(1), \ldots, O(t-1)$ using the outcome regression estimator

    Set $D'_{t,s} = D(g_s, \widehat{\bar{Q}}_{t-1})(O(t))$ for $s = t, \ldots, T$    // (note index order compared to next line)

    Set $\widehat{\sigma}_t^2 = \frac{1}{t-1} \sum_{s=1}^{t-1} \frac{g_t(A(s)|X(s))}{g_s(A(s)|X(s))}(D'_{s,t})^2 - \left(\frac{1}{t-1} \sum_{s=1}^{t-1} \frac{g_t(A(s)|X(s))}{g_s(A(s)|X(s))} D'_{s,t}\right)^2$

**end for**

Set $\Gamma_T = \left(\frac{1}{T} \sum_{t=1}^{T} \widehat{\sigma}_t^{-1}\right)^{-1}$

Return estimate $\widehat{\Psi}_T = \frac{\Gamma_T}{T} \sum_{t=1}^{T} \widehat{\sigma}_t^{-1} D'_{t,t}$ and confidence intervals $\mathrm{CI}_\alpha = [\widehat{\Psi}_T \pm \zeta_{1-\alpha/2} \Gamma_T / \sqrt{T}]$

---

where we define
$$\sigma_0^2(g, \bar{Q}) := \Phi_{0,2}(g, \bar{Q}) - (\Phi_{0,1}(g, \bar{Q}))^2, \quad \Phi_{0,i}(g, \bar{Q}) := P_{Q_0, g}(D')^i(g, \bar{Q}), \; i = 1, 2.$$

Designing an $O(1), \ldots, O(t-1)$-measurable estimator of $\sigma_{0,t}^2$ presents several challenges. First, while we can only use observations $O(1), \ldots, O(t-1)$ to estimate it, $\sigma_{0,t}^2$ is defined as a function of integrals w.r.t. $P_{Q_0, g_t}$, from which we have only one observation, namely $O(t)$. Second, our estimation target $\sigma_{0,t}^2 = \sigma_0(g_t, \widehat{\bar{Q}}_{t-1})$ is *random* as it depends on $g_t$ and $\widehat{\bar{Q}}_t$. Third, $g_t, \widehat{\bar{Q}}_t$ depend on the same observations $O(1), \ldots, O(t-1)$ that we have at our disposal to estimate $\sigma_{0,t}^2$.

**Representation via importance sampling.** We can overcome the first difficulty via importance sampling, which allows us to write $\Phi_{0,i}(g, \bar{Q})$, $i = 1, 2$ as integrals w.r.t. $P_{Q_0, g_s}$, $s = 1, \ldots, t-1$, *i.e.*, the conditional distributions of observations $O(s)$, $s = 1, \ldots, t-1$ given their respective past. Namely, for any $s \geq 1$, $i = 1, 2$, we have that
$$\Phi_{0,i}(g, \bar{Q}) = P_{Q_0, g_s} \frac{g}{g_s}(D')^i(g, \bar{Q}). \tag{2}$$

**Dealing with the randomness of the estimation target.** We now turn to second challenge. Since $\sigma_{0,t}^2$ can be written in terms of $\Phi_{0,i}(g_t, \widehat{\bar{Q}}_{t-1})$ for $i = 1, 2$, Eq. (2) suggests perhaps an approach based on sample averages of $(g_t/g_s)(D')^i(g_t, \widehat{\bar{Q}}_{t-1})$ over $s$. However, whenever $s < t$, the latter is an $O(1), \ldots, O(t-1)$-measurable function due to the dependence on $g_t$ and $\widehat{\bar{Q}}_t$. Namely, $P_{Q_0, g_s}\{(g_t/g_s)(D')^i(g_t, \widehat{\bar{Q}}_{t-1})\}$ does not coincide in general with the conditional expectation $E_{Q_0, g_s}[((g_t/g_s)(D')^i(g_t, \widehat{\bar{Q}}_{t-1}))(O(s)) \mid \bar{O}(s-1)]$, as would arise from a sample average. We now look at solutions to overcome this difficulty, considering first $\widehat{\bar{Q}}_{t-1}$ and then $g_t$.

**Dealing with the randomness of $\widehat{\bar{Q}}_{t-1}$.** We propose an estimator of $\sigma_0^2(g, \widehat{\bar{Q}}_{t-1})$ for any fixed $g$. While requiring that $\widehat{\bar{Q}}_{t-1}$ converges to the true outcome regression function $\bar{Q}_0$ is a strong requirement, most reasonable estimators will at least converge to some fixed limit $\bar{Q}_1$. As a result, under an appropriate stochastic convergence condition on $(\widehat{\bar{Q}}_{t-1})_{t \geq 1}$, $\Phi_{0,i}(g, \widehat{\bar{Q}}_{t-1})$ can be reasonably approximated by the corresponding Cesaro averages, defined for $i = 1, 2$ as

$$\bar{\Phi}_{0,i,t}(g) := \frac{1}{t-1} \sum_{s=1}^{t-1} \Phi_{0,i}(g, \widehat{\bar{Q}}_{s-1}) = \frac{1}{t-1} \sum_{s=1}^{t} E_{Q_0, g_s}\left[((g/g_s)(D')^i(g, \widehat{\bar{Q}}_{s-1}))(O(s)) \mid \bar{O}(s-1)\right].$$

These are easy to estimate from the corresponding sample averages, defined for $i = 1, 2$ as

$$\widehat{\Phi}_{i,t}(g) := \frac{1}{t-1} \sum_{s=1}^{t} ((g/g_s)(D')^i(g, \widehat{\bar{Q}}_{s-1}))(O(s)),$$

since for each $i = 1, 2$, the difference $\widehat{\Phi}_{i,t}(g) - \bar{\Phi}_{0,i,t}(g)$ is the average of a martingale difference sequence (MDS). We then define our estimator of $\sigma_0^2(g, \widehat{\bar{Q}}_{t-1})$ as

$$\widehat{\sigma}_t(g) := \widehat{\Phi}_{2,t}(g) - (\widehat{\Phi}_{1,t}(g))^2. \tag{3}$$

**From fixed $g$ to random $g_t$.** So far, we have proposed and justified the construction of $\widehat{\sigma}_t(g)$ as an estimator of $\sigma_{0,t}(g, \widehat{Q}_{-1})$ for a fixed $g$. We now discuss conditions under which $\widehat{\sigma}_t(g_t)$ is valid estimator of $\sigma_{0,t}(g_t, \widehat{Q}_{-1})$. When $g$ is fixed, for each $i = 1, 2$, the error $\widehat{\Phi}_{i,t}(g) - \Phi_{0,i,t}(g, \widehat{Q}_{t-1})$ decomposes as the sum of the MDS average $\widehat{\Phi}_{i,t}(g) - \bar{\Phi}_{0,i,t}(g)$ and of the Cesaro approximation error $\bar{\Phi}_{0,i,t}(g) - \Phi_{0,i}(g, \widehat{Q}_{t-1})$. Both differences are straightforward to bound. For a random $g_t$, the term $\widehat{\Phi}_{i,t}(g_t) - \bar{\Phi}_{0,i,t}(g_t)$ is no longer an MDS average. Fortunately, under a complexity condition on the logging policy class $\mathcal{G}$, we can bound the supremum of the martingale empirical processes $\{|\widehat{\Phi}_{i,t}(g) - \bar{\Phi}_{0,i,t}(g)| : g \in \mathcal{G}\}$, which in turn gives us a bound on $|\widehat{\Phi}_{i,t}(g_t) - \bar{\Phi}_{0,i,t}(g_t)|$.

**Consistency guarantee for $\widehat{\sigma}_t^2$.** Our formal consistency result relies on the following assumptions.

**Assumption 4** (Outcome regression estimator convergence). *There exists $\beta > 0$, and a fixed function $\bar{Q}_1 : \mathcal{A} \times \mathcal{X} \to \mathbb{R}$ such that $\|\widehat{Q}_t - \bar{Q}_1\|_{1, Q_{0,X}, g^*} = O(t^{-\beta})$ almost surely.*

The next assumption is a bound on the bracketing entropy (see, *e.g.*, [van der Vaart and Wellner, 1996] for definition) of the logging policy class.

**Assumption 5** (Complexity of the logging policy class). *There exists a fixed reference function $g^{\mathrm{ref}}(a \mid x)$, a class of conditional densities $\mathcal{G}$, and some $p > 0$ such that $g_t \in \mathcal{G} \, \forall t \geq 1$ almost surely, $\sup_{g \in \mathcal{G}} \|g/g^{\mathrm{ref}}\|_\infty \leq G$, $\|g^*/g^{\mathrm{ref}}\|_\infty \leq G$, and*

$$\log N_{[]}(\epsilon, \mathcal{G}/g^{\mathrm{ref}}, \|\cdot\|_{2, Q_{0,X}, g^{\mathrm{ref}}}) \lesssim \epsilon^{-p},$$

*where $\mathcal{G}/g^{\mathrm{ref}} := \{g/g^{\mathrm{ref}} : g \in \mathcal{G}\}$.*

Next, we require a condition on the exploration rate that is stronger than Assumption 3.

**Assumption 6** (Exploration rate (stronger)). *$\|\sup_{a \in \mathcal{A}, x \in \mathcal{X}} g^{\mathrm{ref}}(a \mid x)/g_t(a \mid x)\|_\infty \lesssim t^{\alpha(\beta,p)}$, where $\alpha(\beta, p) := \min(1/(3+p), 1/(1+2p), \beta)$.*

**Theorem 2.** *Suppose that Assumptions 4 to 6 hold. Then, $\widehat{\sigma}_t^2 - \sigma_{0,t}^2 = o(1)$ almost surely.*

**Remark 3.** *While we theoretically require the existence of a logging policy class $\mathcal{G}$ with controlled complexity, we do not actually need to know $\mathcal{G}$ to construct our estimator. Moreover, while we require a bound on the bracketing entropy of the logging policy class $\mathcal{G}$, we impose no restriction on the outcome regression model complexity, permitting us to use flexible black-box regression methods. Usually we simply take $g^{\mathrm{ref}}(a \mid x) = 1$ so the boundedness in Assumption 5 is simply boundedness of densities, but the above allows for slightly more flexibility in cases where we do not necessarily explore all actions equally well in Assumption 6.*

**Remark 4.** *Assumption 4 requires $(\widehat{Q}_t)$ to be a sequence of regression estimator, such that for every $t \geq 1$, $\widehat{Q}_t$ is fitted on $O(1), \ldots, O(t)$ and for which we can guarantee a rate of convergence to some fixed limit $\bar{Q}_1$. Note that this can at first glance pose a challenge since observations $O(1), \ldots, O(t)$ are adaptively collected. In the appendix, we give guarantees for outcome regression estimation over a nonparametric model using an importance sampling weighted empirical risk minimization.*

**Remark 5.** *Assumptions 5 and 6 put restrictions on the logging policy, that is, the contextual bandit algorithm that was run. Assumption 5 is not very restrictive, especially for large $p$. It is generally satisfied for $p$ arbitrarily close to 0 for $\epsilon$-greedy with outcome regressions in a VC-subgraph class such as trees or for Thompson sampling with parametric models. For larger $p$ it can accommodate very general policy classes, even non-Donsker classes ($p > 2$). Assumption 6, on the other hand, may be restrictive as it can require a lot of exploration. While it appears that at least $\omega(1/t)$ exploration might be necessary for some normality as in Theorem 1 (see Remark 1), the rates needed in Assumption 6 may be too strong than actually needed. We therefore view it as a high-level assumption, as it does not change the method itself and it may not need to be satisfied in practice, and we leave it as an open question how to tighten our analysis to be able to rely on less exploration.*

**CADR asymptotics.** Our proposed CADR estimator is now given by plugging our estimates $\widehat{\sigma}_t$ from Eq. (3) into Eq. (1), as summarized in Algorithm 1 As an immediate corollary of Theorems 1 and 2 we have our main guarantee for this final estimator, showing CADR is asymptotically normal, whence we immediately obtain asymptotically valid confidence intervals.

**Corollary 1** (CADR Asymptotics and Inference). *Suppose that Assumptions 1 and 4 to 6 hold. Let $\widehat{\sigma}_t$ be given as in Eq. (3). Denote $\Gamma_T := \left(T^{-1}\sum_{t=1}^{T}\widehat{\sigma}_t^{-1}\right)^{-1}$. Then,*

$$\Gamma_T^{-1}\sqrt{T}\left(\widehat{\Psi}_T - \Psi_0\right) \xrightarrow{d} \mathcal{N}(0,1).$$

*Moreover, letting $\zeta_\alpha$ denote the $\alpha$-quantile of the standard normal distribution,*

$$\Pr\left[\Psi(Q_{0,X},\bar{Q}_0) \in \left[\widehat{\Psi}_T \pm \zeta_{1-\alpha/2}\Gamma_T/\sqrt{T}\right]\right] \xrightarrow{T\to\infty} 1-\alpha.$$

## 4 Empirical Evaluation

In this section and in sections E.2-E.5 of the appendix, we present an extensive and reproducible[1] empirical evaluation on public datasets that demonstrates the robustness of CADR confidence intervals using contextual bandit data with comparison to several baselines.

To produce the 8 figures in the paper (Figure 1 in the main paper and Figures 2-8 in the appendix) we worked with: 57-72 real-world classification datasets (OpenML Curated Classification benchmarking suite 2018 [Bischl et al., 2017]) converted to contextual bandit datasets (wide-spread evaluation approach in the offline and online contextual bandit literature); 4 different target policies; 6 estimators (CADR, and 5 other popular baseline estimators in the literature); 8 different training procedures (sequential cross-fitting vs. cross-time cross-fitting in Figures 3 and 3; misspecified vs. well-specified outcome model family in Figures 4 and 5; weighted vs. unweighted outcome model fitting in Figures 6 and 7; large data vs. small data in Figures 3 and 8). For each (dataset, target policy, estimator, training procedure)-combination, 64 simulations were run to obtain and report tight standard errors of the estimators' performance.

In the subsections that follow, we present our approach and the main results.

### 4.1 Baseline Estimators

We compare CADR to several benchmarks (our experiments focus on the case of finitely-many actions, $\mathcal{A} = \{1,\ldots,K\}$). All baselines take the following form for a choice of $w_t, \omega_t, \widehat{\bar{Q}}_t$:

$$\widehat{\Psi}_T = \left(\frac{1}{T}w_t\right)^{-1}\frac{1}{T}\sum_{t=1}^{T}w_t\tilde{D}_t', \quad \text{CI}_\alpha = \left[\widehat{\Psi}_T \pm \zeta_{1-\alpha/2}\sqrt{\frac{\sum_{t=1}^{T}w_t^2(\tilde{D}_t' - \widehat{\Psi})^2}{\left(\sum_{t=1}^{T}w_t\right)^2}}\right],$$

where $\tilde{D}_t' = \omega_t(Y(t) - \widehat{\bar{Q}}_{t-1}(A(t),X(t))) + \sum_{a=1}^{K}\widehat{\bar{Q}}_{t-1}(a,X(t))g^*(a\mid X(t))$.

The Direct Method (DM) sets $w_t = 1, \omega_t = 0$ and fits $\widehat{\bar{Q}}_{t-1}(a,\cdot)$ by running some regression method for each $a$ on the data $\{(X(s),Y(s)) : 1 \leq s \leq t-1, A(s) = a\}$. We will use either linear regression or decision-tree regression, both using default `sklearn` parameters. Note that even in non-contextual settings, where $\widehat{\bar{Q}}_{t-1}$ is a simple per-arm sample average, $\widehat{\bar{Q}}_{t-1}$ may be biased due to adaptive data collection [Xu et al., 2013, Luedtke and van der Laan, 2016, Bowden and Trippa, 2017, Nie et al., 2018, Hadad et al., 2019, Shin et al., 2019]. Inverse Propensity Score Weighting (IPW) sets $w_t = 1, \omega_t = (g^*/g_t)(A(t)\mid X(t)), \widehat{\bar{Q}}_t = 0$. Doubly Robust (DR) sets $w_t = 1, \omega_t = (g^*/g_t)(A(t)\mid X(t))$ and fits $\widehat{\bar{Q}}_{t-1}$ as in DM. More Robust Doubly Robust (MRDR) [Farajtabar et al., 2018] is the same as DR but when fitting $\widehat{\bar{Q}}_{t-1}$ we reweight each data point by $\frac{g^*(A(s)|X(s))(1-g_s(A(s)|X(s)))}{g_s(A(s)|X(s))^2}$. None of the above are generally asymptotically normal under adaptive data collection [Hadad et al., 2019]. Adaptive Doubly Robust (ADR; a.k.a. stabilized one-step estimator for multi-armed bandit data) [Luedtke and van der Laan, 2016, Hadad et al., 2019] is the same as DR but sets $w_t = g_t^{-1/2}(A(t)|X(t))$. ADR is unbiased and asymptotically normal for multi-armed bandit logging policies but is biased for context-measurable adaptive logging policies, which is the focus of this paper. Finally, note that our proposal CADR takes the same form as DR but with $w_t = \widehat{\sigma}_t^{-1}$ using our adaptive conditional standard deviation estimators $\widehat{\sigma}_t$ in Eq. (3).

---

[1]The code can be found at `https://github.com/mdimakopoulou/post-contextual-bandit-inference`.

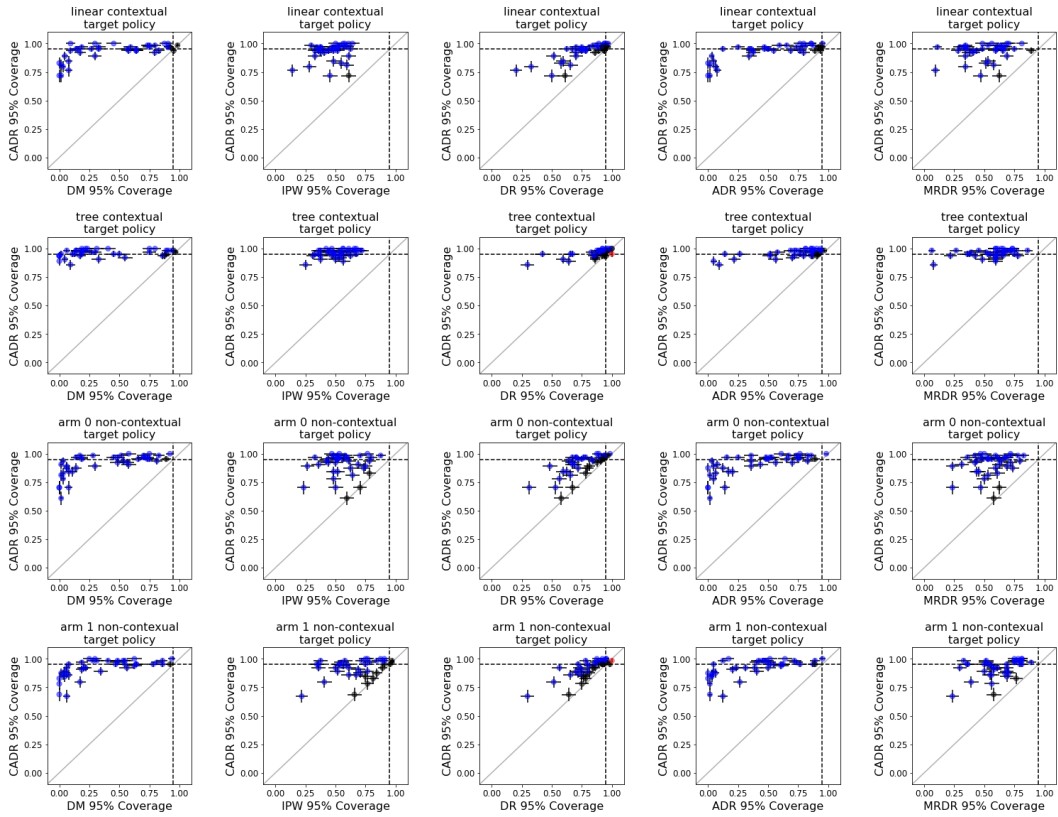

Figure 1: Comparison of CADR estimator against DM, IPW, DR, ADR and MRDR w.r.t. 95% confidence interval coverage on 57 OpenML-CC18 datasets and 4 target policies.

## 4.2 Contextual Bandit Data from Multiclass Classification Data

To construct our data, we turn $K$-class classification tasks into a $K$-armed contextual bandit problems [Dudík et al., 2014, Dimakopoulou et al., 2017, Su et al., 2019], which has the benefits of reproducibility using public datasets and being able to make uncontroversial comparisons using actual ground truth data with counterfactuals. We use the public OpenML Curated Classification benchmarking suite 2018 (OpenML-CC18; BSD 3-Clause license) [Bischl et al., 2017], which has datasets that vary in domain, number of observations, number of classes and number of features. Among these, we select the classification datasets which have less than 100 features. This results in 57 classification datasets from OpenML-CC18 used for evaluation and Table 1 in section E.1 of the appendix summarizes the characteristics of these datasets.

Each dataset is a collection of pairs of covariates $X$ and labels $L \in \{1, \dots, K\}$. We transform each dataset to the contextual bandit problem as follows. At each round, we draw $X(t), L(t)$ uniformly at random with replacement from the dataset. We reveal the context $X(t)$ to the agent, and given an arm pull $A(t)$, we draw and return the reward $Y(t) \sim \mathcal{N}(\mathbf{1}\{A(t) = L(t)\}, 1)$. To generate our data, we set $T = 10000$ and use the following $\epsilon$-greedy procedure. We pull arms uniformly at random until each arm has been pulled at least once. Then at each subsequent round $t$, we fit $\widehat{Q}_{t-1}$ using the data up to that time in the same fashion as used for the DM estimator above using decision-tree regressions. We set $\tilde{A}_x(t) = \arg\max_{a=1,\dots,K} \widehat{Q}_{t-1}(a, X(t))$ and $\epsilon_t = 0.01 \cdot t^{-1/3}$. We then let $g_t(a \mid x) = \epsilon_t/K$ for $a \neq \tilde{A}_x(t)$ and $g_t(\tilde{A}_x(t) \mid x) = 1 - \epsilon_t + \epsilon_t/K$. That is, with probability $\epsilon_t$ we pull a random arm, and otherwise we pull $\tilde{A}_{X(t)}(t)$.

We then consider four candidate policies to evaluate: (1) "arm 1 non-contextual": $g^*(1 \mid x) = 1$ and otherwise $g^*(a \mid x) = 0$ (note that the meaning of label "1" changes by dataset), (2) "arm 2 non-contextual": $g^*(2 \mid x) = 1$ and otherwise $g^*(a \mid x) = 0$, (3) "linear contextual": we sample a *new*

dataset of size $T$ using a uniform exploration policy, then fit $\widehat{\widehat{Q}}_T$ as above using linear regression, fix $a^* = \arg\max_{a \in \{1,\ldots,K\}} \widehat{\widehat{Q}}_T(a, x)$, and set $g^*(a^* \mid x) = 1$ and otherwise $g^*(a \mid x) = 0$, (4) "tree contextual": same as "linear contextual" but fit $\widehat{\widehat{Q}}_T$ using decision-tree regression.

### 4.3 Results

Figure 1 shows the comparison of CADR estimator against DM, IPW, DR, ADR, and MRDR w.r.t. coverage, that is, the frequency over 64 replications of the 95% confidence interval covering the true $\Psi_0$, for each of the 57 OpenML-CC18 datasets and 4 target policies. In each subfigure, each dot represents a dataset, the $y$-axis corresponds to the coverage of the CADR estimator and the $x$-axis corresponds to the coverage of one of the baseline estimators. The lines represent one standard error over the 64 replications. The dot is depicted in blue if for that dataset CADR has significantly (i.e., by more than one standard error) better coverage than the baseline estimator, in red if it has significantly worse coverage, and in black if the difference in coverage of both estimators is within one standard error. Results are averaged over 64 replications that we ran for each dataset, estimator, and target policy. The standard errors (computed as the sample standard deviation over the 64 replications divided by $\sqrt{64}$) are depicted and the 95% coverage targets are shown in dashed lines. In Figure 1, outcome models for CADR, DM, DR, ADR, and MRDR are fit using linear regression (with default `sklearn` parameters). In the appendix, we provide additional empirical results where we use decision-tree regressions, or where we use the MRDR outcome model for CADR, or where we use cross-fold estimation across time.

Across all of our experiments, we observe that the confidence interval of CADR has better coverage of the ground truth than any other baseline, which can be attributed to its asymptotic normality. The second best estimator in terms of coverage is DR. The advantages of CADR over DR are most pronounced when either (a) there is a mismatch between the logging policy and the target policy (*e.g.*, compare the 1st and 2nd rows in Figure 1; the tree target policy is most similar to the logging policy, which also uses trees) or (b) when the outcome model is bad (either due to model misspecification such as with a linear model on real data or due to small sample size).

## 5 Conclusions

Adaptive experiments hold great promise for better, more efficient, and even more ethical experiments. However, they complicate post-experiment inference, which is a cornerstone of drawing credible conclusions from controlled experiments. We provided here a novel asymptotically normal estimator for policy value and causal effects when data were generated from a contextual adaptive experiment, such as a contextual bandit algorithm. This led to simple and effective confidence intervals given by adding and subtracting multiples of the standard error, making contextual adaptive experiments a more viable option for experimentation in practice.

## 6 Societal Impact and Limitations

Adaptive experiments hold particular promise in settings where experimentation is costly and/or dangerous, such as in medicine and policymaking. By adapting treatment allocation, harmful interventions can be avoided, outcomes for study participants improved, and smaller studies enabled. Being able to draw credible conclusions from such experiments make them viable replacements for classic randomized trials. Our confidence intervals offer one way to do so. At the same time, and especially subject to our assumption of vanishing but nonzero exploration, these experiments must be subject to the same ethical guidelines as classic randomized experiments. Additionally, the usual caveats of frequentist confidence intervals hold here, such as its interpretation only as a guarantee over data collection, this guarantee only being approximate in finite samples when we rely on asymptotic normality, and the risks of multiple comparisons and of $p$-hacking. Finally, we note that our inference focused on an *average* quantity, as such it focuses on social welfare and need not capture the risk to individuals or groups. Subgroup analyses may therefore be helpful in complementing the analysis; these can be conducted by setting $g^*(a \mid x)$ to zero for some $x$'s. Future work may be necessary to further extend our results to conducting inference on risk metrics such as quantiles of outcomes.

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
