Supplementary Material for:

# Post-Contextual-Bandit Inference

Aurélien Bibaut, Maria Dimakopoulou, Nathan Kallus,
Antoine Chambaz, Mark van der Laan.

## A  Proof of the asymptotic normality of CADR

*Proof of theorem 1.* Recalling the definition of our estimator, we have that

$$\sqrt{T}(\widehat{\Psi}_T - \Psi(\bar{Q}_0, Q_{0,X}))$$

$$=\Gamma_T \frac{1}{\sqrt{T}} \sum_{t=1}^{T} \widehat{\sigma}_t^{-1} \left( \Psi(\widehat{\bar{Q}}_{t-1}, \widehat{Q}_{X,t-1}) - \Psi(\bar{Q}_0, Q_{0,X}) + D(g_t, \widehat{\bar{Q}}_{t-1}, \widehat{Q}_{X,t-1})(O(t)) \right)$$

$$=\Gamma_T \frac{1}{\sqrt{T}} \sum_{t=1}^{T} \widehat{\sigma}_t^{-1} \left( D(g_t, \widehat{\bar{Q}}_{t-1})(O(t)) - P_{Q_0,g_t} D(g_t, \widehat{\bar{Q}}_{t-1})(O(t)) \right)$$

$$=\Gamma_T \frac{1}{\sqrt{T}} \sum_{t=1}^{T} (\delta_{O(t)} - P_{Q_0,g_t}) \widehat{\sigma}_t^{-1}(D')(g_t, \widehat{\bar{Q}}_{t-1}),$$

where

$$(D')(g, \bar{Q}) := D(g, \bar{Q}, Q_{0,X}) + \Psi(\bar{Q}, Q_{0,X})$$
$$= \frac{g^*}{g}(\widetilde{y} - \bar{Q}) + \int g^*(a \mid \cdot)\bar{Q}(a, \cdot)d\mu_{\mathcal{A}}(a).$$

Note that

$$\mathrm{Var}_{Q_0,g_t}((D')(g_t, \widehat{\bar{Q}}_{t-1})(O(t)) \mid \bar{O}(t-1))$$
$$= \mathrm{Var}_{Q_0,g_t}(D(g_t, \widehat{\bar{Q}}_{t-1}, \widehat{Q}_{X,t-1})(O(t)) \mid \bar{O}(t-1))$$
$$= \sigma_{0,t}^2.$$

Let $Z_{t,T} := T^{-1/2}\widehat{\sigma}_t^{-1}(\delta_{O(t)} - P_{Q_0,g_t})(D')(g_t, \widehat{\bar{Q}}_{t-1}).$

Observe that $\{Z_{t,T} : t = 1, \ldots, T, \ T \geq 1\}$ is a martingale triangular array where, for every $T \geq 1$, $t \in [T]$, $Z_{t,T}$ is $\bar{O}(t)$-measurable. We will apply a martingale central limit theorem for triangular arrays to prove that $\sum_{t=1}^{T} Z_{t,T} \xrightarrow{d} \mathcal{N}(0,1)$. This will hold if we can check that

- the sum of conditional variances $V_T := \sum_{t=1}^{T} \mathrm{Var}_{Q_0,g_t}(Z_{t,T} \mid \bar{O}(t-1))$ converges in probability to 1,

- the Lindeberg condition is satisfied, that is, for any $\epsilon > 0$,

$$\sum_{t=1}^{T} E[Z_{t,T}^2 \mathbf{1}(Z_{t,T} > \epsilon) \mid \bar{O}(t-1)] \xrightarrow{p} 0.$$

**Convergence of the sum of conditional variances.**  We have that

$$V_T := \sum_{t=1}^{T} \mathrm{Var}_{Q_0,g_t}(Z_{T,t} \mid \bar{O}(t-1)) = \frac{1}{T}\sum_{t=1}^{T} \frac{\sigma_{0,t}^2}{\widehat{\sigma}_t^2} = 1 + \frac{1}{T}\sum_{t=1}^{T} \frac{\sigma_{0,t}^2 - \widehat{\sigma}_t^2}{\sigma_{0,t}^2 + (\sigma_{0,t}^2 - \widehat{\sigma}_t^2)}.$$

We now show that the terms of the right-hand side of the last equality above are $o(1)$ a.s. As $\sigma_{0,t}^2 - \widehat{\sigma}_t^2 = o(1)$ a.s. by assumption, it suffices to show that $\sigma_{0,t}$ is lower bounded by a positive constant.

For any fixed $Q_X$, $\bar{Q}$, $g$, we have that, $D(g, \bar{Q}, Q_X) = D(g, \bar{Q}_0, Q_{0,X}) + (D(g, \bar{Q}, Q_X) - D(g, \bar{Q}_0, Q_{0,X}))$. It is straightforward to check that $D(g, \bar{Q}_0, Q_{0,X})$ lies in the Hilbert space $T_1(Q_0) := L_2^0(Q_{0,Y}) \oplus L_2^0(Q_{0,X})$, where

$$L_2^0(Q_{0,Y}) := \left\{ h : \mathcal{O} \to \mathbb{R} : \forall (x, a) \in \mathcal{X} \times \mathcal{A}, \int h(x, a, y) dQ_{0,Y}(y \mid a, x) = 0 \right\},$$

$$\text{and } L_2^0(Q_{0,X}) := \left\{ h : \mathcal{X} \to \mathbb{R} : \int h(x) dQ_{0,X}(x) = 0 \right\},$$

while $D(g, \bar{Q}, Q_X) - D(g, \bar{Q}_0, Q_{0,X})$ lies in the Hilbert space

$$T_2(g) := L_2^0(g) := \left\{ h : \mathcal{X} \times \mathcal{A} \to \mathbb{R} : \forall x \in \mathcal{X}, \int h(x, a) g(a \mid x) d\mu_{\mathcal{A}}(a) = 0 \right\}.$$

It is straightforward to check that $T_1(Q_0)$ and $T_2(g)$ are orthogonal subspaces of $L_2(P_{Q_0,g})$. We have

$$\begin{aligned}
\sigma_0^2(g, \bar{Q}) &= \left\| D(g, \bar{Q}, Q_X) \right\|_{2,Q_0,g}^2 - \left( P_{Q_0,g} D(g, \bar{Q}, Q_X) \right)^2 \\
&\geq \left\| D(g, \bar{Q}, Q_X) \right\|_{2,Q_0,g}^2 \\
&= \left\| D(g, \bar{Q}_0, Q_{0,X}) \right\|_{2,Q_0,g}^2 + \left\| D(g, \bar{Q}, Q_X) - D(g, \bar{Q}_0, Q_{0,X}) \right\|_{2,Q_0,g}^2 \\
&\geq \left\| D(g, \bar{Q}_0, Q_{0,X}) \right\|_{2,Q_0,g}^2.
\end{aligned}$$

where we have used in the third line above that $D(g, \bar{Q}_0, Q_{0,X})$ and $D(g, \bar{Q}, Q_X) - D(g, \bar{Q}_0, Q_{0,X})$ lie in the orthogonal subspaces $T_1(Q_0)$ and $T_2(g)$. Therefore,

$$\begin{aligned}
\inf_{t \geq 1} \sigma_{0,t}^2 &:= \inf_{t \geq 1} \sigma_0^2(g_t, \hat{\bar{Q}}_{t-1}) \\
&\geq \inf_g \left\| D(g, \bar{Q}_0, Q_{0,X}) \right\|_{2,Q_0,g}^2 \\
&> 0,
\end{aligned}$$

where the last inequality is exactly assumption 1.

Therefore,

$$\left| \frac{\sigma_{0,t}^2 - \hat{\sigma}_t^2}{\sigma_{0,t}^2 + (\sigma_{0,t}^2 - \hat{\sigma}_t^2)} \right| \leq \frac{\left| \sigma_{0,t}^2 - \hat{\sigma}_t^2 \right|}{\inf_{s \geq 1} \sigma_{0,s}^2 + o(1)} = o(1)$$

almost surely. Therefore, by Cesaro summation, $V_T - 1 = o(1)$ a.s.

**Checking Lindeberg's condition.** Let $\epsilon > 0$. We want to show that

$$\sum_{t=1}^T E[Z_{t,T}^2 \mathbf{1}(Z_{t,T} \geq \epsilon)] \xrightarrow{p} 0.$$

Let $\delta_t := \inf_{a \in \mathcal{A}, x \in \mathcal{X}} g_t(a \mid x)$. From assumption 3, $\|\delta_t^{-1}\|_\infty = \omega(t^{1/2})$. We have that $Z_{t,T} = O(\delta_t^{-1} T^{-1/2} \hat{\sigma}_t^{-1})$. Notice that $\hat{\sigma}_t^{-1} = (\sigma_{0,t}^2 + \hat{\sigma}_t^2 - \sigma_{0,t}^2)^{-1/2} = O(1)$ a.s. since $\sigma_{0,t}^2 \geq C > 0$ and $\hat{\sigma}_t^2 - \sigma_{0,t}^2 = o(1)$. Therefore, $Z_{t,T} = O(\delta_t^{-1} T^{-1/2}) = o(1)$ a.s. and therefore, a.s., there exists $T_0(\epsilon)$ such that, for any $T \geq T_0(\epsilon)$, all the terms in the sum of the Lindeberg condition, are zero, which implies that the sum converges to zero almost surely.

Therefore, from the central limit theorem for martingale triangular arrays,

$$\sqrt{T} \Gamma_T^{-1} (\hat{\Psi}_T - \Psi_0) \xrightarrow{d} \mathcal{N}(0, 1).$$

$\square$

# B   Estimation of $\sigma_{0,t}^2$ via sequential importance sampling.

## B.1   Errors decomposition

In the following lemma, we provide a useful decomposition of the IS-weighted integrands that appear in the expressions of $\Phi_{0,1}(g, \bar{Q})$ and $\Phi_{0,2}(g, \bar{Q})$.

**Lemma 1.** *It holds that*

$$\frac{g}{g_s} D_1^2(g, \bar{Q}) = \frac{g^{\text{ref}}}{g_s} f_1(g, \bar{Q}) + \frac{g^{\text{ref}}}{g_s} f_2(\bar{Q}) + \frac{g^{\text{ref}}}{g_s} f_3(g, \bar{Q})$$

$$and \ \frac{g}{g_s} D_1(g, \bar{Q}) = \frac{g^{\text{ref}}}{g_s} f_4(\bar{Q}) + \frac{g^{\text{ref}}}{g_s} f_5(g, \bar{Q}),$$

*where*

$$f_1(g, \bar{Q}) := \frac{(g^*/g^{\text{ref}})^2}{(g/g^{\text{ref}})} (\widetilde{y} - \bar{Q})^2,$$

$$f_2(\bar{Q}) := 2(g^*/g^{\text{ref}})(\widetilde{y} - \bar{Q}) \int g^*(a \mid \cdot) \bar{Q}(a, \cdot) d\mu_{\mathcal{A}}(a),$$

$$f_3(g, \bar{Q}) := (g/g^{\text{ref}}) \left( \int g^*(a \mid \cdot) \bar{Q}(a, \cdot) d\mu_{\mathcal{A}}(a) \right)^2,$$

$$f_4(\bar{Q}) := (g^*/g^{\text{ref}})(\widetilde{y} - \bar{Q}),$$

$$f_5(g, \bar{Q}) := (g/g^{\text{ref}}) \int g^*(a \mid \cdot) \bar{Q}(a, \cdot) d\mu_{\mathcal{A}}(a).$$

The decomposition above motivates the following definitions.

$$\widehat{\Phi}_{1,t}^{(1)}(g) := \frac{1}{t-1} \sum_{s=1}^{t-1} \delta_{O(s)} \frac{g^{\text{ref}}}{g_s} f_1(g, \widehat{\bar{Q}}_{s-1}),$$

$$\widehat{\Phi}_{1,t}^{(2)} := \frac{1}{t-1} \sum_{s=1}^{t-1} \delta_{O(s)} \frac{g^{\text{ref}}}{g_s} f_2(\widehat{\bar{Q}}_{s-1}),$$

$$\widehat{\Phi}_{1,t}^{(3)}(g) := \frac{1}{t-1} \sum_{s=1}^{t-1} \delta_{O(s)} \frac{g^{\text{ref}}}{g_s} f_3(g, \widehat{\bar{Q}}_{s-1}),$$

$$\widehat{\Phi}_{2,t}^{(1)} := \frac{1}{t-1} \sum_{s=1}^{t-1} \delta_{O(s)} \frac{g^{\text{ref}}}{g_s} f_4(\widehat{\bar{Q}}_{s-1}),$$

$$\widehat{\Phi}_{2,t}^{(2)}(g) := \frac{1}{t-1} \sum_{s=1}^{t-1} \delta_{O(s)} \frac{g^{\text{ref}}}{g_s} f_5(g, \widehat{\bar{Q}}_{s-1}),$$

and

$$\bar{\Phi}_{0,1,t}^{(1)}(g) := \frac{1}{t-1} \sum_{s=1}^{t-1} P_{Q_0, g_s} \frac{g^{\text{ref}}}{g_s} f_1(g, \widehat{\bar{Q}}_{s-1}),$$

$$\bar{\Phi}_{0,1,t}^{(2)} := \frac{1}{t-1} \sum_{s=1}^{t-1} P_{Q_0, g_s} \frac{g^{\text{ref}}}{g_s} f_2(\widehat{\bar{Q}}_{s-1}),$$

$$\bar{\Phi}_{0,1,t}^{(3)}(g) := \frac{1}{t-1} \sum_{s=1}^{t-1} P_{Q_0, g_s} \frac{g^{\text{ref}}}{g_s} f_3(g, \widehat{\bar{Q}}_{s-1}),$$

$$\bar{\Phi}_{0,2,t}^{(1)} := \frac{1}{t-1} \sum_{s=1}^{t-1} P_{Q_0, g_s} \frac{g^{\text{ref}}}{g_s} f_4(\widehat{\bar{Q}}_{s-1}),$$

$$\bar{\Phi}_{0,2,t}^{(2)}(g) := \frac{1}{t-1} \sum_{s=1}^{t-1} P_{Q_0, g_s} \frac{g^{\text{ref}}}{g_s} f_5(g, \widehat{\bar{Q}}_{s-1}),$$

and

$$\Phi_{0,1}^{(1)}(g,\widehat{\widehat{Q}}_{t-1}) := P_{Q_0,g_t}\frac{g^{\mathrm{ref}}}{g_s}f_1(g,\widehat{\widehat{Q}}_{t-1}),$$

$$\Phi_{0,1}^{(2)}(\widehat{\widehat{Q}}_{t-1}) := P_{Q_0,g_t}\frac{g^{\mathrm{ref}}}{g_s}f_2(\widehat{\widehat{Q}}_{t-1}),$$

$$\Phi_{0,1}^{(3)}(g,\widehat{\widehat{Q}}_{t-1}) := P_{Q_0,g_t}\frac{g^{\mathrm{ref}}}{g_s}f_3(g,\widehat{\widehat{Q}}_{t-1}),$$

$$\Phi_{0,2}^{(1)}(\widehat{\widehat{Q}}_{t-1}) := P_{Q_0,g_t}\frac{g^{\mathrm{ref}}}{g_s}f_4(\widehat{\widehat{Q}}_{t-1}),$$

$$\Phi_{0,2}^{(3)}(g,\widehat{\widehat{Q}}_{t-1}) := P_{Q_0,g_t}\frac{g^{\mathrm{ref}}}{g_s}f_5(g,\widehat{\widehat{Q}}_{t-1}),$$

We have that

$$\widehat{\Phi}_{1,t} = \widehat{\Phi}_{1,t}^{(1)} + \widehat{\Phi}_{1,t}^{(2)} + \widehat{\Phi}_{1,t}^{(3)},\ \text{and}\ \widehat{\Phi}_{2,t} = \widehat{\Phi}_{2,t}^{(1)} + \widehat{\Phi}_{2,t}^{(2)},$$

$$\bar{\Phi}_{0,1,t}(g) = \bar{\Phi}_{0,1,t}^{(1)}(g) + \bar{\Phi}_{0,1,t}^{(2)} + \bar{\Phi}_{0,1,t}^{(3)}(g),\ \text{and}\ \bar{\Phi}_{0,2,t}(g) = \bar{\Phi}_{0,1,t}^{(1)} + \bar{\Phi}_{0,2,t}^{(2)}(g),$$

$$\Phi_{0,1}(g,\widehat{\widehat{Q}}_{t-1}) = \Phi_{0,1}^{(1)}(g,\widehat{\widehat{Q}}_{t-1}) + \Phi_{0,1}^{(2)}(\widehat{\widehat{Q}}_{t-1}) + \Phi_{0,1}^{(3)}(\widehat{\widehat{Q}}_{t-1}),$$

$$\Phi_{0,2}(g,\widehat{\widehat{Q}}_{t-1}) = \Phi_{0,2}^{(1)}(\widehat{\widehat{Q}}_{t-1}) + \Phi_{0,2}^{(2)}(g,\widehat{\widehat{Q}}_{t-1}).$$

We recall the decomposition of the errors $\widehat{\Phi}_{i,t}(g_t) - \Phi_{0,i}(g_t),\widehat{\widehat{Q}}_{t-1})$ in a martingale empirical process term and an approximation term:

$$\widehat{\Phi}_{i,t}(g_t) - \Phi_{0,i}(g_t),\widehat{\widehat{Q}}_{t-1}) = (\widehat{\Phi}_{i,t}(g_t) - \bar{\Phi}_{0,i,t}(g_t)) + (\bar{\Phi}_{0,i,t}(g_t) - \Phi_{0,i}(g_t,\widehat{\widehat{Q}}_{t-1})).$$

We treat the approximation terms in subsection B.4 further down. We further decompose the martingale empirical process terms here. We have that

$$\widehat{\Phi}_{1,t}(g_t) - \bar{\Phi}_{0,1,t}(g_t) = (\widehat{\Phi}_{1,t}^{(1)}(g_t) - \bar{\Phi}_{0,1,t}^{(1)}(g_t)) + (\widehat{\Phi}_{1,t}^{(2)} - \bar{\Phi}_{0,1,t}^{(2)})$$
$$+ (\widehat{\Phi}_{1,t}^{(3)}(g_t) - \bar{\Phi}_{0,1,t}^{(3)}(g_t)),$$
$$\text{and}\ \widehat{\Phi}_{2,t}(g_t) - \bar{\Phi}_{0,2,t}(g_t) = (\widehat{\Phi}_{2,t}^{(1)} - \bar{\Phi}_{0,2,t}^{(1)}) + (\widehat{\Phi}_{2,t}^{(2)}(g_t) - \bar{\Phi}_{0,2,t}^{(2)}(g_t)).$$

The two differences $\widehat{\Phi}_{1,t}^{(2)} - \bar{\Phi}_{0,1,t}^{(2)}$ and $\widehat{\Phi}_{2,t}^{(1)} - \bar{\Phi}_{0,2,t}^{(1)}$ are averages of martingale difference sequences, and can be analyzed with a martingale version of Bernstein's inequality. We bound the three other differences by the supremum of martingale empirical processes

## B.2 Control of the martingale empirical processes

Let, for any $\delta > 0$, $\widetilde{\mathcal{G}}(\delta) := \{g \in \mathcal{G} : \inf_{a,x} g(a \mid x) \geq \delta\}$. In the following lemma, we bound the sequential bracketing entropy of the classes of sequences of functions

$$\mathcal{F}_{k,t}(\delta) := \left\{(f_1(g,\widehat{\widehat{Q}}_{s-1}))_{s=1}^{t-1} : g \in \widetilde{\mathcal{G}}(\delta)\right\},$$

for $k = 1, 3, 5$.

**Lemma 2** (Sequential bracketing entropy bound). *Suppose that assumption 5 holds. Then, for $i = 3, 5$,*

$$\mathcal{N}_{[]}(\epsilon, \mathcal{F}_{1,t}(\delta), L_2(P_{Q_0,g^{\mathrm{ref}}})) \leq N_{[]}(G^{-2}\delta^2\epsilon, \mathcal{G}, L_2(P_{Q_0,g^{\mathrm{ref}}})).$$

*Suppose in addition that assumption 6 also holds. For $k = 3, 5$, we then have that*

$$\mathcal{N}_{[]}(\epsilon, \mathcal{F}_{k,t}(\delta), L_2(P_{Q_0,g^*})) \leq N_{[]}(\epsilon, \mathcal{G}, L_2(P_{Q_0,g^{\mathrm{ref}}})).$$

*Proof of lemma 2.* Observe that

$$0 \leq \int g^*(a \mid \cdot)\bar{Q}(a,\cdot)d\mu_{\mathcal{A}}(a) \leq 1, \qquad \text{and} \qquad 0 \leq (g^*/g^{\mathrm{ref}})^2(\widetilde{y} - \bar{Q})^2 \leq G^2.$$

Let $\{(l^j, u^j) : j \in [N]\}$ be an $\epsilon$-bracketing of $\widetilde{\mathcal{G}}(\delta)/g^{\mathrm{ref}}$ in $L_2(P_{Q_0,g^{\mathrm{ref}}})$. Without loss of generality, we can assume that $u^j \geq l^j \geq \delta g^{\mathrm{ref}}$ for every $j$. Let $g \in \widetilde{\mathcal{G}}(\delta)$. There exists $j$ such that $l^j \leq g \leq u^j$, and therefore,

$$f_1(u^j, \bar{Q}) \leq f_1(g, \bar{Q}) \leq f_1(l^j, \bar{Q})$$
$$\text{and } f_k(l^j, \bar{Q}) \leq f_k(g, \bar{Q}) \leq f_k(u^j, \bar{Q}), \text{ for } k = 3, 5.$$

We have that

$$\left\| f_1(l^j, \bar{Q}) - f_1(u^j, \bar{Q}) \right\|_{2, Q_0, g^{\mathrm{ref}}}$$
$$= \left\| (g^*/g^{\mathrm{ref}}) \frac{(u^j/g^{\mathrm{ref}}) - (l^j/g^{\mathrm{ref}})}{(u^j/g^{\mathrm{ref}})(l^j/g^{\mathrm{ref}})} (\widetilde{y} - \bar{Q})^2 \right\|_{2, Q_0, g^{\mathrm{ref}}}$$
$$\leq \delta^{-2} G^2 \epsilon$$

and for $k = 3, 5$, denoting $i_3 := 2$ and $i_5 := 1$, we have that

$$\left\| f_k(u^j, \bar{Q}) - f_k(l^j, \bar{Q}) \right\|_{2, Q_0, g^{\mathrm{ref}}}$$
$$= \left\| ((u^j/g^{\mathrm{ref}}) - (l^j/g^{\mathrm{ref}})) \int g^*(a \mid \cdot) \bar{Q}(a, \cdot) d\mu_{\mathcal{A}}(a) \right\|_{2, Q_0, g^{\mathrm{ref}}}$$
$$\leq \epsilon.$$

Therefore,

$$\rho((f_1(l^j, \widehat{\bar{Q}}_{s-1}) - f_1(u^j, \widehat{\bar{Q}}_{s-1}))_{s=1}^{t-1}) \leq \delta^{-2} G^2 \epsilon.$$

and, for $k = 3, 5$,

$$\rho((f_k(l^j, \widehat{\bar{Q}}_{s-1}) - f_k(u^j, \widehat{\bar{Q}}_{s-1}))_{s=1}^{t-1}) \leq \epsilon.$$

We have thus shown that an $\epsilon$-bracketing in $L_2(P_{Q_{0,X}, g^{\mathrm{ref}}})$ norm of $\mathcal{G}/g^{\mathrm{ref}}$ induces an $(G^2\delta^{-1}, L_2(P_{Q_0,g^{\mathrm{ref}}}))$ sequential bracketing of $\mathcal{F}_{1,t}(\delta)$, and $(\epsilon, L_2(P_{Q_0,g^{\mathrm{ref}}}))$ sequential bracketings of $\mathcal{F}_{3,t}(\delta)$ and $\mathcal{F}_{5,t}(\delta)$, which yields the claims. $\qquad\square$

**Lemma 3** (Uniform convergence of the martingale empirical process). *Suppose that assumptions 5 and 6 hold. Then, for any $(i, j) \in \{(1, 1), (1, 3), (2, 2)\}$*

$$\sup_{g \in \mathcal{G}} |\widehat{\Phi}_{i,t}^{(j)}(g) - \bar{\Phi}_{0,i,t}^{(j)}(g)| = o(1) \ a.s.$$

*Proof.* Let $\delta := \min_{s \in [t-1]} \inf_{(a,x) \in \mathcal{A} \times \mathcal{X}} g_s(a \mid x)$. In this proof, we treat $G$ as a constant, and we absorb it in the symbols $\lesssim, O, o,$ and $\widetilde{O}$ whenever we use them.

We treat the case $(i, j) = (1, 1)$ and the case $(i, j) \in \{(1, 3), (2, 2)\}$ separately.

**Case** $(i, j) = (1, 1)$. For any $g \in \mathcal{G}$, we have that $s \in [t - 1]$, $\|f_1(g, \widehat{\bar{Q}}_{s-1})\|_\infty \leq G^2\delta^{-1}$. Therefore, from theorem 3, for any $r^- \in (0, \delta^{-1}/2]$, it holds with probability at least $1 - 2e^{-x}$ that

$$\sup_{g \in \widetilde{\mathcal{G}}} \left| \widehat{\Phi}_{1,t}^{(1)}(g) - \bar{\Phi}_{0,1,t}^{(1)}(g) \right|$$
$$\lesssim r^- + \frac{1}{\sqrt{\delta t}} \int_{r^-}^{G^2\delta^{-1}} \sqrt{\log(1 + \mathcal{N}_{[]}(\epsilon, \mathcal{F}_{1,t}(\delta), L_2(P_{Q_0,g^{\mathrm{ref}}})))} d\epsilon$$
$$+ \frac{G^2\delta^{-1}}{\delta t} \log \mathcal{N}_{[]}(G^2\delta^{-1}, \mathcal{F}_{1,t}(\delta), L_2(P_{Q_0,g^{\mathrm{ref}}}))$$
$$+ G^2\delta^{-3/2}t^{-1/2}\sqrt{x} + G^2\delta^{-2}t^{-1}x.$$

Let $x_t := (\log t)^2$ and let $B_t$ the right-hand side above where we set $x$ to $x_t$. From Borel-Cantelli, we have that $\sup_{g \in \widetilde{\mathcal{G}}} |\widehat{\Phi}_{1,t}^{(1)}(g) - \bar{\Phi}_{0,1,t}^{(1)}(g)| = o(B_t)$ almost surely. Let us make $B_t$ explicit.

From lemma 2 and from assumption 5, we have that

$$\frac{G^2\delta^{-1}}{\delta t} \log(1 + \mathcal{N}_{[]}(G^2\delta^{-1}, \mathcal{F}_{1,t}(\delta), L_2(P_{Q_0,g^{\text{ref}}})))$$

$$\leq \frac{G^2\delta^{-1}}{\delta t} \log(1 + N_{[]}(\delta, \mathcal{G}/g^{\text{ref}}, L_2(P_{Q_0,g^{\text{ref}}})))$$

$$\lesssim G^2\delta^{-(2+p)}t^{-1}.$$

Let us now focus on the entropy integral. We have that

$$\int_{r^-}^{G^2\delta^{-1}} \sqrt{\log(1 + \mathcal{N}_{[]}(\epsilon, \mathcal{F}_{1,t}(\delta), L_2(P_{Q_0,g^{\text{ref}}})))}d\epsilon$$

$$\leq \int_{r^-}^{G^2\delta^{-1}} \sqrt{\log(1 + N_{[]}(G^{-2}\delta^2\epsilon, \mathcal{G}/g^{\text{ref}}, L_2(P_{Q_0,g^{\text{ref}}})))}d\epsilon$$

$$= G^2\delta^{-2} \int_{G^{-2}\delta^2 r^-}^{\delta} \sqrt{\log(1 + N_{[]}(u, \mathcal{G}/g^{\text{ref}}, L_2(P_{Q_0,g^{\text{ref}}})))}du$$

$$= G^2\delta^{-2} \int_{G^{-2}\delta^2 r^-}^{\delta} u^{-p/2}du$$

$$= \frac{G^2\delta^{-2}}{1-p/2}(\delta^{1-p/2} - (G^{-2}\delta^2 r^-)^{1-p/2},$$

for any $p \neq 2$. We choose $r^-$ so as to minimize the rate of $r^- + (\delta t)^{-1/2}\int_{r^-}^{G^2\delta^{-1}} \sqrt{\log(1 + \mathcal{N}_{[]}(\epsilon, \mathcal{F}_{1,t}(\delta), L_2(P_{Q_0,g^{\text{ref}}})))}d\epsilon$. We distinguish the cases $p < 2$ and $p > 2$.

**Case $p < 2$.** We just set $r^- = 0$, and we obtain

$$r^- + (\delta t)^{-1/2} \int_{r^-}^{G^2\delta^{-1}} \sqrt{\log(1 + \mathcal{N}_{[]}(\epsilon, \mathcal{F}_{1,t}(\delta), L_2(P_{Q_0,g^{\text{ref}}})))}d\epsilon \lesssim \delta^{-\frac{1}{2}(3+p)}t^{-\frac{1}{2}}.$$

Collecting the other terms yields that $B_t = \widetilde{O}(\delta^{-(3+p)/2}t^{-1/2} + t^{-1}\delta^{-(2+p)})$. From assumption 6, $\delta \gtrsim t^{-\alpha}$, with $\alpha < \min(1/(3+p), 1/(1+2p))$, and we therefore have $B_t = o(1)$.

**Case $p > 2$.** We pick $r^-$ so as to balance both terms of $r^- + (\delta t)^{-1/2}\int_{r^-}^{G^2\delta^{-1}} \sqrt{\log(1 + \mathcal{N}_{[]}(\epsilon, \mathcal{F}_{1,t}(\delta), L_2(P_{Q_0,g^{\text{ref}}})))}d\epsilon$. , that is we pick $r^-$ such that

$$r^- = t^{-1/2}G^p\delta^{-\frac{1}{2}(1+2p)} \iff r^- = G^2\delta^{-\frac{1}{p}(1+2p)}t^{-\frac{1}{p}}.$$

Collecting the other terms then yields $B_t = \widetilde{O}(\delta^{-\frac{1}{p}(1+2p)}t^{-\frac{1}{p}} + \delta^{-(2+p)}t^{-1})$. From assumption 6, $\delta \gtrsim t^{-\alpha}$, with $\alpha < \min(1/(3+p), 1/(1+2p))$, and we therefore have $B_t = o(1)$.

**Case $(i,j) \in \{(1,3),(2,2)\}$.** For any $g \in \mathcal{G}$, $s \in [t-1]$, $k = 3, 5$, we have that $\|f_k(g, \widehat{Q}_{s-1})\|_\infty \leq G$. Therefore, from theorem 3, for any $(i,j,k) \in \{(1,3,3),(2,2,5)\}$, for any $x > 0$, it holds with probability at least $1 - 2e^{-x}$ that

$$\sup_{g \in \mathcal{G}} \left|\widehat{\Phi}_{i,t}^{(j)} - \bar{\Phi}_{0,i,t}^{(j)}\right| \lesssim r^- + \frac{1}{\sqrt{\delta t}}\int_{r^-}^{G} \sqrt{\log(1 + \mathcal{N}_{[]}(\epsilon, \mathcal{F}_{k,t}(\delta), L_2(P_{Q_0,g^{\text{ref}}})))}d\epsilon$$

$$+ \frac{G}{\delta t} \log(1 + \mathcal{N}_{[]}(G, \mathcal{F}_{k,t}(\delta), L_2(P_{Q_0,g^{\text{ref}}})))$$

$$+ G\sqrt{\frac{x}{\delta t}} + G\frac{x}{\delta t}$$

$$\lesssim r^- + \frac{1}{1-p/2}\frac{1}{\sqrt{\delta t}}(G^{1-p/2} - (r^-)^{1-p/2}) + \frac{G^{1-p}}{\delta t} + G\sqrt{\frac{x}{\delta t}} + G\frac{x}{\delta t},$$

where we have used that, from lemma 2 and assumption 5, $\log(1+\mathcal{N}_{[]}(\epsilon, \mathcal{F}_{k,t}(\delta), L_2(P_{Q_0,g^{\mathrm{ref}}}))) \leq \log(1 + N_{[]}(\epsilon, \mathcal{G}/g^{\mathrm{ref}}, L_2(P_{Q_0,g^{\mathrm{ref}}})) \lesssim \epsilon^{-p}$. Setting $x$ to $x_t := (\log t)^2$ in the bound above and denote $B_t$ the resulting quantity. Applying Borel-Cantelli's lemma yields that $\sup_{g \in \mathcal{G}} \left| \widehat{\Phi}_{i,t}^{(j)} - \bar{\Phi}_{0,i,t}^{(j)} \right| = o(B_t)$ almost surely. We now give an explicit bound on $B_t$.

**Case $p \in (0, 2)$.** We set $r^- = 0$. We obtain $B_t = \widetilde{O}((\delta t)^{-1/2} + (\delta t)^{-1})$. Since from assumption 6, $\delta \gtrsim t^{-\alpha}$ with $\alpha < 1$, we have that $B_t = o(1)$.

**Case $p > 2$.** We set $r^- = (\delta t)^{-1/p}$. We have $B_t = \widetilde{O}((\delta t)^{-1/p} + (\delta t)^{-1})$. Since from assumption 6, $\delta \gtrsim t^{-\alpha}$ with $\alpha < 1$, we have that $B_t = o(1)$. $\square$

## B.3 High probability bound for the martingale terms

**Lemma 4.** *Suppose that there exists $\delta > 0$ such that $\|g^*/g_s\|_\infty \leq \delta^{-1}$ for every $s \in [t-1]$. Then For $(i,j) \in \{(1,2),(2,1)\}$, for any $x > 0$, it holds with probability $1 - 2e^{-x}$ that*

$$\left| \widehat{\Phi}_{i,t}^{(j)} - \bar{\Phi}_{0,i,t}^{(j)} \right| \lesssim \sqrt{\frac{x}{\delta t}} + \frac{x}{\delta t}, \tag{4}$$

*and for $(i,j) \in \{(1,3),(2,2)\}$, it holds with probability at least $1 - 2e^{-x}$ that*

$$\left| \widehat{\Phi}_{i,t}^{(j)} - \bar{\Phi}_{0,i,t}^{(j)} \right| \lesssim \sqrt{\frac{x}{t}} + \frac{x}{t}. \tag{5}$$

*Proof of lemma 4.* We have that

$$\widehat{\Phi}_{1,t}^{(2)} - \bar{\Phi}_{0,1,t}^{(2)} = \frac{1}{t-1} \sum_{s=1}^{t-1} (\delta_{O(s)} - P_{Q_0,g_s}) \frac{g^*}{g_s} f_2(\widehat{Q}_{s-1})$$

$$\text{and } \widehat{\Phi}_{2,t}^{(1)} - \bar{\Phi}_{0,2,t}^{(1)} = \frac{1}{t-1} \sum_{s=1}^{t-1} (\delta_{O(s)} - P_{Q_0,g_s}) \frac{g^*}{g_s} f_4(\widehat{Q}_{s-1}).$$

Therefore, both differences are the average of martingale difference sequences. For $k = 2, 4$, we have that $\|\frac{g^*}{g_s} f_k(\widehat{Q}_{s-1})\|_\infty \leq \delta^{-1}$ and $\|\frac{g^*}{g_s} f_k(\widehat{Q}_{s-1})\|_{2,Q_0,g^*} \leq \delta^{-1/2}$. Bernstein's inequality for martingale difference sequences then yields (4).

Concerning the other two differences, we have that

$$\widehat{\Phi}_{1,t}^{(3)} - \bar{\Phi}_{0,1,t}^{(3)} = \frac{1}{t-1} \sum_{s=1}^{t-1} Q_{0,X} f_3(\widehat{Q}_{s-1})^2$$

$$\text{and } \widehat{\Phi}_{2,t}^{(2)} - \bar{\Phi}_{0,2,t}^{(2)} = \frac{1}{t-1} \sum_{s=1}^{t-1} Q_{0,X} f_3(\widehat{Q}_{s-1}).$$

These two terms are the average of martingale sequences too, and since $\|f_3(\widehat{Q}_{s-1})\|_\infty \leq 1$, Bernstein's inequality for martingale difference sequences yields (5). $\square$

## B.4 Approximation error lemma

**Lemma 5.** *For any $\bar{Q}, \bar{Q}_1 : \mathcal{A} \times \mathcal{X} \to \mathbb{R}$, it holds that*

$$\max\left\{ \left| \Phi_{0,i}^{(j)}(\bar{Q}) - \Phi_{0,i}^{(j)}(\bar{Q}_1) \right| : (i,j) \in \{(1,2),(1,3),(2,1),(2,2)\} \right\} \leq 4 \left\| \bar{Q} - \bar{Q}_1 \right\|_{2,Q_0,g^*}$$

*and for any conditional densities $(a,x) \mapsto g(a \mid x)$, and $(a,x) \mapsto g_1(a \mid x)$ such that $g_1, g \geq \delta$ for some $\delta > 0$, it holds that*

$$\left| \Phi_{0,1}^{(1)}(\bar{Q}) - \Phi_{0,1}^{(1)}(\bar{Q}_1) \right| \leq \delta^{-2} \|g - g_1\|_{1,Q_{0,X},g^*} + \delta^{-1} \left\| \bar{Q} - \bar{Q}_1 \right\|_{1,Q_{0,X},g^*}.$$

*Proof.* We treat each case separately.

**Case** $(i, j) = (1, 2)$.

$$\left| \Phi_{0,1}^{(2)}(\bar{Q}) - \Phi_{0,1}^{(2)}(\bar{Q}_1) \right|$$
$$= 2 \left| P_{Q_0, g^*} \left\{ (\widetilde{y} - \bar{Q}) \langle g^*, \bar{Q} \rangle - (\widetilde{y} - \bar{Q}_1) \langle g^*, \bar{Q}_1 \rangle \right\} \right|$$
$$= 2 \left| P_{Q_0, g^*} \left\{ (\bar{Q}_1 - \bar{Q}) \langle g^*, \bar{Q} \rangle + (\widetilde{y} - \bar{Q}_1) \langle g^*, \bar{Q} - \bar{Q} \rangle \right\} \right|$$
$$\leq 4 \left\| \bar{Q} - \bar{Q}_1 \right\|_{1, Q_0, X, g^*}$$

**Case** $(i, j) = (1, 3)$.

$$\left| \Phi_{0,1}^{(3)}(\bar{Q}) - \Phi_{0,1}^{(3)}(\bar{Q}_1) \right|$$
$$= \left| Q_{0,X} \left\{ \langle g^*, \bar{Q} \rangle^2 - \langle g^*, \bar{Q}_1 \rangle^2 \right\} \right|$$
$$\leq 2 Q_{0,X} \langle g^*, |\bar{Q} - \bar{Q}_1| \rangle$$
$$= \left\| \bar{Q} - \bar{Q}_1 \right\|_{1, Q_0, X, g^*}$$

**Case** $(i, j) = (2, 1)$.

$$\left| \Phi_{0,2}^{(1)}(\bar{Q}) - \Phi_{0,2}^{(1)}(\bar{Q}_1) \right|$$
$$= \left| P_{Q_0, g^*} \left\{ (\widetilde{y} - \bar{Q}) - (\widetilde{y} - \bar{Q}_1) \right\} \right|$$
$$\leq \left\| \bar{Q} - \bar{Q}_1 \right\|)_{1, Q_0, X, g^*}$$

**Case** $(i, j) = (2, 2)$.

$$\left| \Phi_{0,2}^{(2)}(\bar{Q}) - \Phi_{0,2}^{(2)}(\bar{Q}_1) \right|$$
$$= \left| Q_{0,X} \left\{ \langle g^*, \bar{Q} \rangle - \langle g^*, \bar{Q}_1 \rangle \right\} \right|$$
$$\leq \left\| \bar{Q} - \bar{Q}_1 \right\|_{1, Q_0, X, g^*}$$

**Case** $(i, j) = (1, 1)$.

$$\left| \Phi_{0,1}^{(1)}(g, \bar{Q}) - \Phi_{0,1}^{(1)}(g_1, \bar{Q}_1) \right|$$
$$= \left| P_{Q_0, g^*} \left\{ \frac{g^*}{g}(\widetilde{y} - \bar{Q}) - \frac{g^*}{g}(\widetilde{y} - \bar{Q}_1) \right\} \right|$$
$$\leq \left| P_{Q_0, g^*} \left\{ \frac{1}{g g_1}(g - g_1) + \frac{1}{g}(\bar{Q} - \bar{Q}_1) \right\} \right|$$
$$\leq \frac{1}{\delta^2} \left\| g - g_1 \right\|_{1, Q_0, X, g^*} + \frac{1}{\delta} \left\| \bar{Q} - \bar{Q}_1 \right\|_{1, Q_0, X, g^*}.$$

$\square$

## B.5 Proof of theorem 2

*Proof of theorem 2.* As noted at the beginning of this section, the estimation error $\widehat{\sigma}_t^2 - \sigma_{0,t}^2$ decomposes as

$$\widehat{\sigma}_t^2 - \sigma_{0,t}^2 := \sum_{(i,j) \in \mathcal{S}} \widehat{\Phi}_{i,t}^{(j)} - \bar{\Phi}_{0,i,t}^{(j)} \tag{6}$$

$$+ \sum_{(i,j) \in \mathcal{S}} \bar{\Phi}_{0,i,t}^{(j)} - \Phi_{0,i}^{(j)}(g_t, \bar{Q}_1) \tag{7}$$

$$+ \sum_{(i,j) \in \mathcal{S}} \Phi_{0,i}^{(j)}(g_t, \bar{Q}_1) - \Phi_{0,i}^{(j)}(g_t, \widehat{\bar{Q}}_{t-1}). \tag{8}$$

The terms in line (6) are MDS averages or martingale empirical processes evaluated at $g_t$. Setting $x_t := (\log t)^2$ in lemma 4 and using Borel-Cantelli gives that the MDS averages are $o(1)$ almost surely. Lemma 3 gives that the martigale empirical process terms evaluated at $g_t$ are $o(1)$ almost surely as well.

From lemma 5, and assumptions 4 and 6,

$$\sum_{(i,j)\in\mathcal{S}} \Phi_{0,i}^{(j)}(g_t, \bar{Q}_1) - \Phi_{0,i}^{(j)}(g_t, \widehat{\bar{Q}}_{s-1})$$
$$=O(s^{\alpha-\beta}) \text{ a.s.}$$
$$=o(1) \text{ a.s..}$$

Therefore the third line above (8) is $o(1)$ almost surely, and by Cesaro summation, the second line above (7) is $o(1)$ almost surely as well. □

## C  Maximal inequality for importance sampling weighted martingale empirical processes

In this section, we restate a maximal inequality for so-called importance sampling martingale empirical processes from Bibaut et al. [2021]. We include it for our reader's convenience.

**Sequential bracketing entropy.**  Let $\Theta$ be a set, and let $T \geq 1$. For any $\theta \in \Theta$, let $(\xi_t(\theta))_{t=1}^T$ be a sequence of functions $\mathcal{O} \to \mathbb{R}$ such that for any $t \in [T]$, $\xi_t(\theta)$ is $\bar{O}(t-1)$-measurable. We denote

$$\Xi_T := \left\{ (\xi_t(\theta))_{t=1}^T : \theta \in \Theta \right\}.$$

Let $g^{\text{ref}}$ be a fixed reference policy. For any sequence $(f_t)_{t=1}^T$ of $\mathcal{O} \to \mathbb{R}$ functions such that $f_t$ is $\bar{O}(t-1)$-measurable for any $t$, we introduce the norm

$$\rho((f_t)_{t=1}^T) := \left( \frac{1}{T} \sum_{t=1}^T \|f_t\|_{2,Q_0,g^{\text{ref}}}^2 \right)^{1/2}.$$

Following the definition of van Handel [2011], we say that a collection of sequences of pairs of functions $\mathcal{O} \to \mathbb{R}$ of the form

$$\left\{ ((\lambda_t^j, v_t^j))_{t=1}^T : j \in [N] \right\}$$

forms an $(\epsilon, L(P_{Q,g^{\text{ref}}}))$ sequential bracketing of $\Xi_T$ if

- for any $t \in [T]$ and any $j \in [N]$, $\lambda_t^j$ and $v_t^j$ are $\bar{O}(t-1)$-measurable $\mathcal{O} \to \mathbb{R}$ functions,
- for any $\theta \in [\Theta]$, there exists $j \in [N]$ such that, for any $t \in [T]$, $\lambda_t^j \leq \xi_t(\theta) \leq v_t^j$.
- for any $j \in [N]$, $\rho((v_t^j - \lambda_t^j)_{t=1}^T) \leq \epsilon$.

We denote $\mathcal{N}_{[]}(\epsilon, \Xi_T, L_2(P_{Q,g^{\text{ref}}}))$ the cardinality of any $(\epsilon, L_2(P_{Q,g^{\text{ref}}})$ sequential bracketing of $\Xi_T$ of minimal cardinality.

**Importance sampling weighted martingale empirical process.**  We term importance sampling weighting martingale empirical processes stochastic processes of the form

$$\left\{ \frac{1}{T} \sum_{t=1}^T (\delta_{O(t)} - P_{Q_0,g_t}) \frac{g^{\text{ref}}}{g_t} \xi_t(\theta) : \theta \in \Theta \right\}.$$

The result below is theorem 1 from Bibaut et al. [2021].

**Theorem 3** (Maximal inequality for IS weighted martingale processes)**.**  *Suppose that*

- *there exists $\gamma > 0$ such that $\|g^*/g_t\|_\infty \leq \gamma$ for every $t \in [T]$,*

- *there exists $B > 0$ such that $\sup_{\theta \in \Theta} \|\xi_t(\theta)\|_\infty \leq B$ for every $t \in [T]$,*
- *there exists $p > 0$ such that*

$$\log \mathcal{N}_{[]}(\epsilon, \Xi_T, L_2(P_{Q,g^{\mathrm{ref}}})) \lesssim \epsilon^{-p}.$$

*Then, for any $r > 0$, $r^- \in [0, r/2]$ and $x > 0$, it holds with probability at least $1 - 2e^{-x}$ that*

$$\sup_{g \in \mathcal{G}} \left\{ \frac{1}{T} \sum_{t=1}^T (\delta_{O(t)} - P_{Q_0, g_t}) \frac{g^{\mathrm{ref}}}{g_t} \xi_t(\theta) : \theta \in \Theta, \rho((\xi_t(\theta))_{t=1}^T) \leq \epsilon \right\}$$

$$\lesssim r^- + \sqrt{\frac{\gamma}{T}} \int_{r^-}^r \sqrt{\log(1 + \mathcal{N}_{[]}(\epsilon, \Xi_T, P_{Q_0, g^{\mathrm{ref}}}))} d\epsilon + \frac{\gamma B}{T} \log(1 + \mathcal{N}_{[]}(r, \Xi_T, P_{Q_0, g^{\mathrm{ref}}}))$$

$$+ r \sqrt{\frac{\gamma x}{T}} + \frac{\gamma B x}{T}$$

# D High probability bound for IS weighted nonparametric least squares from adaptively collected data

Suppose $\mathcal{Y} \subseteq [-\sqrt{M}, \sqrt{M}]$ for some $M > 0$ and let $\bar{\mathcal{Q}}$ be a convex class of functions $\mathcal{A} \times \mathcal{X} \to \mathcal{Y}$. For any $\bar{Q} : \mathcal{A} \times \mathcal{X} \to \mathbb{R}$, and any $o = (x, a, y) \in \mathcal{O}$, let $\ell(\bar{Q}, o) := (y - \bar{Q}(a, x))^2$. Let $g^{\mathrm{ref}}$ be a fixed (as opposed to random) density w.r.t. some dominating measure $\mu$ on $\mathcal{A}$. For any $\bar{Q}$, define the corresponding population risk w.r.t. $P_{Q_0, g^{\mathrm{ref}}}$ as $R_0(\bar{Q}) := P_{Q_0, g^{\mathrm{ref}}} \ell(\bar{Q}, \cdot)$. Observe that the population risk can be rewritten in terms of the conditional distributions $(P_{Q_0, g_s})_{s=1}^t$ of observations $(O(s))_{s=1}^t$ given their respective past, via IS weighting:

$$R_0(\bar{Q}) := \frac{1}{t} \sum_{s=1}^t P_{Q_0, g_s} \frac{g^{\mathrm{ref}}}{g_s} \ell(\bar{Q}, \cdot).$$

We define the corresponding IS weighted empirical risk as

$$\widehat{R}_t(\bar{Q}) := \frac{1}{t} \sum_{s=1}^t \delta_{O(s)} \frac{g^{\mathrm{ref}}}{g_s} \ell(\bar{Q}, \cdot).$$

Let $\widehat{\bar{Q}}_t \in \arg\min_{\bar{Q} \in \bar{\mathcal{Q}}} \widehat{R}_t(\bar{Q})$ be an empirical risk minimizer over $\bar{\mathcal{Q}}$. In the upcoming theorem, we provide a high probability bound on the excess risk $R_0(\widehat{\bar{Q}}_t) - R_0(\bar{Q}_1)$. Our result requires the following assumptions.

**Assumption 7** (Entropy of the loss class)**.** *There exists $p > 0$ such that $\log N_{[]}(M\epsilon, \ell(\bar{\mathcal{Q}}), L_2(P_{Q_0, g^{\mathrm{ref}}})) \lesssim \epsilon^{-p}$, where $\ell(\bar{\mathcal{Q}}) := \{\ell(\bar{Q}) : \bar{Q} \in \bar{\mathcal{Q}}\}$.*

**Assumption 8** (Bounded IS ratios)**.** *There exists $\gamma_t > 0$ such that $\|g^*/g_s\|_\infty \leq \gamma_t$ for every $s = 1, \dots, t$.*

Theorem 4 in Bibaut et al. [2021] gives a high probability excess risk bound on the least squares estimator. We restate it here under the current notation for our reader's convenience.

**Theorem 4.** *Consider the setting of the current section, and suppose that 7 and 8 hold. Then, for any $x > 0$, it holds with probability $1 - 2e^{-x}$ that*

$$R(\widehat{\bar{Q}}_t) - \inf_{\bar{Q} \in \bar{\mathcal{Q}}} R(\bar{Q}) \lesssim M \begin{cases} \left(\frac{\gamma_t}{t}\right)^{\frac{1}{1+p/2}} + \frac{\gamma_t x}{t} & \text{if } p < 2, \\ \left(\frac{\gamma_t}{t}\right)^{\frac{1}{p}} + \frac{\gamma_t}{t} + \sqrt{\frac{\gamma_t x}{t}} + \frac{\gamma_t x}{t} & \text{if } p > 2. \end{cases}$$

# E Additional Empirical Results

## E.1 Data

We use the public OpenML Curated Classification benchmarking suite 2018 (OpenML-CC18; BSD 3-Clause license) [Bischl et al., 2017], which has 72 datasets that vary in domain, number of observations, number of classes and number of features.

| Samples | Count |
|---|---|
| $< 1000$ | 17 |
| $\geq 1000$ and $< 10000$ | 30 |
| $\geq 10000$ | 10 |

| Classes | Count |
|---|---|
| $= 2$ | 31 |
| $> 2$ and $< 10$ | 17 |
| $\geq 10$ | 9 |

| Features | Count |
|---|---|
| $\geq 2$ and $< 10$ | 14 |
| $\geq 10$ and $< 50$ | 34 |
| $\geq 50$ and $\leq 100$ | 9 |

Table 1: Characteristics of the 57 OpenML-CC18 datasets used for evaluation in the experiments of Figure 1 of the main paper (repeated as Figure 2 of the appendix) and of Figure 3.

| Samples | Count |
|---|---|
| $< 1000$ | 17 |
| $\geq 1000$ and $< 10000$ | 39 |
| $\geq 10000$ | 16 |

| Classes | Count |
|---|---|
| $= 2$ | 35 |
| $> 2$ and $< 10$ | 20 |
| $\geq 10$ | 17 |

| Features | Count |
|---|---|
| $\geq 2$ and $< 10$ | 14 |
| $\geq 10$ and $< 50$ | 43 |
| $\geq 50$ and $\leq 100$ | 15 |

Table 2: Characteristics of the 72 OpenML-CC18 datasets used for evaluation in the experiments of Figures 4-8 of the appendix.

Among these, we select the classification datasets which have less than 100 contextual features to produce Figure 1 of the main paper that uses sequential sample splitting for the training of the outcome model $\widehat{\widehat{Q}}_{t-1}$ of all estimators that use it. This results in 57 classification datasets from OpenML-CC18. Figure 3 and Figure 2 (same as Figure 1 of the main paper), which establish parity between the results of training $\widehat{\widehat{Q}}_{t-1}$ with cross-time-fitting vs. sequential sample splitting also use the subset of the 57 datasets with less that 100 contextual features. Table 1 summarizes the characteristics of these 57 datasets.

Figures 4-8 use cross-time-fitting training of the outcome model $\widehat{\widehat{Q}}_{t-1}$ of all estimators that use it and therefore use all 72 OpenML datasets. Table 2 summarizes the characteristics of all 72 datasets.

### E.2 Sequential Sample Splitting vs. Cross-Time-Fitting

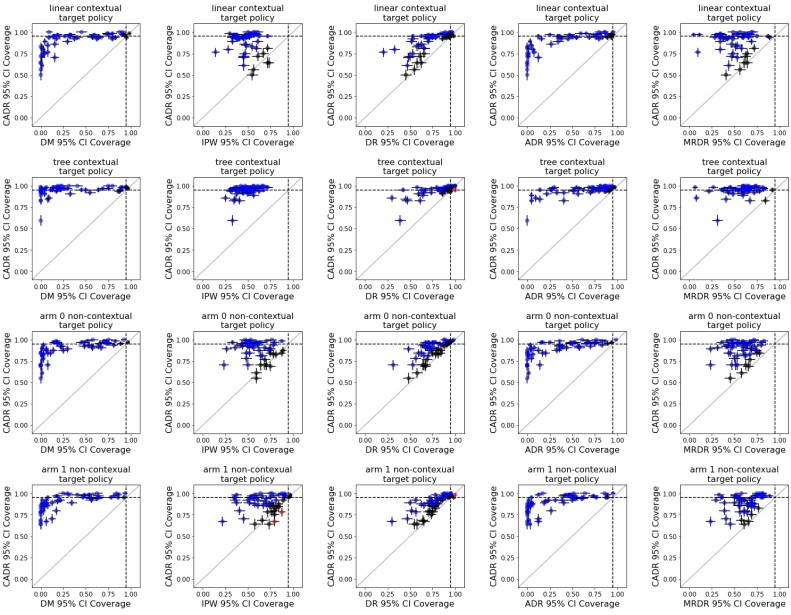

Figure 2: Comparison of CADR estimator against DM, IPW, DR, ADR, MRDR w.r.t 95% confidence interval coverage on 57 OpenML-CC18 datasets and 4 target policies with **sequential sample splitting** for training the linear outcome regression model of all estimators that use them.

The approach we proposed in the main text estimates $\widehat{\widehat{Q}}_{t-1}$ using only the data $O(1), \ldots, O(t-1)$. This means that potentially few data are available for earlier estimates. In this section, we empir-

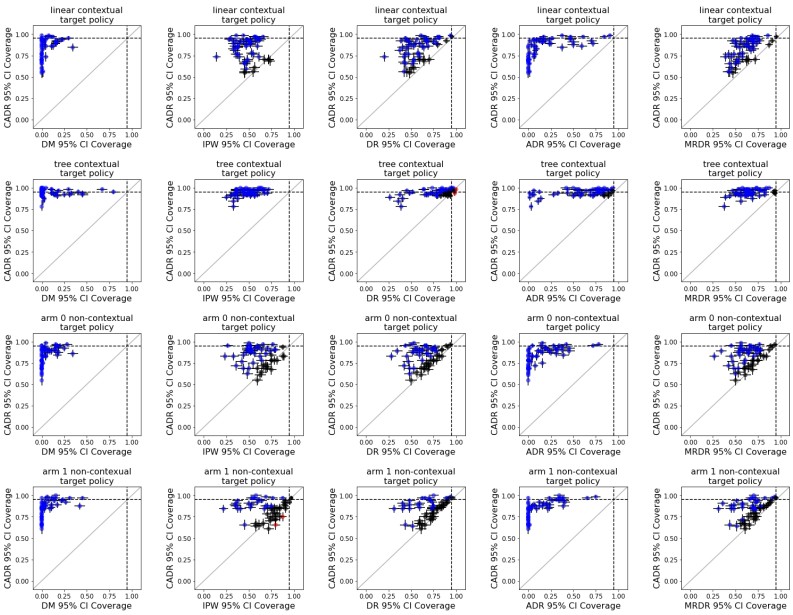

Figure 3: Comparison of CADR estimator against DM, IPW, DR, ADR, MRDR w.r.t. 95% confidence interval coverage on 57 OpenML-CC18 datasets and 4 target policies with **cross-fitting** for training the linear outcome regression model of all estimators that use them.

ically explore an alternative strategy for fitting $\widehat{\widehat{Q}}_{t-1}$ inspired by the cross-time-fitting procedure proposed in Kallus and Uehara [2019] and which would be theoretically justified under some sufficient mixing (which is not necessary for our sequential approach). Specifically, we split our data into $F = 4$ folds and train $F$ outcome regression models, $\widehat{\widehat{Q}}_f$, $f = 1, 2, 3, 4$, each to be used to make predictions on data in the corresponding fold. The model $\widehat{\widehat{Q}}_f$ is trained using observations in all folds except for folds $f$ and $\min(f + 1, F)$. As long as the data is sufficiently mixing, dropping fold $f + 1$ ensures sufficient independence from future data. At the same time, each model now uses an amount of data that grows linearly in $T$. Further, unlike sequential sample splitting, which requires training of $T - 1$ models, cross-time-fitting requires training only $F$ models. Figures 2 and 3 establish parity in the conclusions w.r.t. CADR's coverage compared to all other baseline estimators on 57 OpenML-CC18 datasets, 4 target policies and linear outcome regression models for all estimators that use them when these models are trained with sequential sample splitting (as in Figure 1 of the Section 4.2 in the main text) and with time cross-fitting respectively.

### E.3  CADR in Misspecified vs. Well-Specified Outcome Regression Models

Although CADR's advantage over DR is more pronounced when the off-policy estimator's outcome regression model is misspecified (*e.g.*, using linear model on real data), this section establishes the advantage of CADR over all other estimators when they all use a well-specified outcome regression model (*e.g.*, tree). Figure 4 shows CADR's coverage performance when the outcome regression model of DM, DR, MRDR and CADR is misspecified (linear regression model trained with the default `sklearn` parameters) and Fig. 5 shows CADR's coverage performance when the outcome regression model of DM, DR, MRDR and CADR is well-specified (decision tree regression model trained with the default `sklearn` parameters). Each dot represents each one of the 72 datasets and is colored blue when CADR has significantly better coverage than the corresponding baseline column estimator, in red when it has significantly worse coverage and in black when the two coverage are within standard error. Results are averaged over 64 simulations per dataset and standard errors are shown. CADR remains the best estimator in both cases but as expected, in the misspecified outcome regression model case there are more datasets where CADR has significantly better coverage than DR compared to the well-specified outcome regression model case where there are more datasets for which CADR's and DR's coverage are within standard error. This is because when the error is large

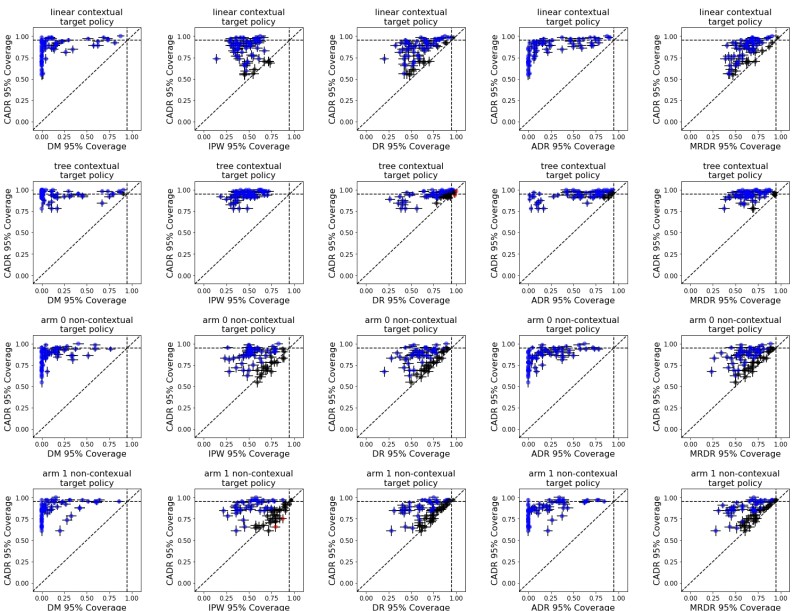

Figure 4: Comparison of CADR estimator against DM, IPW, DR, ADR, MRDR w.r.t. 95% confidence interval coverage on all 72 OpenML-CC18 datasets and 4 target policies with **linear outcome regression model (misspecified)** trained with cross-fitting of all estimators that use them.

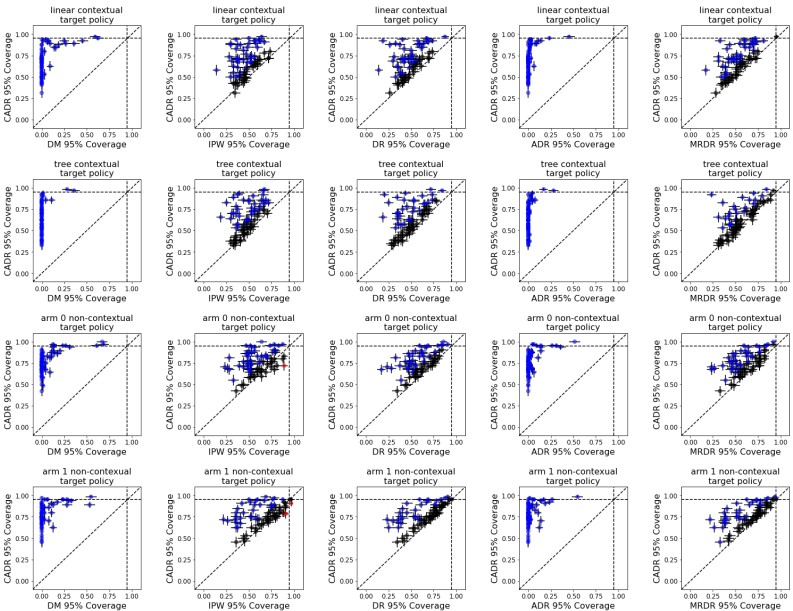

Figure 5: Comparison of CADR estimator against DM, IPW, DR, ADR, MRDR w.r.t. 95% confidence interval coverage on all 72 OpenML-CC18 datasets and 4 target policies with **tree outcome regression model (well-specified)** trained with cross-fitting of all estimators that use them.

and is multiplied by a potentially large inverse propensity score of the logging policy, the variance stabilization performed by CADR is the most effective.

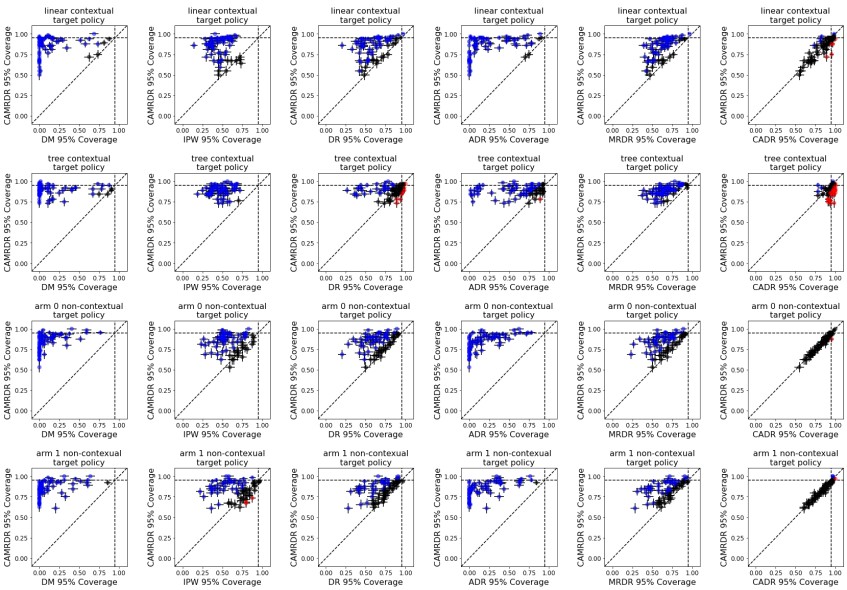

Figure 6: Comparison of CAMRDR estimator against DM, IPW, DR, ADR, MRDR and CADR (last column) w.r.t. 95% confidence interval coverage on all 72 OpenML-CC18 datasets and 4 target policies with **linear outcome regression model (misspecified)** trained with cross-fitting.

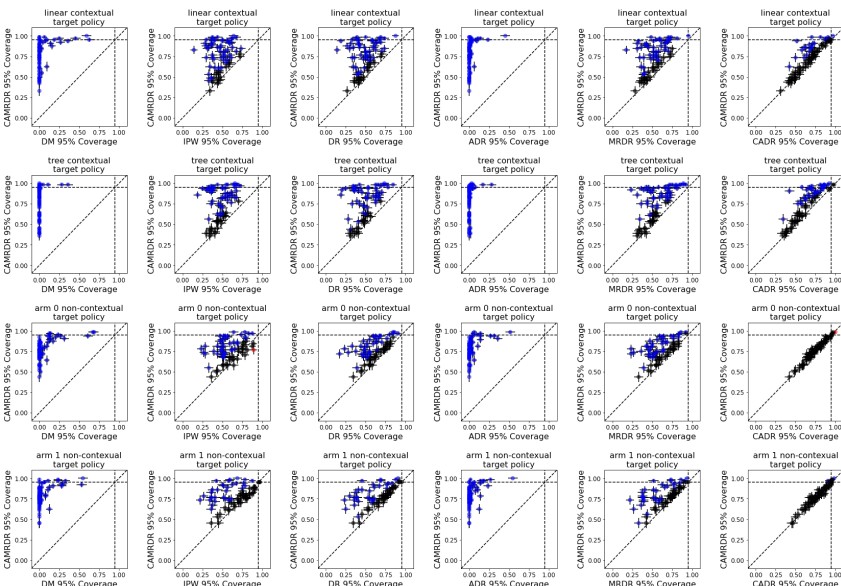

Figure 7: Comparison of CAMRDR estimator against DM, IPW, DR, ADR, MRDR and CADR (last column) w.r.t. 95% confidence interval coverage on all 72 OpenML-CC18 datasets and 4 target policies with **tree outcome regression model (well-specified)** trained with cross-fitting.

### E.4 Importance Sampling Weighted Training of CADR Outcome Regression Model

Finally, we consider the effect of using weighted training in the outcome model fitting of CADR akin to MRDR's outcome model fitting, where each training sample $O(s) = (X(s), A(s), Y(s))$ is weighted by $w(s) = \frac{g^*(A(s)|X(s))}{g_s(A(s)|X(s))}$. We call this estimator CAMRDR. Figure 6 shows CAMRDR's coverage performance against baselines and CADR when the outcome regression model of DM, DR, MRDR, CADR and CAMRDR is misspecified (linear regression model trained with the default `sklearn` parameters). Figure 7 shows CAMRDR's coverage performance against baselines and

CADR when the outcome regression model of DM, DR, MRDR, CADR and CAMRDR is well-specified (decision tree regression model trained with the default `sklearn` parameters). Again, each dot represents each one of the 72 datasets and is colored blue when CAMRDR has significantly better coverage than the corresponding column estimator, in red when it has significantly worse coverage and in black when the two coverage are within standard error. Results are averaged over 64 simulations per dataset and standard errors are shown. Importance sampling weighted training makes a small positive difference compared to CADR in the well-specified case and a small negative difference compared to CADR in the mis-specified case. CAMRDR is better than all other baselines in both cases.

## E.5 Performance in Small vs. Large Datasets

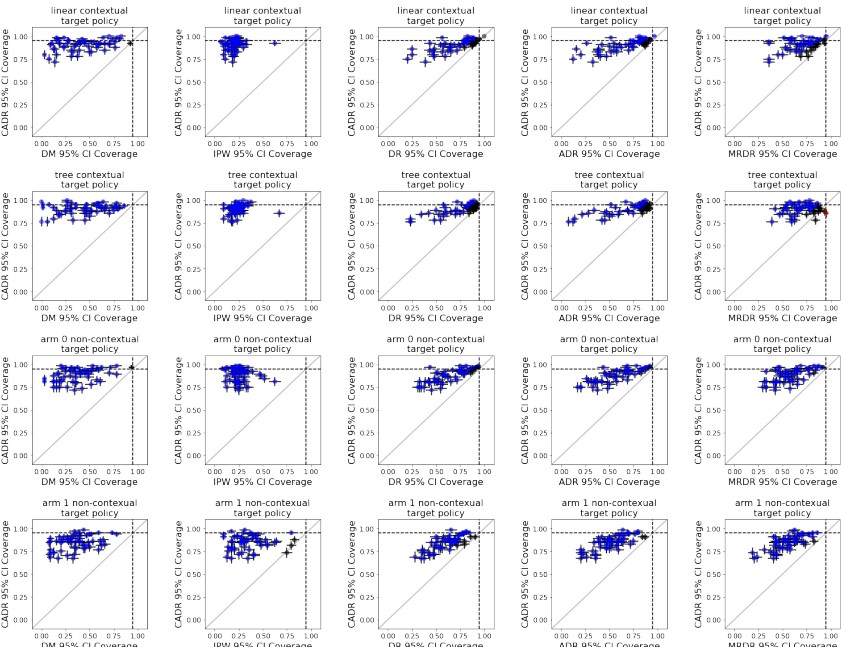

Figure 8: Comparison of CADR estimator against DM, IPW, DR, ADR, MRDR w.r.t. 95% confidence interval coverage on 57 OpenML-CC18 datasets and 4 target policies with cross-fitting, a linear regression model for all estimators that use an outcome model and $T = 1000$ observations **(small data)**.

In the main empirical evaluation in section 4 and in the additional evaluations in sections E.2, E.3 and E.4 of the appendix, we used an off-policy evaluation dataset with $T = 10000$ observations. The purpose of using large samples is that the normality property holds in the asymptotic regime. That said, in more moderately sized samples, one should still expect an asymptotically normal estimator to perform better than an estimator which is not asymptotically normal. So, in this subsection, we re-ran the same experimental setup as in subsection E.2, Figure 3 but with $T = 1000$. As seen in Figure 8, all estimators' coverage deteriorates with smaller samples compared to the performance of estimators in the same setup but with $T = 10000$ observations shown in Figure 3. Despite, the expected deterioration of all estimators' performance under the "small data" regime, CADR still remains the clearly best-performing estimator demonstrating its robustness in more moderately-sized training samples.

## E.6 Execution Specifics of Experiment Code

The IPython notebook to reproduce the experimental results of the main paper and the appendix can be found at `https://github.com/mdimakopoulou/post-contextual-bandit-inference`. One needs to obtain an OpenML API key to run this code (instructions can be found at https://docs.openml.org/Python-guide/) and replace the string `'YOURKEY'` in

`summarize_openmlcc18()` and in `download_openmlcc18()` functions with it. After that, if the notebook is executed as is, it reproduces Figure 3 (1h 26min on a 64 CPU Intel Xeon). Changing variable `ope_outcome_model_training` from `cross_fitting` to `sequential_sample_splitting` reproduces Figures 1/2 (same) (22h 23min on a 64 CPU Intel Xeon). Changing variable `task_min_samples` from 1000 to 0 and variable `task_max_contexts` to `np.inf` reproduces Figure 4 (20h 20min on a 64 CPU Intel Xeon). Changing variable `ope_outcome_model` from `LinearRegression()` to `DecisionTreeRegressor()`, variable `task_min_samples` from 1000 to 0 and variable `task_max_contexts` to `np.inf` reproduces Figure 5 (26h 8min on a 64 CPU Intel Xeon). Figures 6 and 7 are from the same execution as Figures 4 and 5 but with adding 'CAMRDR' in the `competitors` variable of the `visualize_coverage()` function. Figure 8 is the same as Figure 3 but with changing variable `batch_size` from 100 to 10.

## Supplementary Material References

Aurélien Bibaut, Antoine Chambaz, Maria Dimakopoulou, Nathan Kallus, and Mark van der Laan. Risk minimization from adaptively collected data: Guarantees for supervised and policy learning. *arXiv preprint arXiv:2106.01723*, 2021.

Nathan Kallus and Masatoshi Uehara. Efficiently breaking the curse of horizon in off-policy evaluation with double reinforcement learning. *arXiv preprint arXiv:1909.05850*, 2019.

R. van Handel. On the minimal penalty for Markov order estimation. *Probability Theory and Related Fields*, 150:709–738, 2011.