# OpenReview forum: "Post-Contextual-Bandit Inference"
_NeurIPS.cc/2021/Conference — NeurIPS 2021 Poster_

### Official Review · Reviewer_2WyM · 2021-07-09

**Rating:** 3
**Confidence:** 5

**Summary:**

This study proposes a variance-stabilized estimator for policy evaluation from data obtained from contextual bandit algorithms, which is inspired by [Luedtke and van der Laan 2016]  and [Hadad et al. 2019].  In this study, the authors estimate the variance of the score function (canonical gradient) from samples, which standardizes the MDS for applying martingale CLT.
[Luedtke, and van der Laan 2016] Statistical inference for the mean outcome under a possibly non-unique optimal treatment strategy
[Hadad et al 2019] Confidence Intervals for Policy Evaluation in Adaptive Experiments.

**Limitations And Societal Impact:**

This paper does not describe a specific bandit algorithm, where the proposed method can be applied.
As pointed in Main review, there are three possible settings for the bandit problem settings:
- regret minimization: Cannot be applied to regret minimization where the variance of the IPW part jumps infinitely (Assumption 6?).
- best arm identification: There is no generally accepted research on contextual best arm identification.
- adversarial bandit: This is not the data generating process considered in this paper.
I do not believe the proposed method can be used for any of these.

In addition, for example, in a problem where the logging policy converges and does not take $0$ value, the IPW (IS) estimator is sufficient without the proposed method.
The authors state that IPW (IS) estimators "are (not) generally asymptotically normal under adaptive data collection" (line 253), but this is not true. (line 253) because if the logging policy converges, the IPW estimator will asymptotically follow normality by the martingale CLT.
The authors also pointed out the problem that the logging policy take $0$.
Indeed, under this situation, the IPW (IS) estimator cannot be used, but the proposed method also cannot be used (Assumption 6).
After all, the authors have made (implicit) assumptions on the sampling strategy, and they should discuss whether those assumptions are more valid than other assumptions.

Moreover, in the setting of the experiment, it appears that the authors consider a problem where the logging policy converges.
So, I think that the authors do not have to use their proposed method to obtain asymptotic normal estimator.
If my comment is incorrect, the authors should clarify this in the paper.

**Main Review:**

I vote for clear reject.
I appreciate that this problem tackles a mathematically challenging problem.
Also, there is not enough of a survey of the bandit problem, and it is not clear under what circumstances the proposed method would be needed.
Although the authors do not mention it much, several methods of statistical inference from contextual moon bandits have already been proposed.
Compared with them, I consider that the authors' method aims to enable statistical inference in more difficult situations.
However, for me, it seems that it is unnecessary for the bandit problem, that is, there is a few concrete examples of a bandit algorithm in which the proposed method is needed.
Furthermore, Assumptions 5 and 6 restrict the logging policy (sampling strategy of bandit algorithm), and it is not clear what specific bandit algorithms satisfy these assumptions.
In addition, the essential contribution of this paper seems to be the proof of Theorem 2 (other parts have already been known).
However, the notation is complicated, and there are some omissions and typos, so I could not check it sufficiently.
The inability to read the proofs is partly my problem and partly due to the time constraints of the conference review process, but I also think that the authors could make an effort to make the proof more readable.
In order to increase the score, the authors need to
- explain the connection with existing studies of the "bandit problem" or change the field that the authors want to contribute.
- make proofs and notations easier to read.
Detailed comments are listed below.

# Major concerns
I have three major concerns for this paper:
- Point 1: lack of connection to bandit problem and the usefulness of the proposed methods.
- Point 2: theoretical correctness and implicit restriction to the bandit algorithms;
- Point 3: the setting of the experiment may not be appropriate.
Among them, I am particularly concerned with Point 1, that is, the proposed method is not designed for the bandit problem.

## Point 1
### Existing studies
The paper does not cite much of the bandit literature, and it is not clear how the proposed method can be applied.
In addition, there are also several studies on evaluation from data gathered from adaptive experiments (including bandit algorithms) although not cited in this paper.
For example,
[van der Laan 2008] The Construction and Analysis of Adaptive Group Sequential Designs
[Hahn et al 2011] Adaptive Experimental Design Using the Propensity Score
I think that the following papers also propose such methods.
[van der Laan and Lendle] Online Targeted Learning
[Chambaz et al. 2014] Targeted Covariate-Adjusted Response-Adaptive LASSO-Based Randomized Controlled Trials
[Tabord-Meehan 2020] Stratification Trees for Adaptive Randomization in Randomized Controlled Trials
I can list other more studies tackling this problem.

Some of these studies assumes that the contextual policy $g_t(a | x)$ converges to a time-independent measure and $g_t(a| x)\neq 0$.
This is a natural assumption for some bandit problems (but not for regret minimization).
Under this assumption, we can obtain asymptotically normal estimator with a simpler form, such as the IPW (IPS) estimator without the variance-stabilization.

In situations where the data is in batch form, statistical inference also can be performed even if the policy does not converge.
[Zhang et al 2020] Inference for Batched Bandits
This is essentially a generalized method of moments (GMM).
(Although it is not well known, a GMM estimator is known to have asymptotic normality for martingales)

Compared to these uncited previous studies, I cannot understand how useful the proposed method is to the bandit community.
In addition, as described below, I cannot think of a specific bandit case where this proposed method would be necessary.

### Variance-stabilized estimators and fewer applications in bandit problem
I consider that the contribution of this study lies in avoiding sample splitting, which (I think) is used in [Luedtke, and van der Laan 2016] and [Kato 2020], to obtain $\sqrt{N}$-consistent estimator.
[Kato 2020] Confidence Interval for Off-Policy Evaluation from Dependent Samples via Bandit Algorithm: Approach from Standardized Martingales
By using sample splitting, we can show asymptotic normality without making assumptions about the contextual policy.
Instead, with that sample split, it is no longer $\sqrt{N}$-consistent.
Recently, [Hadad et al 2019] proposed a method for statistical inference from a bandit data without context, which attains  $\sqrt{N}$-consistency.
It can avoid estimation of the variance by considering bandit problem without context.
In this study, the authors do not use sample-splitting and obtain a  $\sqrt{N}$-consistent estimator.

However, I believe that these methods are essentially unnecessary for the bandit problem, and a simpler estimator is sufficient.
The reasons for this are listed below.
- If the logging policy converges, asymptotic normality can be obtained using other simpler estimators, such as IPW (IPS) estimators without using the methods of [Hadad et al 2019] or [Luedtke, and van der Laan 2016].
- These papers presume stochastic bandit problem, but it is difficult to find a bandit algorithm, where the logging policy does not converge.
- None of [Hadad et al 2019] or [Luedtke, and van der Laan 2016] can be used for the regret minimization, where the logging policy converges to zero. (In this case, the cannonical gradient diverges).
- In bandit problems, the target policy (denoted by $g^*$ in this paper) to be evaluated is also time-dependent, and there is no problem setting in which a time-independent target policy exists in advance and it is evaluated.
I also think that [Luedtke, and van der Laan 2016] and [Hadad et al 2019] consider a different situation from bandit problem.

### Lack of connection to bandit problem
I explain this in more details.  As well as [Luedtke, and van der Laan 2016] and [Hadad et al 2019],  the proposed method is needed if a bandit problem satisfies the following situations:
1. Bandit model is stochastic bandit (so not adversarial bandit);
2. The goals is not regret minimization (logging policy does not converges to $0$ or take $0$ value);
3. The policy (sapling strategy) does not converge;

If not 1., we cannot assume the data generating process and define the causal effect.
If the bandit algorithm aims to minimize the regret, the canonical gradient will diverge.
(Variance also becomes $\infty$. Exclusion of regret minimization is Asssumtpion 6?)
The authors raised a problem that "IS ratios can both diverge to infinity or converges to zero" in line 114.
If the IS ratios really diverges, we cannot define the variance.
To avoid this problem, we need to carefully restrict the class of the logging policy.
For instance, [Hadad et al 2019] also restricted the class in their Assumptions 1--3.
In this paper, it seems that Assumption 5 corresponds to the  Assumptions 1--3 of [Hadad et al 2019].
Can the authors show the method of Li et al 2010, Kallus and Udell 2020, Qiang and Bayati 2016 etc, which the authors cited in Introduction, satisfy these conditions?
In contrast, if the sampling strategy converges, we do not have to use variance stabilization.
For instance, the IPW (IPS) estimator $y/ g(a| x)$ also have asymptotic normality owing to the martingale property.
(In this paper $\hat{\Psi}^{IPS}$ without variance stabilization. Without variance stabilization, this type of estimator is often used.)

In addition,
 4. The authors also seem to restrict the sampling strategy (logging policy) in Assumption 5.
This is a specific assumption of this paper to show Theorem 2.
What kind of bandit algorithms satisfy this assumption?
In bandit literature, this kind of assumption is carefully discussed in the paper.
For instance, the following paper considers a nonparametric method for regret minimization with carefully restricting the class of sampling strategy.
[Yang and Zhu 2002] Randomized Allocation with nonparametric estimation for a multi-armed bandit problem with covariates
Thus, I think that Assumption 5 (and 6) is not trivial, and the authors should discuss more carefully.

In conclusion, I do not know whether there is a bandit problem where the proposed method is needed.
This is not to deny the author's attempt, but if the author thinks the proposed method is necessary for bandit problem, I would like to know the specific bandit paper.
The paper cites some papers in the field of adaptive experimental design and causal inference, but not many of papers in bandit problem, which is main topic of this paper.
There are several cited papers, but I think that all of them converge on the sampling strategy or do not satisfy Assumption 5--6,.
That is, the authors should motivate the proposed method more from the viewpoint of bandit algorithms.
Otherwise, the authors should change the field that they want to contribute.
I believe that the current structure of the paper, if accepted, will confuse the bandit community.

## Point 2:
### Assumption 5 and 6
As pointed in Point 1, Assumption 5 and 6 restrict the logging policy.
These assumptions should be more paid attention because it is closely related to design of the bandit algorithm. For example, [Yang and Zhu 2002] more carefully discuss the assumption.
In addition, the authors should cite an example of bandit algorithm satisfying these assumptions.
(But the logging policy does not converge. As mentioned above, if it converges, we do not have to use the proposed method)

### Proof of Theorem 2
I think that Theorem 2 is the most important in this paper.
Because if we use sample splitting, we can show the asymptotic normality more easily by losing the $\sqrt{T}$-consistency for a sample size $N$ ([Luedtke, and van der Laan 2016]).
However, owing to the difficulty of the notations, we cannot fully understand the proof.
Could the authors answer the following questions?
- Does line (6) of $\ell 571$ in supplementary material is MDS?
The authors may omit $g_t$, but I think $g_t$ is correlated with past samples. Does it satisfy MDS?
- Do we need to assume boundedness or sub-gaussian to the outcome $Y$?
- If we do sample splitting, can't we prove asymptotic normality without assuming Assumptions 5 and 6, instead of losing $\sqrt{N}$-consistency? [Luedtke, and van der Laan 2016] and [Kato 2020] (I think this paper is essentially same as [Luedtke, and van der Laan 2016]) do this?
If sample splitting can solve the problem, the authors should explain this fact even if they are inferior to the proposed method in the sense of $\sqrt{N}$-consistency?

## Part 3
In experiments, the logging policy seems to converge.
If so, we do not have to use the proposed variance stabilization and obtain asymptotic normality with simpler estimators proposed in existing studies.
In addition, If so, the authors should change the setting of experiments.

# Minor points
- As pointed in Part 1, there are many methods for causal inference from contextual experimental designs in situations where the logging policy converges, but they are almost never cited.
- I am a bit familiar with this notation, so I could read it, but this notation would be very difficult for machine learning people to read. More explanation of terminology is needed, since prior knowledge differs between the bandit and causal inference communities.
- The notation in this paper is difficult and the proof is very hard to read. I don't mind that the proof is difficult, but there are many notations that are not explicitly defined, making it difficult to read. In addition, there are typos in some places.
- Assumption 5 constrains the behavior of the original Bandit algorithm. However, if we are going to put constraints on the algorithm, why not design the algorithm so that it is easy to evaluate measures?

# Requirements to the author
I would require the authors to answer the following point
- Raise an example of a situation of bandit problems, where this algorithm is needed, with justifying Assumption 5 and 6;
and improve following points in the next revision:
- Reorganization of Notation;
- Another experiment using  logging policy that needs the proposed method;
- A better literature review.

**Time Spent Reviewing:**

More than 20 hours

---

> ### Author Response · Authors · 2021-08-10
> **Authors' Response to Reviewer 2WyM: Main Response**
>
> Thank you for taking the time to review our paper. We hope to clarify in our response that the proposed method is relevant to the users of contextual bandit algorithms because it enables valid inference based on data collected by contextual bandits which minimize regret and as a result converge to placing zero probability to actions that are suboptimal. Such algorithms see increasing use in practice in place of A/B tests, but inference is still often a requirement at the end. The paper proposes the first estimator for policy value that is asymptotically normal under such a contextual adaptive data collection and an extensive empirical evaluation on a variety of domains corroborates the theory and the better performance of our estimator compared to several other well-known estimators in the literature.
>
> First, we want to clarify that our paper focuses on offline inference, *after* a contextual bandit algorithm has concluded; we assume access to historic data, but no ability to gather new data. Therefore, the focus of our method is not regret minimization in an online setting but valid inference in the offline setting. And by valid inference, we mean, how can we use the logged data of a contextual bandit that has concluded in order to form confidence intervals with calibrated coverage on the value of a new target policy (be it constant or personalized policy).
>
> Second, our paper (and the entire research area of off-policy evaluation, which doesn’t target regret minimization) is directly relevant to the bandit problem and to the bandit community as it directly tackles the following key challenges:
> 1. We want to use logged data from a contextual bandit algorithm deployed online to run inferences offline after the fact about any one policy.
> 2. As they minimize regret, contextual bandit algorithms gradually stop exploring suboptimal arms for each context (placing probability zero on them, given a context). If a non-zero exploration probability was required on all arms, these bandits would suffer linear regret.
> 3. Under scenario (1) and (2) which are both very common in practice, there isn’t an estimator that obtains an unbiased and asymptotically normal estimate of the value of a target policy using logged data for contextual bandit policies for which the exploration of suboptimal arms is allowed to go to zero. Existing estimators (several of which you mention in your comments) either:
>     - consider independent and identically distributed data, i.e., *non-adaptively collected data* (e.g., Horvitz and Thompson, 1952; Dudik et al 2011; Wang et al., 2017; Luedtke and van der Laan, 2016; Su et al., 2019, 2020),
>     -  consider adaptively collected data by a *non-contextual* multi-armed bandit (Hadad et al., 2019),
>     - consider adaptively collected data from a contextual bandit but restrict the logging policy to *always have a constant positive lower bound on the probability of exploring any one arm* (e.g. van der Laan, 2008; Zhang 2021) and therefore to have linear regret.
>
> Third, we thank you for pointing out these citations. We will add the above information to clearly explain why they do not solve the problem we tackle (indeed, they are intended for other problems, and naively applying them in our setting yields bad results). We again emphasize that these prior works do not undermine the novelty and significance of the contribution our paper makes, as they haven’t proposed an unbiased and asymptotically normal estimator for data collected by a contextual bandit that allows the probability of suboptimal actions for each context to go to zero. Our estimator is the first in the literature that does this.
>
> Fourth, with respect to a realistic and concrete example of when the method we propose in our paper is necessary, here is one: To compare multiple versions of a UI change, instead of running a standard randomized trial, one has convinced the product manager to run a contextual bandit algorithm, but such both generates data that is no longer independent and the probability of choosing some arms in some contexts may decay to zero (as is necessary for sublinear regret or for maximizing the probability of choosing the best arm). At the end of the test, one still needs to analyze the data to make managerial decisions and so one needs a valid, calibrated confidence interval on any of: average treatment effect, a subgroup effect, or a new personalized policy. When Assumption 5 holds with p<2, we can even draw inferences on the value of the optimal personalized policy learned by the contextual bandit, to judge, for example, how much benefit it has relative to a constant policy. Then, the method we propose is directly relevant to them and the only method in literature that can provide unbiased and asymptotically normal estimates that support valid inference, and hence drawing credible conclusions from the data.
>
> This example we mention above is the one studied in our experimental section as well, under more than 57 different real datasets, 4 different target policies, 6 estimators and 6 estimator fitting procedures. We see that simpler estimators such as IPW, DR fail to provide correct coverage due to their lack of asymptotic normality on data collected by a contextual bandit logging policy that doesn’t impose the non-zero exploration requirements of all arms. To achieve the right coverage not only do we need stabilization weights, but the stabilization weights must take into account the local variance of each datapoint, which is dependent on both arm and context. The latter point is an important point of departure from previous work on the non-contextual problem (Hadad et al., 2019), and we can see that naively applying this work in the contextual setting (for which it was not intended) gives bad results empirically. We have included the code alongside the submission in the supplemental material for anyone who is interested to reproduce or extend our experimental setup and results.
>
> Next, we follow-up in a separate thread below to provide some additional point-by-point responses to your comments.

---

> > ### Author Response · Authors · 2021-08-10
> > **Authors' Response to Reviewer 2WyM: Detailed Point-By-Point Answers to Comments & Questions**
> >
> > In this thread, we provide some additional responses to your points.
> >
> > - Hadad et al. 2019 & our paper’s connection with the bandit problem:
> >
> >   It is mentioned in the review that Hadad et al. 2019 & our paper’s cannot be used for the bandit problem because they do not do regret minimization. This is expected, because the goal of these methods is off-policy evaluation, meaning to estimate and get confidence intervals for the value of a counterfactual policy, not online regret minimization. Our estimator uniquely enables one to use data collected by contextual bandit algorithms that do regret minimization (i.e., which are allowed to reduce exploration with time) in order to do unbiased and asymptotically normal inference.
> >
> > - Assumptions 5 & 6
> >
> >   We consider the general setting of data collected by a contextual bandit policy where the no propensity score clipping constraint is imposed and the exploration of suboptimal arms for a given context is allowed to decay to zero. Assumptions 5 and 6 are far less restrictive than the requirements of (ordered roughly from more restrictive to less restrictive):
> >     1. iid data,
> >     2. non-adaptive data,
> >     3. adaptive but non-contextual data,
> >     4. adaptive and contextual and of propensity score clipping constraints that do not allow the exploration of any action to converge to 0.
> >
> >   All previous off-policy evaluation works operate under one of these assumptions, which fail for contextual bandit algorithms that minimize regret. We allow unbiased and asymptotically normal inference on data collected by an adaptive and contextual logging policy that belongs to a large policy class (e.g., tree-based logging policies with finite tree depth are covered under our Assumption 5) and further we allow the logging policy to converge to 0 exploration as would be needed to achieve good regret performance, so long as it does it at a reasonable rate (Assumption 6). We will add more explanation along the lines we mention above about how our method allows more flexibility than other works and what kind of policy classes are covered by Assumption 5.
> >
> > - Need of stabilizing weights in logging policies that have converged:
> >
> >   Convergence of the logging policy to some fixed point is *not* a sufficient condition for the IPS estimator to be unbiased and asymptotically normal. For this to hold, the limit logging policy needs to be such that the probability of playing each action in each context is lower-bounded away from zero by a positive constant. This is an assumption shared among past articles on off-policy evaluation from adaptive contextual bandit data that propose asymptotically normal estimators [van der Laan 2008, Chambaz et al. 2017] and in our work we very crucially lift it. The main issue of IS-based estimators without variance stabilizing weights for converged policies where the propensity score is not lower-bounded is the following: if the logging policy converges to an optimal arm $a^*(x)$ per context $x$ and the suboptimal arms stop being explored for that $x$ (i.e. $g_t(a |x) \to 1$ when $t \to \infty$ for $a = a^*(x)$ and $g_t(a |x) \to 0$ when $t \to \infty$ for $a \neq a^*(x)$), then the IS ratio $g^*(a | x) /  g_t(a |x)$ converges to $g^*(a | x)$ for $a = a^*(x)$ whereas for $a \neq a^*(x)$, the IS ratio diverges to infinity as $t \to \infty$. As a result, the CLT condition of bounded moments is not satisfied for simple estimators such as IPS and DR and they are not asymptotically normal. This failure is also clearly seen in our experiments.
> >
> > - Notation:
> >   Although there are some small differences, our notation is relatively close to that of: van der Laan, Mark J., "The Construction and Analysis of Adaptive Group Sequential Designs", 2008, and Antoine Chambaz, Wenjing Zheng, Mark J. van der Laan "Targeted sequential design for targeted learning inference of the optimal treatment rule and its mean reward," The Annals of Statistics, 2017. We worked hard to make this digestible but we appreciate the feedback and we will add more clarifications wherever needed.
> >
> > Finally, we provide some direct answers to some of your questions:
> >
> > - Question: *"I consider that the contribution of this study lies in avoiding sample splitting, which (I think) is used in [Luedtke, and van der Laan 2016] and [Kato 2020], to obtain a sqrt(N)-consistent estimator."*
> >   - Answer: Like Luedtke and vdL 2016, we also use what we call sequential one-step ahead sample splitting, which indeed allows one to only use one single bandit run, as opposed to Kato 2020, which requires several independent bandit runs.
> >
> > - Question: *"If the logging policy converges, asymptotic normality can be obtained using other simpler estimators, such as IPW (IPS) estimators without using the methods of [Hadad et al 2019] or [Luedtke, and van der Laan 2016]."*
> >   - Answer: This is false. Convergence of the logging policy is not enough for IPS to be asymptotically normal. You need in addition that the logging policy converges to a policy such that the probability of playing each action is lower bounded away from zero for each context. That means regret-minimization algorithms are not allowed. In contrast, we permit decaying exploration.
> >
> > - Question: *"None of [Hadad et al 2019] or [Luedtke, and van der Laan 2016] can be used for the regret minimization, where the logging policy converges to zero. (In this case, the canonical gradient diverges)."*
> >     - Answer: This is false. This divergence is precisely the reason for stabilization, as developed by Luedtke and van der Laan 2016 and then used by Hadad et al 2019 for the problem of (non-contextual) adaptively collected data.
> >
> > - Question: *"These papers presume stochastic bandit problem, but it is difficult to find a bandit algorithm, where the logging policy does not converge.*"
> >     - Answer: One example is a (non-contextual) multi-armed bandit environment where two identical arms exist and Thompson sampling is used. But this is not the crucial point: the crucial point is that the exploration is decaying to zero -- that is what makes inference difficult
> >
> > - Question: *"As pointed in Point 1, Assumption 5 and 6 restrict the logging policy. These assumptions should be more paid attention because it is closely related to design of the bandit algorithm. For example, [Yang and Zhu 2002] more carefully discuss the assumption."*
> >     - Answer: Regarding assumption 5:
> >         - Yang and Zhu 2002 do not deal with OPE but rather with online regret minimization. Their method is value-based and assumes consistent value-function (regression) estimators; this is opposed to policy-based, and as such they do not make an assumption on the complexity of a policy class. The comparison is therefore inapt.
> >         - It is commonplace in work on policy-based contextual bandit algorithms to make assumptions on the complexity of the policy class, e.g., Dudik et al 2011 “Efficient Optimal Learning for Contextual Bandits”, Agarwal et al. 2014 “Taming the Monster: A Fast and Simple Algorithm for Contextual Bandits”, Bibaut et al. 2020 “Generalized Policy Elimination: an efficient algorithm for Nonparametric Contextual Bandits”, Foster and ​​Krishnamurthy 2018 “Contextual bandits with surrogate losses: Margin bounds and efficient algorithms”, among others.
> >
> > - Question: *"In addition, the authors should cite an example of bandit algorithm satisfying these assumptions."*
> >     - Answer:
> >         - Epsilon greedy with a finite sectional variation norm exploit policy (e.g., tree policies).
> >         - Epsilon greedy with exploit policy defined as the softmax of parametric scores (e.g., linear).
> >
> > - Question: *"(But the logging policy does not converge. As mentioned above, if it converges, we do not have to use the proposed method)"*
> >     - Answer:
> >         - It doesn’t matter for the proposed method whether the logging policy converges or not, although in most practical situations the logging policy does converge.
> >         - Convergence of the logging policy is not sufficient for previously proposed estimators to be asymptotically linear, as explained earlier. One would also need the limit to have a positive exploration rate - which is the setting studied in van der Laan 2008 and Chambaz et al. 2017.

---

> > > ### Comment · Reviewer_2WyM · 2021-08-10
> > > **Re: Authors' Response to Reviewer 2WyM: Main Response**
> > >
> > > I greatly appreciate the authors' reply to my long review.
> > >
> > > Although the authors replied to my question, I think that the most core part of my question has been misunderstood, unfortunately (partly due to my poor expression).
> > > However, from reading the replies, I felt again that the authors might not consider the background of the regret minimization problem.
> > > Perhaps the authors' method and the method of Hadad et al. (2019)'s method cannot be used for policy evaluation from samples obtained via regret minimization algorithms.
> > >
> > > I haven't finished reading all of the replies, but I would like to point out a possible theoretical error (I think).
> > > In addition, the review is very long, so I want to solve the problems one by one.
> > >
> > > In this reply, I would like to focus on the next question.
> > > **Can the authors' method (and Hadad et al. (2019)) apply to samples obtained from regret minimization algorithms?**
> > >
> > > My comment "the authors' method cannot be used for regret minimization" actually means
> > >  "**the authors' method cannot be used for policy evaluation from samples obtained via regret minimization algorithm**."
> > > I fully understand that the authors' motivation is not in the regret minimization but policy evaluation (I also have read the paper of Hadad et al and Ludtke and van der Laan).
> > > Therefore, although the authors responded as
> > > > First, we want to clarify that our paper focuses on offline inference
> > > , after a contextual bandit algorithm has concluded; we assume access to historic data, but no ability to gather new data. Therefore, the focus of our method is not regret minimization in an online setting but valid inference in the offline setting. And by valid inference, we mean, how can we use the logged data of a contextual bandit that has concluded in order to form confidence intervals with calibrated coverage on the value of a new target policy (be it constant or personalized policy).
> > >
> > > I already know this point.  On the other hand, the authors have replied as
> > > > Question: "If the logging policy converges, asymptotic normality can be obtained using other simpler estimators, such as IPW (IPS) estimators without using the methods of [Hadad et al 2019] or [Luedtke, and van der Laan 2016]."
> > > Answer: This is false. Convergence of the logging policy is not enough for IPS to be asymptotically normal. You need in addition that the logging policy converges to a policy such that the probability of playing each action is lower bounded away from zero for each context. That means regret-minimization algorithms are not allowed. In contrast, we permit decaying exploration.
> > >
> > > but I think this is based on the authors' inaccurate understanding on the bandit problem.
> > > This is because the authors' assumption is insufficient for the usual exploration rate of regret minimization methods,
> > > that is, for policy evaluation, the authors must assume that samples are gathered by sub-optimal bandit algorithms.
> > >
> > > This opinion is based on the following reasons:
> > > - In regret minimization, algorithms designed to achieve the regret bound with the order of
> > > $$ log T.$$
> > > - To achieve this regret bound, the exploration rate must converge to $0$ with a  faster rate than
> > > $$ 1/t .$$
> > > - This is because the integral of $1/t$ is $log T$.
> > > - Hadad et al. (2019) and the authors' work seem to assume a slower exploration rate than $1/t$, such as $1/\sqrt{t}$. In this case, the regret is $\sqrt{T}$, which is not usually considered in regret minimization.
> > > - When we can use discrete contextual information, we can obviously achieve $\log T$ regret bound.
> > > - Even when we can use continuous contextual information, we can achieve $\log T$ regret bound if we can use margin condition.
> > > - **For example, In Assumption 3 and 6 of the authors' manuscript, the authors require an exploration rate slower than $1/\sqrt{t}$, which is insufficient for policy evaluation from samples obtained via (optimal) regret minimization algorithm.**
> > > - This is quite strange because the authors presume a sub-optimal algorithm for regret minimization.
> > > - The authors mentioned "sub-linear" regret, but "sub-linear" is very pessimistic result, which mainly appear in adversarial (non-stationary) bandit. But if the process is non-stationary, we cannot conduct policy evaluation (violates the problem setting of this study).
> > > - Under some cases, the upper bound of the algorithm $\sqrt{T} \log T$. Even for this case, Assumption 5 and 6 does not hold.  For instance, exploration rate $1/t^{1/3}$ is insufficient for such a logging policy, but $\alpha(\beta, p)$ seems strictly slower than $1/3$. This is because a logging policy with exploration rate strictly slower than $1/t^{1/3}$ causes a regret strictly large than the order of $T^{2/3}$ (integral of $1/t^{1/3}$), which is obviously larger than $\sqrt{T} \log T$ regret. So, I could not  imagine a situation where samples are gathered from such a very bad algorithm (logging policy).
> > >
> > > Therefore,
> > > - I do not think we can use the method of Hadad et al. 2019 since most regret minimization methods are of the order of $\log T$ when we do not use the contextual information.  The order $\log T$ means exploration rate $1/t$, but I think that this exploration rate violates the assumptions of Hadad et al. 2021.
> > > - **The authors' method also can be applied to very limited situations where we use a "very sub-optimal" algorithm for regret minimization.**
> > >
> > > Can you please answer if my opinion is correct? That is,
> > > 1. Can Hadad et al. (2019) apply to samples gathered by (optimal) regret minimization algorithm?
> > > 2. Can the authors' work apply to samples gathered by (optimal) regret minimization algorithm?
> > > 3. Can the authors raise a well-known example of contextual bandit algorithm, which can be applied to the authors' method. And can the authors show that the algorithm theoretically satisfy Assumption 5 and 6?
> > > 4. Can $\alpha(\beta, p)$ in Assumption 6 be less than $1$, which is need for policy evaluation from sample gathered by (optimal) regret minimization algorithm?
> > > 5. Can Assumption 5 and 6 hold for policy evaluation from sample gathered by regret minimization algorithm with $\sqrt{T}\log T$ regret bound? This regret bound can be improved (not so tight). However, even for this case, Assumption 5 and 6 do not hold because $\alpha(\beta, p) > 1/3$, that is, the policy evaluation can be conducted only for logging policy with a regret larger than $T^{2/3} \gg \sqrt{T}\log T$.
> > >
> > > I will raise my score when this theoretical question is resolved.
> > > However, if this question is not resolved, I have to conclude that the theory is not adequate.
> > > Indeed, this method may enable to evaluate the policy value when the logging policy converges to $0$.
> > > However, this is very limited situation and not standard in regret minimization.
> > > (Later, I will refer the authors to some papers on contextual bandits algorithm for regret minimization and send other questions and replies.)
> > >
> > > Note that when I asked
> > > > Raise an example of a situation of bandit problems, where this algorithm is needed, with justifying Assumption 5 and 6; and improve following points in the next revision:
> > >
> > > I meant that I would like to know that for which existing well-known algorithms for regret minimization the proposed method can be applied **theoretically**, not that I would like to know about applications such as the following reply:
> > >
> > > > Fourth, with respect to a realistic and concrete example of when the method we propose in our paper is necessary, here is one: To compare multiple versions of a UI change, instead of running a standard randomized trial, one has convinced the product manager to run a contextual bandit algorithm, but such both generates data that is no longer independent and the probability of choosing some arms in some contexts may decay to zero (as is necessary for sublinear regret or for maximizing the probability of choosing the best arm). At the end of the test, one still needs to analyze the data to make managerial decisions and so one needs a valid, calibrated confidence interval on any of: average treatment effect, a subgroup effect, or a new personalized policy. When Assumption 5 holds with p<2, we can even draw inferences on the value of the optimal personalized policy learned by the contextual bandit, to judge, for example, how much benefit it has relative to a constant policy. Then, the method we propose is directly relevant to them and the only method in literature that can provide unbiased and asymptotically normal estimates that support valid inference, and hence drawing credible conclusions from the data.
> > >
> > > That is, I want to ask that for which well-known contextual bandit algorithm Assumption 5 and 6 hold.
> > > However, even for this example, $p < 2$ is insufficient (or a condition on $\beta$, i.e., outcome estimator, is needed), and the logging policy (bandit algorithm) cannot achieve $log T$ regret bound.
> > > **If $p=2$, then $\alpha(\beta, p) = \min(1/5, \beta)$. Ignoring $\beta$, the exploration rate is required to be slower than $1/5$. Then, the regret will be the order of $T^{4/5}$ (integral of $1/t^{1/5}$). This implies that the logging policy has very bad regret for regret minimization**.
> > > Therefore, in this example, the authors presume that the samples are gathered by using a very sub-optimal regret minimization algorithm.
> > > So, this example is not appropriate or the authors need to restrict the situation more.

---

> > > > ### Author Response · Authors · 2021-08-11
> > > > **Re-response to Reviewer 2WyM**
> > > >
> > > > Thank you for reading our response and following up.
> > > >
> > > > Let us clear up a few points:
> > > > 1. **For the stochastic (not adversarial) contextual bandit problem, there are many regret-rate-optimal algorithms with uniform exploration rates of $1/\sqrt{t}$ or slower**. For example, all of the following algorithms have a hard-coded $1/\sqrt{t}$ per-step uniform exploration: the PolicyElimination and RandomizedUCB algorithms of Dudik et al. 2011 “Efficient Optimal Learning for Contextual Bandits”, the ILOVETOCONBANDITS algorithm of Agarwal et al. 2014 “Taming the Monster: A Fast and Simple Algorithm for Contextual Bandits”, the FALCON algorithm of Simchi-Levi and Xu “Bypassing the Monster: A Faster and Simpler Optimal Algorithm for Contextual Bandits under Realizability”, and the AdaCB algorithm of Foster et al. “Instance-Dependent Complexity of Contextual Bandits and Reinforcement Learning: A Disagreement-Based Perspective” (see the $\mu_t$ parameter in the first three and $\gamma_m$ in the last two). Moreover, the algorithm of Bibaut et al. 2020 “Generalized Policy Elimination: an efficient algorithm for Nonparametric Contextual Bandits” has a hard-coded $1/t^{(\frac{1}{2}\wedge\frac{1}{2p})}$ uniform exploration when dealing with a policy class with bracketing entropy exponent $p$. And, the SmoothFTL and HINGE-LMC algorithms of Foster and Krishnamurthy 2018 "Contextual bandits with surrogate losses: Margin bounds and efficient algorithms" both have a hard-coded $t^{-1/(p+1)}$ uniform exploration when dealing with an action-score class with sup-norm-covering entropy exponent $p$. All of these eight algorithms are **optimal** in their respective settings and all of them have a nontrivial per-step propensity bounded away from zero by at least $1/\sqrt{t}$ and sometimes much more.
> > > > 2. **The optimal regret of stochastic contextual bandits is generally much bigger than $\log(T)$**, and the particular rate depends on the particular problem class. For example, the problem in Dudik et al. 2011 has the optimal regret rate $\sqrt{T}$ and the problem in Bibaut et al. 2020 has the optimal regret rate of $T^{1-(\frac{1}{2}\wedge\frac{1}{2p})}$ (up to polylogs). Similarly, the stochastic contextual bandit problem in Rigollet and Zeevi 2010 “Nonparametric Bandits with Covariates” has the optimal regret rate $T^{\frac{d+(1-\alpha)\beta}{2\beta+d}}$ for $\beta\in(0,1]$ and $\alpha\in[0,1]$ and the problem in Hu et al. 2019 “Smooth Contextual Bandits: Bridging the Parametric and Non-differentiable Regret Regimes” has the optimal regret rate $T^{\frac{(d+(1-\alpha)\beta)_+}{2\beta+d}}$ for $\beta\in[1,\infty)$ and $\alpha\geq0$, where in both of these $\beta$ is a smoothness parameter and $\alpha$ is a margin parameter (essentially the distribution of the arm gap over contexts, giving the instance-dependent bound). This is just more examples of the kind of optimal regret rates seen in contextual bandits. Nonetheless, unlike the algorithms mentioned in point 1, the algorithms from Rigollet and Zeevi and Hu et al. are UCB-based algorithms and sometimes act deterministically so they do not have a per-step decaying uniform exploration like the other mentioned algorithms.
> > > > 3. We therefore do agree that for clarification purposes we can offer more discussion on the exploration assumptions as to which algorithms fit our framework and which do not. We will definitely add a couple sentences on this. For example, UCB-based algorithms simply do not fit into our framework as they do not have a per-step decaying uniform exploration, and studying these may require a different analysis. On the other hand, decay-epsilon-greedy, PolicyElimination, RandomizedUCB, ILOVETOCONBANDITS, FALCON, AdaCB, GeneralizedPolicyElimination, SmoothFTL, and HINGE-LMC all do have a decaying per-step uniform exploration.
> > > > 4. It may be that in some certain problems, our exploration assumptions require more exploration than is optimal for that problem (but the suboptimality in regret is the difference between polynomials, not the difference between polynomial and logarithmic). And, we never claimed to tackle rate-optimal contextual bandit algorithms, although we agree we can make this point more explicit. Nonetheless, we still argue that it is a significant theoretical advance to be able to tackle for the first time valid statistical inference from contextual bandit algorithms that obtain sublinear regret, as such has *never* been possible before (this is *also* the first method to work in the contextual adaptive setting when the policy does not converge, which can happen in practice as was the original motivation of Luedkte and van der Laan 2016 in the iid setting). We **strongly** believe that this is a significant breakthrough on a high-profile problem. It may well be that in certain classes of contextual bandit problems, our analysis can and may need to be tightened to permit optimal exploration, and future work looking at specific classes of problems (e.g., any one problem studied by any one of the many papers above) may seek to do this. We will definitely add a couple sentences on possible future improvements and problems where our current assumptions may enable one to get sublinear albeit suboptimal regret rates.
> > > >
> > > > Please let us know if you agree. On our end, while we strongly believe in the significance and novelty of our results, we agree that a few more sentences of discussion can help contextualize it better and we thank you for pointing out these points where we can offer more clarity to the reader. This is very useful. Thanks again for taking the time to read both our paper and our response as well as responding again. We do greatly appreciate it, and hope the above offers some clarity.

---

> > > > > ### Comment · Reviewer_2WyM · 2021-08-11
> > > > > **Re-Re-response to Reviewer 2WyM**
> > > > >
> > > > > Thanks for the authors' reply!!
> > > > > **Are the authors confusing the results of minimax regret and expected pseudo regret?**
> > > > >
> > > > > - The optimal regret I am pointing out is a regret in the sense of expected pseudo regret, such as Lai and Robbins [1985].
> > > > > - In the previous reply, the authors cited the results of expected pseudo regret and minimax regret mixed together, but they are different.
> > > > > - To achieve an order of $\log T$ in that sense of expected pseudo regret, we need an explosion rate that decays faster than $1/t$.
> > > > > - This can be shown by integration; that is, the integral of $1/t$ is $\log T$.
> > > > > - On the other hand, I do not know how much is needed for the explosion rate in the case of minimax regret.
> > > > > - For example, if we have an algorithm with an expected pseudo regret $\sqrt{T}$, then the exploration rate should be the order of $1/t^{1/2}$. On the other hand, even if we use the exploration rate with the order of $1/t^{1/2}$, I do not know if the algorithm can achieve the order $\sqrt{T}$ in the sense of minimax regret. I think it will be worse order.
> > > > > - I'm not familiar with the minimax regret, so I'm not sure I can give a rigorous discussion on the minimax regret. However, from the viewpoint of the expansion rate, we should consider the expected pseudo regret in this rebuttal.
> > > > > - In the sense of expected pseudo regret, the following papers guarantee the regret of $log T$ for the linear contextual bandits.
> > > > > [1] Goldenshluger and Zeevi. A Linear Response Bandit Problem. Stoch. Syst. 2013.
> > > > > [2] Bastani and Bayati. Online Decision Making with High-Dimensional Covariates. Operations Research 2019.
> > > > > [3] Tsuchiya et al. "Analysis and Design of Thompson Sampling for Stochastic Partial Monitoring" NeurIPS2020
> > > > > - Also, even looser than that, for example, around $\sqrt{T} log T$ is usually achieved. However, I do not think that the authors' method can be applied to that loose regret either.
> > > > >
> > > > > For clarifying the authors' contribution, I would like you to answer the following questions one by one:
> > > > >
> > > > > ### Q1.
> > > > > The authors' method is only applicable to bandit algorithms that are larger than $T^{2/3}$ in the sense of expected regret. Is this True?
> > > > >
> > > > > ### Q2.
> > > > > In their previous reply, the authors answered
> > > > > > The optimal regret of stochastic contextual bandits is generally much bigger than $\log T$, and the particular rate depends on the particular problem class. For example, the problem in Dudik et al. 2011 has the optimal regret rate, and the problem in Bibaut et al. 2020 has the optimal regret rate of  ...
> > > > >
> > > > > but the theoretical analysis of Rigollet and Zeevi (2018) is expected regret, and Hu et al. (2020) is minimax regret? I think that they are different. Is it True?
> > > > >
> > > > > ### Q3
> > > > > In their previous reply, the authors answered
> > > > > > The optimal regret of stochastic contextual bandits is generally much bigger than $\log T$
> > > > >
> > > > > but as I cited, there are studies showing $\log T$ regret. So, I think the authors' reply is False. Is it True?
> > > > >
> > > > > ### Q4
> > > > > The authors cited Rigollet and Zeevi (2018). Does this mean that the authors' method intended for application to the very special case, such as nonparametric contextual bandits?
> > > > >
> > > > > ### Q5
> > > > > In the Introduction,  the authors cited Li et al. (2010), which considered linear contextual bandits, but for linear contextual bandits, we can show roughly $\sqrt{T}\log T$ regret, and we can also achieve $\log T$ regret as in the paper I cited. Can the authors' method apply to linear bandits?
> > > > >
> > > > > ### Q6
> > > > > Can the authors' method be adapted to the method of Rigollet and Zeevi (2018)? Certainly, in the nonparametric case regret maybe $T^{\alpha}$ for $\alpha < 1$. In that case, the exploration rate would certainly be looser than $1/t$. On the other hand, $p$ in Assumption 6 of the authros' manuscript also seems to increase when considering nonparametric models. Can the author show that the authors' method can apply to the algorithm of Rigollet and Zeevi (2018) while achieving the regret of Rigollet and Zeevi (2018) and satisfying the authors' Assumptions 5 and 6 simultaneously? If so, what are the circumstances?
> > > > >
> > > > > ### Q7
> > > > > In Hadad et al. (2019), Theorem 3, $1/t^{\alpha}$ holds for $\alpha \in [0, 1)$, so we can evaluate bandit algorithms, which are suboptimal, but close to $\log T$ regret. On the other hand, the authors' result may look similar to Hadad et al. (2019) but is quite different. The $\alpha$ of $1/t^(\alpha)$ in the authors' Assumption 6 is at least $1/3$. This means that it suffers $T^(2/3)$ in the sense of the expected regret, and I feel that it is hopelessly inapplicable to bandit algorithms, unlike Hadad et al. (2019). Is it True?
> > > > >
> > > > > ### Q8
> > > > > In the rebuttal reply, the authors introduced the case of nonparametric bandits, but if the result is theoretically consistent only with such a limited case, why did the authors not specify it?  (although I am afraid that the authors' result may not hold even in that situation with nonparametric bandit).
> > > > >
> > > > > For some questions, I also would like to see the proofs, such as Q1, Q3, Q5, Q6, and Q7,
> > > > >
> > > > > **The reason I voted "clear reject" is that the title and claims of this paper are different from the results being derived.**
> > > > > **The title of this paper would lead the reader to imagine that it could be applied to general contextual bandits.**
> > > > > **In addition, I think that the situation where this algorithm can apply is very limited. Compared to the results of Hadad et al., which can evaluate algorithms with regret close to $\log T$ order, the authors' results are quite different.**
> > > > > **If it can only be used in very limited situations, the authors should have written about its limitations. Also, they should have shown that Assumption 5 and 6 are valid for some well-known algorithms.**
> > > > >
> > > > > I am afraid I also have to disagree with that the following comment
> > > > > > Nonetheless, we still argue that it is a significant theoretical advance to be able to tackle for the first time valid statistical inference from contextual bandit algorithms that obtain sublinear regret, as such has never been possible before. We strongly believe that this is a significant breakthrough on a high-profile problem. It may well be that in certain classes of contextual bandit problems, our analysis can and may need to be tightened to permit optimal exploration, and future work looking at specific classes of problems (e.g., anyone problem studied by any one of the many papers above) may seek to do this.
> > > > >
> > > > > I will send my reply to this comment after I receive satisfactory answers to the above eight questions.

---

> > > > > > ### Author Response · Authors · 2021-08-11
> > > > > > **Re-Re-Response to Reviewer 2WyM**
> > > > > >
> > > > > > No; we are **not** "confusing the results of minimax regret and expected pseudo regret" as we know extremely well this literature. In fact, in stochastic contextual bandit problems, **minimax expected regret is the standard notion of hardness of a class of problems** as it formalizes the notion that a bandit algorithm does not know a priori the particular problem instance in the class, as it is equal to the minimal expected pseudo regret that can be uniformly be obtained on a class (in contrast, on a given instance, one can always obtain zero regret by playing the optimal policy; minimax regret is exactly how one mathematically formalizes that this is "cheating"). Given a class of instances $\mathcal Q$ (set of joint distributions over contexts and potential rewards for each arm), the minimax expected regret is defined as $\inf_{g_t}\sup_{Q\in\mathcal Q}\sum_{t=1}^T (P_{Q,g_t}\bar Q-P_{Q,g^*_{Q}}\bar Q)$, where the infimum is over random policies adapted the filtration of the data (i.e., non-anticipatory) and $g^*_Q$ is the optimal policy for the instance $Q$. The inner sum is the "expected pseudo regret" for the policy on instance $Q$, so this is the minimum over algorithms of the maximum over instances of the expected pseudo regret.
> > > > > >
> > > > > > **The very works you cite such as Goldenshluger and Zeevi, 2013 study the minimax expected regret**. We quote from the abstract of Goldenshluger and Zeevi (2013): "We study this problem in a minimax setting". See their Eq. (1), where minimax regret is defined.
> > > > > >  Goldenshluger and Zeevi (2013) show logarithmic **minimax** regret when $\mathcal Q$ is crucially restricted in two important ways (among other things): conditional mean rewards are linear and a Tsybakov margin assumption holds with exponent 1 (analogue of the arm gap for contextual problems). (By the way, with a lower Tsybakov margin exponent, $\alpha<1$, Bastani et al. 2017 "Mostly Exploration-Free Algorithms for Contextual Bandits" Appendix D.1 show that the minimax regret becomes polynomial.) Indeed, Goldenshluger and Zeevi's rate-optimal algorithm does not have a decaying uniform exploration of $1/\sqrt{t}$ as that would entail a suboptimal regret rate in this class of problems. Correspondingly, their algorithm does not fit into our framework. This is completely in line with what we wrote in the above response: "It may be that in some certain problems, our exploration assumptions require more exploration than is optimal for that problem". We also repeat another point: "**we never claimed to tackle rate-optimal contextual bandit algorithms, although we agree we can make this point more explicit**".
> > > > > >
> > > > > > While some algorithms do not have a decaying per-step uniform exploration and therefore do not fit into our framework, other algorithms such as all of decay-epsilon-greedy, PolicyElimination, RandomizedUCB, ILOVETOCONBANDITS, FALCON, AdaCB, GeneralizedPolicyElimination, SmoothFTL, and HINGE-LMC do have a decaying per-step uniform exploration. Some of these have a faster decay than our analysis is currently able to handle. This is again completely in line with what we wrote in the above response: "It may be that in some certain problems, our exploration assumptions require more exploration than is optimal for that problem".
> > > > > >
> > > > > > Moreover, a note on **non-contextual** bandit problems. You mentioned Lai and Robbins (1985); that paper studies the **non-contextual** bandit problem. The $O(\log T / \Delta)$ regret bound they show matches the instance-dependent lower bound for so-called consistent bandit algorithms. In the bandit literature (see e.g. page 206 of Bandit algorithms, Lattimore and Szepesvari, 2020), a consistent bandit algorithm refers to an algorithms that has regret rate $O(T^\alpha)$ for any $\alpha$ and any fixed bandit problem instance). That is not to say that the minimax regret for the problem is better than $\sqrt{T}$. In fact, the very techniques used to show this instance dependent lower bound imply that the minimax regret is at least $\sqrt{T}$ for this problem. This is not a distinction between "minimax regret and expected pseudo regret" -- this is a distinction between "minimax regret and instance dependent lower bounds" which depend on parameters of the environment instance at hand such as a gap (correspondingly, a Tsybakov margin condition in contextual bandits).
> > > > > >
> > > > > > We finally respectfully disagree on the significance of our results. As we wrote before, we still **strongly believe that ours is a significant breakthrough on a high-profile problem, as it enables for the very first time valid statistical inference from data collected contextual bandit algorithms that have any opportunity of sublinear regret**.

---

> > > > > > > ### Comment · Reviewer_2WyM · 2021-08-12
> > > > > > > **Re-Re-Response to Reviewer 2WyM**
> > > > > > >
> > > > > > > We appreciate the authors for replying to my comments on minimax regret.
> > > > > > > I disagree with some of the authors' opinions on minimax regret and expected pseudo regret.
> > > > > > > However, the issue of minimax regret is not intrinsically and directly related to the contribution of this paper, so I will give my opinion in the next reply after I get the answers to the main questions.
> > > > > > >
> > > > > > > **What is more important is whether the authors' method can be used for algorithms with such orders of expected pseudo regret or minimax regret.**
> > > > > > > **Could you please answer Q1 to Q8 of my previous question?**
> > > > > > > **My questions are quite simple but important: Is there a theoretical situation in which we can use the authors' proposed method?**
> > > > > > > **And, if there is, can the authors clarify the limitation?**.
> > > > > > >
> > > > > > > Please mathematically show that Assumption 5 and 6 can be achieved for some well-known algorithms.
> > > > > > > After all, doesn't the authors' method assume a least $T^{2/3}$ regret even in a fairly ideal situation?
> > > > > > > I think it's hopeless in linear bandits.
> > > > > > > On the other hand, in the case of nonparametric bandits, doesn't $p$ in Assumption 5 become larger?
> > > > > > >
> > > > > > > I hope to get an answer to each of the questions, Q1-Q8.
> > > > > > > I think that answers to these questions are necessary to clarify the theoretical novelty of the authors.
> > > > > > >
> > > > > > > I would not have voted for clear rejection if the theoretical limitations had been clearly stated in the paper, and if there had been proof for specific situations where it could be used, but there was not.

---

> > > > > > > > ### Author Response · Authors · 2021-08-13
> > > > > > > > **Final Response to Reviewer 2WyM**
> > > > > > > >
> > > > > > > > Let us begin by pointing out that we are not stating “opinions" about minimax regret and expected pseudo regret -- we are stating well-established definitions from the bandit literature and basic mathematical facts. These are facts known very well to all who are familiar with the literature, not "opinions" that you can "disagree" on, as we also have no doubt is apparent to anyone else reading this thread. It is ok if you are not an expert in the literature, but the appropriate response would be to recognize that you did not know the details of what you were talking about and calibrate your confidence score appropriately.
> > > > > > > >
> > > > > > > > Despite our efforts to make this exchange constructive, you have not reciprocated with constructive responses. Your “grounds for dismissal” of our contribution have pivoted and contradicted themselves several times over the course of the thread:
> > > > > > > > - from “variance stabilization weights are useless for most problems because IPS is asymptotically normal as long as the logging policy converges'' which is widely known to be a **false** statement,
> > > > > > > > - to “previous work has solved the same problem with simpler methods” which is also a **false** statement as all previous OPE methods on contextual bandit data require the logging policy to have linear regret and the very authors of these prior works you cited have confirmed that this is true,
> > > > > > > > - to “this contribution is not significant because it doesn’t apply to OPE from data collected by a contextual bandit with logarithmic regret” which both mischaracterizes the vast body of work on contextual bandits where many algorithms enforce a $1/\sqrt{t}$ rate of exploration (or more) and minimax regret in many problem classes being studied is higher than logarithmic (besides the linear&sharp-margin setting), as well as mischaracterizes our paper and response, neither of which claimed this,
> > > > > > > > - to “our goal of applying OPE to data collected by a contextual bandit with sublinear regret is very pessimistic” even though this is a high-profile open problem in the literature and even though many of the respected works you cited as superior to ours in your initial review focus on OPE from data collected by contextual bandits with linear regret.
> > > > > > > >
> > > > > > > > Despite these pivots and errors -- and also disregarding several condescending remarks you have made towards us along the way -- we have put a lot of time and effort into diligently responding to you. We have provided you with a detailed authors’ response answering all your initial set of questions from your initial review and trying to make the most out of it by recognizing some clarity gaps potentially left to the reader and discussing how we can fill them in our revision (e.g., by further discussing our assumptions and providing examples about the settings that fit and do not fit our framework, as we did in our responses to you). We have also taken the time to correct you on your incorrect statements about contextual bandits and explain to you in detail the literature's basic definitions and results. Further, we have provided you with detailed follow-ups answering your new set of questions (including to your questions Q1-Q8, if you read our follow-up responses carefully) and providing a literature review supporting our points.
> > > > > > > >
> > > > > > > > For the last time, we will re-iterate the main points we have made in all our previous responses:
> > > > > > > > - Assumptions 5 and 6 are far less restrictive than conditions required by previous work. Previous work required either iid data, or non-contextual multi-armed bandit data, or contextual bandit data but where we converge to a policy with lower-bounded exploration on all actions and therefore must suffer linear regret (none of this of course doesn’t make these works less pioneering as they each solved very important problems which stood open at their respective writing times). Thus, our results permit for the very first time valid inference from data collected by **contextual** bandit algorithms with **vanishing exploration** (and even potentially non-converging policies).
> > > > > > > > - We have not claimed that our estimator applies to rate-optimal contextual bandit algorithms -- and besides rate-optimality is a notion that may only be defined for a given bandit problem class, to which we make no reference -- but we strongly affirm the practical importance of our estimator and the significance of the theory. From a theory perspective, its merit is to motivate the design of an algorithm that is valid under some assumptions that are less restrictive than those in prior work and might prove experimentally robust to certain violations thereof that arise in practical settings. For instance, the linear realizability assumption required by milestone papers on linear bandits almost never holds in real-world problems, yet this doesn’t make the theory of linear bandits any less significant. From a practitioner’s perspective, one may want to run a contextual adaptive experiment and tolerate a level of additional online regret to support valid offline inference afterwards; our estimator is designed in such way so that it requires a slower decaying exploration rate which achieves sublinear regret rather than uniform exploration which results in linear regret, as all unbiased and asymptotically normal OPE estimators on contextual bandit data have done so far.
> > > > > > > > - The CADR estimator our paper proposes enables for the very first time valid statistical inference from data collected by contextual bandit algorithms that have any opportunity at sublinear regret. This is a big deal given that all previous OPE estimators on contextual bandit data required linear regret. Furthermore, moving from estimators appropriate for linear-regret logging policies to sublinear-regret logging policies requires a significantly different estimator design and analysis. Tighter analysis for certain classes of contextual bandit problems to enable optimal exploration is an excellent avenue of future work that may need to crucially rely on assumptions made in a given class of problems (such as linear; we make no such assumptions). That said, our paper has proposed an estimator that closes an important literature gap, as shown by both the theory and our extensive experimental section.
> > > > > > > >
> > > > > > > > It appears that, regardless, you are determined to declare facts as "opinions" you can "disagree" with and to discredit the merits of our work with a purported highest confidence score despite clear gaps in your arguments. You back your assessment with clearly wrong mathematical facts and statements counter to what has been well-established in the literature (regarding the contributions of the previous literature, regarding basic regret definitions, regarding assumptions that guarantee normality of the IPS estimator, etc.). You have not produced a single reference to an off-policy evaluation method that provides asymptotic normality guarantees under data collected by a contextual bandit that isn’t required to have linear regret, like our paper does.
> > > > > > > >
> > > > > > > > Given all this, we don’t believe there is anything more for us to add to this thread going forward.

---

> > > > > > > > > ### Comment · Reviewer_2WyM · 2021-08-13
> > > > > > > > > **Re: Final Response to Reviewer 2WyM**
> > > > > > > > >
> > > > > > > > > If I have offended the authors, it is my fault.
> > > > > > > > > I am a non-native English speaker, and my writing may have been poor.
> > > > > > > > > However, I do not think that I undermined the authors' research.
> > > > > > > > > I think that the authors did not try to answer my question directly and shifted the point.
> > > > > > > > > Furthermore, the last reply is one-sided. I have volunteered a great deal of my time to review the manuscript, and this is very disrespectful of that effort.
> > > > > > > > >
> > > > > > > > > I wanted the authors to answer my questions mainly consisting of the following three topics:
> > > > > > > > > 1. Is not the authors' algorithm, at best, applicable only for algorithms with $T^{2/3}$ regret (either minimax or expected pseudo regret)?;
> > > > > > > > > 2. Are not such situations extremely limited (such as nonparametric bandit)?
> > > > > > > > > 3. Could the authors prove that some specific algorithms can satisfy Assumptions 5 and 6?
> > > > > > > > > 4. Is it misleading to the readers not to discuss this in the paper?
> > > > > > > > >
> > > > > > > > > For questions consisting of these topics, I wanted direct answers with mathematical explanations, e.g., proof of the topic 3.
> > > > > > > > >
> > > > > > > > > Even if the proof is correct, it may be a theoretical error because it is a bit different from what is written in the paper.
> > > > > > > > > Even if there is a novelty in the authors' method (which should be discussed separately), I think it is a bad thing from an academic point of view that the limit of the method is rarely discussed.
> > > > > > > > >
> > > > > > > > > I am very disappointed by the last reply from the authors because I worked hard to keep the reputation of the authors and the NeurIPS community.

---

> > > > > > > > > > ### Author Response · Authors · 2021-08-24
> > > > > > > > > > **Point-by-point answers**
> > > > > > > > > >
> > > > > > > > > > For your reference, the answers to your last set of questions are the following. We have answered these either in our initial response or in our follow-up responses, but for the sake of clarity we outline the answers here point-by-point:
> > > > > > > > > >
> > > > > > > > > > 1. Yes, that is what Assumption 6 in our manuscript says, but this slow exploration decay assumption, which may, in some classes of problems, result in higher-than-optimal regret rate, is still better than the linear regret required by all state-of-the-art OPE estimators on contextual bandits that satisfy the asymptotic normality property. So, our estimator closes an important gap in the literature by addressing asymptotically normal OPE from contextual bandits that are not required to have linear regret and required a very different design and analysis.
> > > > > > > > > > 2. No, this is not an impractical assumption given that the alternative for obtaining valid confidence intervals would be to suffer linear regret. Given that valid inference is often a practical requirement, our results are an important and practical step forward. Note that we crucially do not rely on the realizability assumptions that would be needed to ensure a given contextual bandit algorithm obtains rate optimality in a given class of problems -- we focus on completely model-agnostic inference, and that comes with some costs in the analysis. Finally, the method itself does not actually rely on the strengthened rates in Assumption 6, only the analysis.
> > > > > > > > > > 3. Yes, and we provided a thorough literature review on bandits that have slower exploration decaying rates and specified in detail which ones fit our framework in our responses from Aug 10 and Aug 11, in which we also mentioned that we will include this discussion in the revision.
> > > > > > > > > > 4. No, the goal obviously is not to mislead the reader and we have been saying in every single one of our responses that we recognize there may be points where an additional sentence or two can offer more clarity to the reader. As we have mentioned before, we will add to the discussion that, while we address for the first time asymptotically normal OPE from contextual bandits with sublinear regret, an excellent avenue for future work may be tighter analysis for certain classes of contextual bandit problems to permit less exploration, while doing this would likely need to crucially rely on assumptions made in a given class of problems (such as linear response; we make no such assumptions).

---

> > > > > > > > > > > ### Comment · Reviewer_2WyM · 2021-08-24
> > > > > > > > > > > **Re: Point-by-point answers**
> > > > > > > > > > >
> > > > > > > > > > > Thanks for the authors' reply!
> > > > > > > > > > >
> > > > > > > > > > > I have a couple of questions, but let me start with the most important one.
> > > > > > > > > > >
> > > > > > > > > > > > No, this is not an impractical assumption given that the alternative for obtaining valid confidence intervals would be to suffer linear regret. Given that valid inference is often a practical requirement, our results are an important and practical step forward. Note that we crucially do not rely on the realizability assumptions that would be needed to ensure a given contextual bandit algorithm obtains rate optimality in a given class of problems -- we focus on completely model-agnostic inference, and that comes with some costs in the analysis. Finally, the method itself does not actually rely on the strengthened rates in Assumption 6, only the analysis.
> > > > > > > > > > >
> > > > > > > > > > > > Yes, and we provided a thorough literature review on bandits that have slower exploration decaying rates and specified in detail which ones fit our framework in our responses from Aug 10 and Aug 11, in which we also mentioned that we will include this discussion in the revision.
> > > > > > > > > > >
> > > > > > > > > > > Could the authors prove these during this rebuttal?
> > > > > > > > > > > Please show that Assumptions 5 and 6 hold (or nearly hold even if they don't hold exactly) for the linear bandit and nonparametric bandit methods, respectively, proposed in the widely accepted papers, such as Rigollet and Zeevi 2010, which the authors cited in the previous rebuttal.
> > > > > > > > > > >
> > > > > > > > > > > I think this is a pretty tough problem. In particular, in the case of nonparametric bandits, I suspect that the computation of $p$ in Assumption 5 will be difficult. This $p$ will increase as the problem is complicated, and it requires a larger exploration rate.
> > > > > > > > > > >
> > > > > > > > > > > The main reason I am voting for "clear reject" is that this is probably only applicable in very special cases.
> > > > > > > > > > > If the authors claim that it is False, then I think the authors need to prove it.
> > > > > > > > > > >
> > > > > > > > > > > After answering this question, I would like to ask about sub-linear regret, comparing this study with Hadad et al. (2020).

---

### Official Review · Reviewer_hz9e · 2021-07-10

**Rating:** 7
**Confidence:** 3

**Summary:**

This paper is motivated by the fact that no asymptotically normal estimators exist when the data is collected context-dependent adaptively. This work provides such estimators (named CADR, Algo. 1) exhibiting asymptotic normality (Thm. 1).

**Limitations And Societal Impact:**

Yes. Assumptions are written in explicit and transparent manner. Still, I think it'd be more helpful if authors can provide a DAG representation for encoding the dependency assumption in Line 62-72.

**Main Review:**

This paper is motivated by the fact that no asymptotically normal estimators exist when the data is collected context-dependent adaptively. This work provides such estimators (named CADR, Algo. 1) exhibiting asymptotic normality (Thm. 1).

This paper is based on the solid theoretical foundation and sophisticated observation that asymptotically normal estimators have only been developed under a non-contextual adaptive setting. This observation came as a surprise to me because the contextual adaptive setting is quite practical, but we still don't have a good estimator on it. Theories are strong, and experiments corroborate with the theory.

Here are some points that I'd like to ask for authors to help readers' understanding a bit more.
* C1. Line 30-32 — "However, due to the adaptive nature of the data collection, unlike classic randomized trials, standard estimates and their confidence intervals actually fail to provide correct coverage" seems to be a key motivation of the paper. I think it should be more enlightened to make readers motivated.

* C2. Line 62-72. It would be helpful if you could provide a graphical representation (i.e., DAG) for encoding the assumption, such as "Xt is independent of all else given A(t)." This would help to be more explicit and automatically tell some questions such as "What is all-else in the assumption"?

* C3. Line 110-117. Can you formalize this observation with some examples? Despite the importance of this paragraph (this paragraph gives the very motivation of this work), explanations seem weak.

I have one question about this work.
* Q1. In Algo. 1, did you use sample-splitting (or cross-fitting) technique? If not, does the result still hold?



**Time Spent Reviewing:**

4 hours

---

> ### Author Response · Authors · 2021-08-10
> **Authors' Response to Reviewer hz9e**
>
> Thank you for your review and your encouraging comments. We are glad to hear that you found our contribution valuable to the community and our empirical and theoretical results strong and we appreciate your feedback.
>
> We respond below to your comments and question in the order they appear in the review.
>
> - C1: Thank you for the very helpful feedback on how we can better motivate the reader. We will add a paragraph break before this sentence and complement it with the statement that addressing this challenge is the main motivation of our work, and in particular explain the new technical challenges posed in the contextual setting, where the it is necessary to take into account both the context and the arm when estimating the local variance of a datapoint in order to design unbiased and asymptotically normal estimators.
>
> - C2: That is a very good suggestion to capture the statement succinctly with a graphical model and we will add this in the camera ready. To be clear, when we write “all else” here, we literally mean all other variables, so it is a specific mathematical statement (but certainly it can be nicely illustrated graphically).
>
> - C3: That is a great idea. To complement the extensive empirical investigation with real datasets later in the paper, we will include early on here a simple histogram figure using a very simple simulation setting that shows that, over replications, all of IPS, DR, and ADR are unbiased but not normal, while CADR is both. Essentially, we will extend Fig 1 of Hadad et al. (2019) to the contextual setting.
>
> - Q1: The approach we proposed in the main text uses sequential sample splitting, i.e, it estimates the outcome model for the CADR score of observation O(t) only using data O(1), ..., O(t − 1). This has the same property as cross-fitting that the fitted model is independent of the observation to which it is applied. The empirical results of Figure 1 in the main paper are based on this sequential sample splitting, consistent with our theory. In Appendix E2 we also consider a non-sequential cross-time cross-fitting, which splits the data into folds over time and excludes adjacent folds when fitting a model in order to avoid dependence and which may facilitate running fewer model fitting subroutines. Figures 2 and 3 of appendix E2 show that same conclusions regarding the benefits of CADR persist even when using this alternative fitting procedure when compared to all other baseline estimators on the 57 OpenML-CC18 datasets, 4 target policies, and linear outcome regression models.

---

### Official Review · Reviewer_hHgb · 2021-07-14

**Rating:** 7
**Confidence:** 4

**Summary:**

This paper presents a method for estimating the value of some fixed evaluation policy of interest using data that was collected with a contextual bandit algorithm. They prove that their estimator is asymptotically normal even when the model of the expected reward conditional on state and action is misspecified.

**Limitations And Societal Impact:**

See above comments.

**Main Review:**

The result the authors present is interesting and appears to be an important contribution to the literature particularly because the model of the reward model can be misspecified. However, I believe that the paper is currently still weak because (1) insufficient discussion of the relationship to previous literature and (2) limited empirical evaluation. I would be willing to raise my score if the authors improve in these two respects.

Regarding previous literature
-	I am still unclear about how exactly your method differs from that of Hadad et al., 2019 and I think there should be additional discussion. In a non-contextual setting, is your estimator equivalent to that of Hadad et al. in a special case? How is your weighting scheme similar / different from that of Hadad et. al, 2019? There should be a more explicit comparison to understand how your contribution is different.
-	Line 127-128: This comment is vague. Could you elaborate on the relationship to Hadad et al., 2019. Later at Line 256, you say the main focus of the paper is that the estimator of Hadad is biased on contextual bandit data; this should be stated earlier if this is the case. And more discussion should be made, is your proposed estimator unbiased on contextual bandit data? What exactly is your argument that Hadad et. al, 2019 is insufficient for contextual bandit data?
-	How is your work related to more recent related work:
o	Off Policy Evaluation via Adaptive Weighting with Data Collected from Contextual Bandits: https://arxiv.org/abs/2106.02029
o	Statistical Inference with M-Estimators on Adaptively Collected Data: https://arxiv.org/abs/2104.14074

Empirical Evaluation
-	The Figure 1 plots are quite difficult to interpret, what do the colors mean? What are the different settings you consider? How do you compute standard errors?
-	In your empirical evaluation you have T=10000, it is unclear how your method performs in smaller sample settings, which is commonly the case in real world problems.
-	How is the empirical performance of your estimator affected if the model Q does not converge to the true value (if at all)?
-	Are there any limitations or downsides of your approach empirically due to the large number of terms that must be plugged-in?

Additional Questions
-	Line 80: You define g^*, but in the equation above you use g. Is this intentional?
-	Theoretically / in practice do we expect Assumption 4 to hold?


**Time Spent Reviewing:**

3 hours

---

> ### Author Response · Authors · 2021-08-10
> **Authors' Response to Reviewer hHgb: Main Response**
>
> Thank you for taking the time to review our paper. We were glad to read that you think our paper makes an important contribution to the literature.
>
> We address your main concerns below and explain how our estimator is fundamentally different from previous work and how the empirical evaluation establishes the advantages of our estimator in a wide variety of datasets, target policies, and choices of outcome regression estimates.
>
> First, we would like to establish that our estimator is different from that of Hadad et al. 2019 in that:
> 1. Our estimator is suitable for unbiased and asymptotically normal inference in the *contextual* bandit problem setting which is more general than the (non-contextual) multi-armed bandit setting considered in Hadad et al. 2019 (in their paper it is stated that their estimator satisfies the unbiasedness and asymptotic normality properties only for non-contextual, multi-armed bandit data).
> 2. Although both our estimator and the estimator in Hadad et al., 2019 assume the same form of a weighted doubly robust estimator, the way we construct the stabilizing weights is very different and it is this novel construction of the stabilizing weights that allows our estimator to be unbiased and asymptotically normal in the more general contextual setting. Our weighting scheme incorporates the presence of a contextual propensity score from the logging policy and of a contextual outcome model used by the estimator when estimating the local variance of each datapoint’s doubly robust score. On the other hand, in Hadad et al. 2019 the local variance is proxied only by the non-contextual propensity score. Hence, the way we estimate the stabilizing weights is more general and doesn’t reduce Hadad et al. 2019 stabilizing weights.
> 3. We need to estimate the stabilizing weights while they are known a priori in the non-contextual setting of Hadad et al., 2019. This further requires a special sequential cross-fitting procedure, where we only use past data, and a thoroughly new analysis.
> These statements are true both in the case of a well-specified and misspecified model. E.g. in Figure 1 in the main paper and Figure 4 in the supplemental material one can see that CADR far outperforms ADR (Hadad et al. 2019’s estimator) in the contextual bandit setting both when the estimators’ models are misspecified (Figure 1) and well-specified (Figure 4). In fact, how different CADR is from the estimator in Hadad et al. 2019 is established throughout the entire experimental section, where ADR achieves much lower coverage than CADR when they are both applied on data collected by a contextual bandit under a wide range of different settings. We would like to stress that none of this is to detract from Hadad et al. 2019’s pioneering work in the area: it is only meant to stress that they consider a different setting and illustrate that if one naively applied their method in a different setting than they intended then the method obviously fails, as expected.
>
> We agree with you that these points need to be further emphasized early-on in the paper and we will do so per your suggestion to enhance clarity for the reader.
>
> Second, we are sorry to hear that you felt our empirical evaluation was limited, but we respectfully disagree. We worked hard to have an extensive, thorough, and reproducible empirical evaluation. To produce the 7 figures in the paper (1 in the main paper and the additional 6 figures in the supplemental material; code included in the supplemental material) we worked with:
> - 57 real-world classification datasets converted to contextual bandit datasets (this is a wide-spread evaluation approach in the offline and online contextual bandit literature)
> - 4 different target policies
> - 6 estimators (CADR, and 5 other popular baseline estimators in the literature).
> - 6 different estimation procedures
>   - sequential cross-fitting vs. cross-time cross-fitting (Figures 1 and 3)
>   - misspecified vs. well-specified outcome model family (Figures 4 and 5)
>   - weighted vs. unweighted outcome model fitting (Figures 6 and 7)
> - 64 simulations were run for each dataset/target policy/estimator/estimation procedure combination to obtain and report tight standard errors of their performance.
>
> The goal of this thorough empirical study was to give the reader the sense of how these decisions of fitting the estimators affect the performance of CADR over baselines. In all these scenarios, CADR significantly outperformed alternatives and hopefully we were able to indicate under which settings CADR’s advantages should be expected to be even more pronounced.
> We are thankful that your review suggested a 4th comparison setting for us to include, that of large data vs. small data, which we found interesting and have added it. This was easy given the reproducible code we submitted in the supplemental material and that will be publicly available with any published version. Although CADR’s normality is an asymptotic property, we believe it is informative to show the performance of all estimators under smaller sample size and so we ran it with T=1000. The results are qualitatively the same. They show that, although all estimators’ coverage deteriorated, CADR’s relative advantages held up in such a “small-data” setting as well. We will include this additional figure as Figure 8 in the supplement.
>
> In the next thread, we follow up with detailed answers to your other points and questions you raised, in order to address them one-by-one as they appear in the review.

---

> > ### Author Response · Authors · 2021-08-10
> > **Authors' Response to Reviewer hHgb: Detailed Point-By-Point Answers to Comments & Questions**
> >
> > We now answer your other points and questions you raised one-by-one in the order they appear in the review.
> >
> > Previous Literature:
> > - Addressed in main response above.
> > - Addressed in main response above.
> > - Thank you for this question. We would be very glad to comment on the connections and differences to these works as this would add clarity to our readers. Of course we could not have done this in the original submission as the first work “Off Policy Evaluation via Adaptive Weighting with Data Collected from Contextual Bandits” was posted after the deadline, while the second “Statistical Inference with M-Estimators on Adaptively Collected Data” was posted soon before.
> >   - The second work is least related to ours: they consider inference for the coefficients of a well-specified parametric models; this has a different intended purpose. If the outcome model (that is the conditional distribution of reward given context and action) can be modeled by a parametric model, then their method could be applied to perform inference for the coefficients of this parametric model. For any given target policy, a plug-in estimator based on estimators of the outcome model's coefficients could then be derived. The main limitation of this approach is that it requires correct parametric specification.
> >    - The first work is more closely related to ours as it is also motivated by establishing an unbiased and asymptotically normal estimator from data collected by a contextual bandit. Both our method and theirs rely on adaptively weighted DR scores and yield, under conditions, asymptotically normal estimators. However, the way they construct their stabilizing weights appears quite different from ours. In particular, their weights at time t (a) do not incorporate an outcome model and do not rely on it and (b) depend only on the propensity scope. As a result of these different constructions, our analysis and theirs require different sets of conditions. While we require slightly more stringent conditions on the exploration rate and the complexity of the logging policy class than they do (our assumptions 5 and 6), they require some form of asymptotic stabilization of the logging policies (their conditions (8) and (12)). These two sets of conditions therefore cover different settings, and our respective approaches are thus complementary. An instance where the logging policy doesn’t stabilize is, for example, when various actions have the same expected reward on certain subsets of the context space. Addressing this situation in the iid setting is specifically what motivated Luedkte and van der Laan (2016) to develop their stabilizing weights approach. In that sense, our approach can be seen as an adaptive generalization of Luedtke and van der Laan (2016) while we see their approach as generalizing to the contextual setting Hadad et al. (2019).
> >
> > Empirical Evaluation
> > - Thanks for the clarification questions.
> >   - On lines 292-294, we write “The dot is depicted in blue if for that dataset CADR has significantly better coverage than the baseline estimator, in red if it has significantly worse coverage, and in black if the difference in coverage of both estimators is within one standard error.” We will add a summary of this to the caption of Figure 1. We will also replace the word “significantly” with “by more than one standard error” so to be specific.
> >   - The different settings are described on lines 280-285. Each row in Figure 1 represents a different candidate policy to be evaluated. Additional settings are considered in the supplementary materials in Figures 2-7.
> >   - Standard errors are computed as sample standard deviation over the 64 replications that we ran for each dataset, estimator, and target policy, divided by sqrt(64)=8. We will clarify this.
> >
> >   As described in the paper, we operate in the setting of generating contextual bandit datasets from 57 fully-labeled, public multi-class classification datasets (Open-ML CC18 suite) and then computing the value of various target policies using the CADR estimator or one of several baseline off-policy evaluation estimators using the contextual bandit data from each one of these datasets. This conversion of a multi-class classification dataset into a contextual bandit dataset has the benefits of reproducibility using public datasets and being able to make uncontroversial comparisons using actual ground truth data with counterfactuals, hence it is widely used for empirical evaluation in the offline and online contextual bandit literature (Dudik et al. 2014, Dimakopoulou et al. 2017, Bietti et al. 2018, Su et al. 2019, Zhan et al. 2021 to name a few). A description of how we generate a contextual bandit dataset from each classification dataset is detailed in lines 270-279.
> >
> > - Thanks for raising this point. The purpose of using large samples is that the normality property holds in the asymptotic regime. That said, in more moderately sized samples we should still expect an asymptotically normal estimator to perform better than an estimator which is not asymptotically normal. So, motivated by your point, we re-ran the same experimental setup but with T=1000, which is easy given the reproducible code provided in the supplementary material, and although all estimators’ coverage deteriorates with smaller samples, CADR still remains the clearly best performing estimator. We will add this large data vs. small data comparison in the appendix to complement the many other comparisons we have already included regarding sequential cross-fitting vs. cross-time cross-fitting, misspecified vs. well-specified outcome model family, and weighted vs. unweighted outcome model fitting.
> >
> > - Thank you for raising that point. In the paper, we show the performance of our estimators both for a parametric linear regression estimator (which is expected to be misspecified and not converge to the ground truth under real data) and for a non-parametric decision tree estimator (which is considered more expressive to model the ground truth). Figure 1 shows the estimator’s performance for the former, when Q is linear and therefore misspecified and not converging to the ground truth. Section E.3 of the supplemental material discusses the performance of all estimators when Q is a decision tree. CADR has a clear advantage in both settings. In fact, the advantage of CADR is even more pronounced over DR in the case where Q is misspecified, its error is larger and it is that error that is multiplied by a potentially large importance-sampling weight de-stabilizing the variance and exacerbating asymptotic normality. CADR stabilization weights make it robust under Q’s misspecification. We will further emphasize this advantage of CADR in the paper.
> >
> > - That is an excellent question. The implementation of our estimator is quite straightforward (please refer to lines 240-270 of the $\verb|OPEEstimator|$ class in the code we have submitted in the supplemental material). One additional requirement of our estimator is that it needs not just the propensity score of observation $O(t)$ under policy $g_t$, but also the propensity scores of observations $O(1), …, O(t-1)$ under policy $g_t$. However, this doesn’t need to be computed online at decision time and can be computed after the fact, so long that the trained models used by the contextual bandit algorithm to extract its policy at each time period are stored. The runtime of all estimators is dominated by the sequential sample splitting procedure anyway, so this requirement of our estimator doesn’t make its runtime significantly different from the others. We will make this requirement clearer by calling it out earlier on.
> >
> > Additional Questions:
> > - Thank you for catching this typo, indeed it should be $g^*$ in the equation above and we have fixed it.
> > - Thanks for the clarification question. As Assumption 4 does not require we converge to the *true* outcome regression model, it is easy to truly satisfy it in practice with $\beta=1/2$ by just fitting a parametric model. The better this parametric model, the lower the variance, but it need not be perfect. At the same time, we can also use nonparametric regression estimators that may have slower convergence rates; making such rate assumptions is common to recent semiparametric causal inference literature leveraging blackbox ML to learn nuisances.

---

> > > ### Comment · Reviewer_hHgb · 2021-08-25
> > > **Response to Authors**
> > >
> > > Thank you for your detailed response! I have raised my score as a result.
> > >
> > > I appreciate that you ran experiments with T=1000 and will include them in the final version. I think this makes for a more realistic evaluation. Your explanation of the empirical evaluation being thorough with respect to many datasets, estimators, well-specified/misspecified, ect. makes sense. It would also be nice to see results for another algorithm besides epsilon-greedy (at least for a subset of the cases you consider), to see if results will be at all affected by this. I think this will be straightforward to do. Besides that, I do not have any other issues with the simulations, except perhaps in the presentation if there is space you can mention the variety in the types of simulations you perform and refer to appendix.
> > >
> > > Could you clarify the comments you made about needing to estimate the stabilizing weights? My understanding from the paper is that the weights are importance ratios that are assumed to be known, but that the outcome model and variance must be estimated. Also, the Hadad estimator does have an outcome model (albeit a very simple one).
> > >
> > > In particular, could you elaborate on the following comments to help better explain exactly do your stabilizing weights differ from those of Hadad? I think making this very clear in the paper would help make the distinction between Hadad et al. and your work more clear.
> > >
> > > "Both our method and theirs rely on adaptively weighted DR scores and yield, under conditions, asymptotically normal estimators. However, the way they construct their stabilizing weights appears quite different from ours. In particular, their weights at time t (a) do not incorporate an outcome model and do not rely on it and (b) depend only on the propensity scope. As a result of these different constructions, our analysis and theirs require different sets of conditions."

---

> > > > ### Author Response · Authors · 2021-08-26
> > > > **Follow up to Reviewer hHgb**
> > > >
> > > > Thank you for reading our response carefully. We are very glad to hear that you are supportive of our submission as a result.
> > > >
> > > > You raise very valuable feedback on how we can better clarify the differences to Hadad et al (2019), as well as their more recent follow up Zhan et al. (2021) that was posted after the NeurIPS deadline. We agree that the clearer this point is, the more helpful for the reader.
> > > >
> > > > All three estimators (ours, Hadad et al., 2019, and Zhan et al., 2021) take the same generic form: they are all a reweighted version of the DR estimator, $\sum_{t=1}^T w_t D_t / \sum_{t=1}^T w_t$, where $D_t$ is an approximation of the canonical gradient (aka DR score) at observation $t$ using an estimated outcome model $\hat Q_{t-1}$ (trained on observations $1, …, t-1$; there should be a bar above $Q$ but OpenReview refuses to render it) and $w_t$ is a stabilizing weight. So, for the DR score $D_t$, as you correctly point out, all estimators use an outcome model. In the non-contextual setting of Hadad et al. (2019) the "model" is just an arm's mean reward (no regression function) and one can simply use the sample mean of an arm's outcomes so far. Both we and Zhan et al (2021) use a plug-in regression model that incorporates both context and arm information, and we both require some form of convergence to a limit point but not necessarily consistency. (Although this is easily satisfied by just using $\hat Q_{t-1}=0$, one would usually want a better model for lower variance, but we actually could not find good guarantees in the literature for such convergence when nonparametrically learning a regression function from adaptively collected data, so we actually provide new guarantees for that auxiliary problem in Appendix D; see our Remark 3. Zhan et al., 2021, leave convergence of $\hat Q_{t-1}$ as a high-level assumption in their Assumption 2c and also require stronger sup-norm convergence.)
> > > >
> > > > What is crucially different is the construction of the stabilizing weights $w_t$. Our previous comment mentioning that Hadad et al. (2019) and Zhan et al. (2021) do not use outcome models was referring to the construction of these weights alone, $w_t$, and not to their overall estimator, and we will make this clear when adding discussion on this. In Hadad et al. (2019) the weights $w_t$ ($h_t$ in their notation) are given by Equation (12) in their paper and only depend on the (non-contextual) propensity score ($e_t$ in their notation), the time $t$, and the total number of observations $T$. In Zhan et al. (2021), the weights  $w_t$ ($h_t$ in their notation) are given by Equation (7) in their paper and only depend on the (contextual) propensity score ($e_t$ in their notation), the time $t$, and the target policy ($\pi$ in their notation). So, in both these papers, the construction of the stabilizing weights depends on known quantities that can be directly plugged-in rather than estimated. And, in particular, they do not incorporate the outcome model in the construction of the weights.
> > > >
> > > > Our approach in this paper is to use the variance-stabilization weights for the contextual bandit setting, just like Hadad et al. (2019) do the in the non-contextual bandit setting and Luedtke and van der Laan (2016) do in an iid setting with multiple optimal policies. In particular, we aim to directly estimate the variance $\sigma^2_t$ of $D_t$ by $\hat\sigma^2_t$ and then set the stabilizing weights to $w_t = \hat{\sigma}^{-1}_t$. Crucially, in the contextual setting, the variance $\sigma^2_t$ depends on the outcome model, and as a result we cannot just obtain it by just plugging in known quantities. Instead, we must depend on the learned outcome regression model, and this sequential estimation of weights is a unique technical challenge in the analysis of our estimator that differentiates us from both Hadad et al. (2019) and Zhan et al. (2021).  (Although Zhan et al., 2021, also tackle the contextual bandit setting, they do not use the variance-stabilization weights; correspondingly their results apply in different, complementary settings. Our approach of using variance-stabilization weights is a direct extension of the idea of Hadad et al., 2019 to the contextual setting and the key challenge we tackle in doing this extension is dealing with the need to use sequentially estimated weights. Again we emphasize that Hadad et al., 2019, is set in the non-contextual setting.)
> > > >
> > > > In the non-contextual setting, $\sigma^2_t$ does scale by arm reward variance but this is a constant scalar and it gets cancelled out anyway so Hadad et al., 2019, correctly did not have to worry about it. As we mentioned, even in the non-contextual setting, we are still different from the estimator of Hadad et al., 2019 because of our weight-estimation step before plug-in. However, if instead one used the oracle true variances $\hat\sigma^2_t=\sigma^2_t$, then our estimator would indeed coincide with that of Hadad et al. (2019) in the non-contextual setting (with $\lambda_t$ as in Equation (15) in their paper), as we are both doing variance stabilization. On the other hand, if one (incorrectly and counter to their proposal) used Hadad et al. (2019)'s weights in the contextual setting, one would fail to get stabilization and fail to get correct coverage as their weights no longer proxy the variance in the contextual setting; this is clear from our experiments as shown in the fourth column of Figure 1. Hope this help clarify that point.

---

### Official Review · Reviewer_SpRC · 2021-07-17

**Rating:** 7
**Confidence:** 4

**Summary:**

The current paper concerns inference of quantities such as
treatment effects or policy values from data that is collected
from contextual bandit experiments, Motivated by applications
in healthcare, revenue management or product recommendations
where randomized experimentation is not often the norm.
They isolate a key issue (variance stabilization), identified
in previous works on analysis of adaptively collected data (see
paper for references), and propose estimators for average causal effects of the
`doubly robust` form to overcome this issue. The main result
is a central limit theorem for this estimator, allowing
for frequentist inference (i.e. confidence intervals, p values) in the standard way.
The estimator relies heavily on a sequence of conditional variance estimators, which
they construct using a judicious use of averaging and a base regression model for the outcome (
which may be misspecified even in the limit).


**Limitations And Societal Impact:**

The authors' work allows to use observational data from contextual bandit experiments for statistical inference purposes. In principle this can have potential societal impact, but would be within the context of a specific application (e.g. personalized medicine, policy making...). The authors' conclusion makes some aspects of this apparent.

**Main Review:**

The problem dealt with in the paper is well-motivated and of significant interest
to the NeurIPS community. The paper is, modulo some suggestions below, well-written.
I would advocate its inclusion in the conference program once the major comments
below are addressed.

Major comments:

1. Assumption 6 supersedes Assumption 3, saying the exploration rate needs to be
significantly greater than the worst case 1/sqrt(t) rate. This is crucial to the
construction of the CADR estimator and it is somewhat disingenuous to isolate it thus, unless
you have concrete scenarios where an alternative estimation scheme might exist that
does not require this exploration rate.

2. In the same vein as point 1, I don't think the form of Assumption 3 is sufficient for
the Lindeberg condition argument as laid out. For instance, consider the array Z_{t, T} = \sqrt{t/T} \eps_t
where \eps_t are i.i.d. random signs. This satisfies Assumption 3 but not the Lindeberg condition.
Possibly the argument may be tightened, but at the moment, it seems you need that the exploration is
\Omega(1/sqrt(t)).

Moderate comments:

1. Can you say how your CADR estimator reduces to the variance stabilizing construction in Hadad et al 2019 and, if not, what are the
differences? This would aid intuition for the reader.

2. Can you comment on the efficiency of the constructed estimator, or if we might need to look outside this class of estimators
for the most efficient estimator? Alternatively, a simple setting where it is the most efficient?

3. Some popular policies like UCB are not covered by your scheme given they have trivial propensities, it is good
to mention this. I would also welcome a remark on how Thompson sampling type policies might apply.

Minor comments:

1. L.80 ambiguity with g vs g^*
2. Assumption 5 needs to be restated; perhaps missing punctuation/sentence splitting.
3. Similar ambiguity with Q_{0, X}, Q_{0,Y} vs Q_X, Q_Y in L78, L79. In general, the `0` subscript is a little overused.
4. Define \cal{O} as the space of a, x, y triples (I assumed this and faced no obvious errors)
5. The use of a \bar{} is a bit inconsistent: \bar{O}(t), \bar{Q} and \bar{\Phi} mean different things (a sequence,
an empirical average and just another thing). This causes some unnecessary readability issues.




**Time Spent Reviewing:**

4

---

> ### Author Response · Authors · 2021-08-10
> **Authors' Response to Reviewer SpRC**
>
> Thank you for taking the time to review our paper and we are very glad to read that you appreciated the value of our paper’s contribution to the community. We address all your comments below in the order they appear.
>
> Major Comments:
> 1. Thank you for raising this point. Of course, we do not mean to hide this and we both state “we require a condition on the exploration rate that is stronger than Assumption 3” on line 220 and require it in our Theorem 2 and Corollary 1. Our aim was only to clearly separate the two independent lines of argument: constructing a generic stabilized estimator and actually realizing the stabilized estimator with our variance estimators. To address this concern, we will clearly foreshadow the need later-on to strengthen Assumption 3 when we present the latter.
> 2. Thank you for your comment and we hope to clear any misunderstanding here. First, “that the exploration is $\Omega(1/\sqrt{t})$” is *exactly* what Assumption 3 is requiring. It states that for some constant $c$, any action has probability at least $c/\sqrt{t}$ conditioned on any $X=x$. Second, the array you instantiated does seem to satisfy the Lindeberg condition. Perhaps we misunderstood your point, in which case we would be grateful for a clarification. But, it seems we are in agreement: we need at least $1/\sqrt{t}$ exploration, and we indeed also require at least $1/\sqrt{t}$ exploration.
>
> Moderate Comments:
> 1. Thank you for raising this point and helping us clarify the discussion of that paper. As stated in their work, the estimator of Hadad et al., 2019 satisfies the unbiasedness and asymptotic normality properties under non-contextual, multi-armed bandit data. In order to design an estimator that satisfies unbiasedness and asymptotic normality properties under contextual bandit data, we have to construct weights that depend both on the arm and the context. The weighting scheme we propose is more general than Hadad et al. 2019 and therefore does not reduce to theirs, because our estimator incorporates the presence of a contextual propensity score from the logging policy and of a contextual outcome model used by the estimator when estimating the local variance of each datapoint’s doubly robust score. On the other hand, in Hadad et al. 2019 the local variance is proxied only by the non-contextual propensity score. The experimental section further establishes how different CADR is from the estimator in Hadad et al., 2019 as the latter achieves much lower coverage when applied on data collected by a contextual bandit than the nominal 95% that is achieved by CADR. In case your question was instead meant about whether our estimator reduces to that of Hadad et al. 2019 when ours is applied in the non-contextual setting, the answer is still no because of the way we estimate the variance so that it works also in the contextual setting means our estimator is different even in the non-contextual setting.
> 2. In the general dependent-data setting, there is no efficiency theory of the OPE problem, to the best of our knowledge. So, the notion of “most efficient estimator” is not well-defined here. We can, however, say the following in terms of efficiency in some simple settings, as suggested. If the data happens to be iid and our outcome model is consistent, then our estimator does attain the corresponding efficiency bound for the iid setting. Moreover, if the data is adaptively collected but the logging policy converges to a limit policy with a positive probability of playing any one action in any context (non-decaying exploration), then our estimator attains the efficiency bound corresponding to iid data from this limit policy. We will point out these settings and also the lack of general non-iid efficiency theory.
> 3. That is a great point; thanks for bringing it up. Indeed, both our paper and several related works in the de-biasing literature (van der Laan 2008, Hadad et al. 2019, Zhang et al. 2021) require the propensity score to be non-trivial, and UCB does not fit under this framework, being a non-stochastic algorithm. $\epsilon$-greedy and Thompson sampling fit better in the framework considered by our and these papers. UCB may require future methodology and/or analysis. We will make this point much more explicit.
>
> Minor Comments:
>
> Thanks for flagging these readability improvements; it indicates a careful review of our paper, for which we are thankful, and we will address them in our revision.

---

> > ### Comment · Reviewer_SpRC · 2021-08-27
> > **Followup**
> >
> > Re: your response to major comment:
> >
> > I think partially there is a misunderstanding of notation: $\Omega(1/\sqrt{t})$ vs $\omega(1/\sqrt{t})$. You use the first which allows the rate to be proportional to $1/sqrt{t}$ while the latter means it should be strictly larger i.e. $exploration(t) / t^{-1/2} \to \infty$ and part of the proof suggests this is what you want (e.g. when you state 1/\delta_t = o(\sqrt t)). This apart, my example still makes the proof incomplete. With $Z_{t, T} = \sqrt{t,/T} \epsilon_t$ where $\epsilon_t$ is a random sign. The set of $Z_{t,T}$'s that can exceed a fixed $\epsilon >0$ are the ones where $t$ sufficiently large so that $\sqrt{t/T} > \epsilon$, i.e. $t > \epsilon^2 T$. Using this the LHS of the lindeberg condition reads:
> > $$\sum_{t > 1} E (Z_{t, T}^2 \mathbb{I}(Z_{t,T} > \epsilon) ) = \sum_{t > \epsilon^2 T} \frac{t}{T} P(\epsilon_t = 1) = \frac{ \sum_{t > \epsilon^2 T} t }{2T}$$
> > The latter is linear in T, quite far from converging to 0. What is going wrong here is that of $Z_{t, T}$'s a bunch of them are order one, instead of being order $1/\sqrt T$ which is what you would like and also what the conditional variance calculation suggests. I think that (with the exploration misunderstanding above mentioned cleared) there should be a proof of the lindeberg condition. However the one you provide is not enough.
> >
> > Re: UCB and Thompson sampling:
> >
> > I am not aware of an explicit proof that Thompson sampling indeed has an asymptotic exploration rate like (say) $1/\sqrt t$ although the regret guarantees and minimax bounds might be conceivably combined to yield such a rate. If the authors are aware of one such, it would be useful to include that here. There are other methods (e.g. the W-decorrelation method in https://arxiv.org/abs/1712.06695 that allow trivial propensities).

---

> > > ### Author Response · Authors · 2021-08-31
> > > **Re-response to Reviewer SpRC**
> > >
> > > (fixed a typo below re $\sigma_{0,t}\gtrsim \delta_t^{-1/2}$)
> > >
> > > Re Lindeberg condition:
> > >
> > > Thank you for clarifying your point. You are right that the way we write our proof we do need $\omega$ and not $\Omega$, but at the same time one does not really actually need Assumption 3 for Theorem 1. Let us explain.
> > >
> > > First, thanks for pointing us to where we write $\delta_t^{-1}=o(t^{1/2})$ in the proof (and, as you guessed, there's also an errant minus sign typo we need to remove). You are absolutely right that, as written, we are making a stronger assumption about the exploration than stated in the Theorem, the difference between $\Omega$ and $\omega$, as you write. We were indeed not careful here because, at any rate, we later strengthened the exploration requirement in Assumption 6. Our only intent was to prove a generic form of the CLT under stabilization with estimated variances. Since changing $\Omega$ to $\omega$ in Assumption 3 will still make it a consequence of Assumption 6, this is an easy fix that addresses this issue. Thank you so much for pointing it out to us! We greatly appreciate your close reading of our paper -- this is incredibly helpful -- both to fix inaccuracies and to make us think harder about what is really needed here.
> > >
> > > The thing is that, if one were truly interested in a minimal form of Theorem 1, Assumption 3 is not even really needed (even in the $\omega$ form). At the end of the proof of Theorem 1, where you point to, we use quite loose bounds to argue the Lingeberg condition simply because we assume a stronger exploration later anyway. But, one can satisfy Lindeberg here with a much more lax constraint on the exploration rate, essentially just $\omega(1/t)$. Since we're scaling each term by the (consistently estimated) standard deviation, we are accounting for potentially diminishing variation. Essentially, in line 497 in the supplement, the $\hat\sigma_t^{-1}$ term should cancel with one $\delta_t^{-1/2}$ term, so we really only need $\delta_t=\omega(1/t)$ (currently we just bound the former). For example, one sufficient assumption would simply be that $\sigma_{0,t}\gtrsim \delta_t^{-1/2}$, such as would occur if (a) $g^*$ placed any mass on actions with decaying propensity (e.g., any suboptimal actions) and (b) the conditional variance of outcomes is lower bounded, $\mathrm{Var}(Y\mid X,A)\gtrsim 1$. Then, assuming only a $\omega(1/t)$ exploration rate, the Lindeberg condition would be immediately satisfied. Similarly, if we just assume $\sigma_{0,t}=\omega(1)$, as one should certainly expect if exploration is diminishing, then the Lindeberg condition is satisfied even with the current version of Assumption 3 (i.e., $\Omega$). There may be other minimal assumptions just so to satisfy the Lindberg condition. The really important part of our proof is our Cesaro summation argument where we show $V_T\to1$, as this is truly the crux of where we deal with stabilization by _estimated_ standard deviations. (As an aside, note that in the display equation on line 462 in the supplement, the first $\frac1T$ is a typo and should not be there; we'll fix that.)
> > >
> > > We want to thank you for pushing us to consider more carefully our proof of Theorem 1 and for pointing us to our small error. As we explain, this is somewhat a moot point, but it does require a fix. One option is to use the $\omega$ form, especially since we assume a stronger exploration later on anyway that implies the $\omega$ form of the assumption. Another option is to introduce other conditions such as on the scaling of the variance that give Theorem 1 with just $\omega(1/t)$ exploration. Our proposal is to do the former, as it is minimal, fully compatible with our later assumptions, and does not introduce additional assumptions we do not later use, such as growing $\sigma_{0,t}$. At the same time, we will make sure to comment and make clear in the exposition how Assumption 3 is not really central for Theorem 1 to hold and one can do much more lax requirements on the exploration rate for this part of the results.
> > >
> > > Finally, we want to provide more detail on why Lindeberg condition does indeed hold for the array you gave in your original report: $Z_{t,T}=\sqrt{t/T}\epsilon_t$ with $\epsilon_t$ iid Rademacher. In the Lindeberg condition (https://en.wikipedia.org/wiki/Lindeberg%27s_condition), we always normalize by the sum of variances, which for your example is exactly $2\sum_{t=1}^T(t/T)=T+1$. Therefore, the Lindeberg condition for your example is $\frac{1}{T+1}\sum_{t=1}^T\mathbb E[\frac{t}{T}\mathbb I[\sqrt{\frac{t}{T}}\geq \epsilon\sqrt{T+1}]]$, which indeed goes to 0. Note that on lines 461 and 477 in the supplement we did not include the sum of variances $V_T$ in our statement of the Lindeberg condition because by construction of our $Z_{t,T}$ we scaled things to already ensure that $V_T\to_p1$ so that it makes no appearance in the condition as we write it (this is in particular a common form when dealing with the martingale version, as we are, where we need to deal with the limit of the sum of variances). In your example, only if we set $Z_{t,T}'=\frac{Z_{t,T}}{\sqrt{T+1}}$ so that the sum of variances is 1, can we write the (completely equivalent) Lindeberg condition as $\sum_{t=1}^T\mathbb E[(Z_{t,T}')^2\mathbb I[|Z_{t,T}'|>\epsilon]]$. Perhaps that was the source of confusion about the Lindeberg condition here.
> > >
> > >
> > > Re: UCB and Thompson sampling:
> > >
> > > You are right, and we are not aware of such either. We will make that clear. That said, as the above makes clear regarding Theorem 1 at least, the exploration rate is not so crucial. At the same time, our Theorem 2 does make a requirement on exploration and we do rely on it in a meaningful way there. It may be worth investigating -- especially empirically -- to what extent these requirements are truly necessary to obtain consistent variance estimation and to what extent they are just "high-level" assumptions since our method itself does not explicitly rely on them, only the analysis; we will note this as important next steps.
> > >
> > > Thank you for the Deshpande et al. reference on inference on linear models from adaptive data. It is extremely relevant and we will make sure to cite and review it in our literature review.

---

### Comment · Reviewer_2WyM · 2021-08-26

Dear all,

I have a serious concern that I would like the authors and reviewers to confirm.
As I have already explained in several replies, it seems that Assumptions 5 and 6 constrain the bandit problem that the proposed method can be used.
- Under these assumptions, the problem suffers a $T^{2/3}$ regret even in the ideal situation. We agree with the authors on this point.
- This constraint becomes more difficult depending on the problem (Assumption 5). This is very different from the case of Hadad et al. (2020), where a bandit problem (without contextual information) with a regret close to $\log T$ can be evaluated independently of the problem (Theorem 3 in Hadad et al. (2020)).
- It is not clear what kind of bandit problems can be evaluated under Assumption 5 and Assumption 6.
- In the limited case of nonparametric bandits, which may suffer from a $T^{2/3}$ regret, the $p$ in Assumption 5 will become large, requiring a slower exploration rate from Assumption 6. Under these assumptions, it is unclear how much of the regret will be incurred.
- It is also unclear whether the experimental setup of the paper satisfies Assumptions 5 and 6. I am concerned that these assumptions may be violated. The paper sets an exploration rate of at least $t^{-1/3}$, which implies that $p$ is zero unless the expansion rate of the bandit algorithm is sufficiently slow, but $p=0$ contradicts with Assumption 6.

I believe that mathematical proof is needed for justifying Assumption 5 and 6.
Also, these are not explained in the introduction and the problem formulation in this paper, and I think that the authors need to explain these limitation there.

Also, I want to discuss the novelty of this paper.
- I think that Hadad et al. (2020)'s contribution was to simplify Luedtke and van der Laan (2016) by omitting the variance estimation and not considering the context. Luedtke and van der Laan (2016) used a rolling window (sample splitting) to estimate the variance, but avoiding it would require assumptions like Assumption 5 and 6.
-  Therefore, if the authors use assumptions like Assumptions 5 and 6 to estimated the variance without a rolling-window, I think that the authors need to provide a justification for them.
-  Also, as I already mentioned, I think that the authors' results are a bit different from Hadad et al. (2020), where we can arbitrarily evaluate the bandit algorithm with a $\log T$ regret, while the authors' proposed method is applicable only to the bandit algorithm with a $T^{2/3}$ regret even in the ideal conditions (Assumption 5).

The novelty may be a bit subjective issue, so not a core.
Therefore, if the other reviewers agree that this study is novel, I have no objection.
However, I think that theoretical justification is a critical issue.

I feel that there is a big gap between the claims of the paper and the results as it is now.
Readers will think that the proposed method can be widely used for the linear bandit after reading the beginning of the paper and problem setting.
And they will not consider that the targeted bandit algorithms suffer from at least the $T^{2/3}$ regret.
In order for this paper to be published, I strongly believe that during the reviewing period, the authors need to prove that the proposed method can be applied to some bandit algorithms, such as Rigollet and Zeevi (2010), which is cited by the authors in their reply to my comment.
In addition, I think it is desirable to get agreement from the reviewers on which bandit algorithms the proposed method can and cannot be applied for and whether the experimental setup of the paper meets the assumptions, etc.

By clarifying them, I hope to avoid misunderstandings of readers and subsequent researchers.
I would like authors and reviewers to check these points.
And if I am wrong, please tell me about it.

---

### Decision · Program_Chairs · 2021-09-27

**Decision:**

Accept (Poster)

**Comment:**

Thank you to the authors and the reviewers for their contributions! There was a long discussion on this paper, and we sought external opinions to add clarifications. The final consensus is that the paper makes an interesting technical contribution to the literature on inference over adaptively collected data, specifically via contextual bandits. However, the paper is not very clear in terms of its key results and their limitations, and I urge the authors to clarify the exposition in the next version. Key issues have already been noted in the discussion period; one reviewer also specifically requested clarifications on whether Assumptions 5 and 6 hold in the experiments (i.e., whether the \epsilon is decaying too fast) or to adapt the experiment parameters accordingly.